# The TRIM37 variant rs57141087 contributes to triple-negative breast cancer outcomes in Black women

Rachisan Djiake Tihagam[1,6], Song Lou[1,6], Yuanji Zhao[1], Kammi Song-Yan Liu [ID][1], Arjun Tushir Singh[1,5], Bon Il Koo[1], Piotr Przanowski[2], Jie Li[3], Xiaosong Huang[4], Hong Li [ID][4], Jogender Tushir-Singh[1,5], Laura Fejerman[4] & Sanchita Bhatnagar [ID][1,5 ✉]

## Abstract

Triple-negative breast cancer (TNBC) disproportionately affects younger Black women, who show more aggressive phenotypes and poorer outcomes than women of other racial identities. While the impact of socioenvironmental inequities within and beyond health systems is well documented, the genetic influence in TNBC-associated racial disparities remains elusive. Here, we report that cancer-free breast tissue from Black women expresses *TRIM37* at a significantly higher level relative to White women. A reporter-based screen for regulatory variants identifies a non-coding risk variant rs57141087 in the 5′ gene upstream region of the TRIM37 locus with enhancer activity. Mechanistically, rs57141087 increases enhancer–promoter interactions through NRF1, resulting in stronger TRIM37 promoter activity. Phenotypically, high *TRIM37* levels drive neoplastic transformations in immortalized breast epithelial cells. Finally, context-dependent *TRIM37* expression reveals that early-stage *TRIM37* levels affect the initiation and trajectory of breast cancer progression. Together, our results indicate a genotype-informed association of oncogenic *TRIM37* with TNBC risk in Black women and implicate *TRIM37* as a predictive biomarker to better identify patients at risk of aggressive TNBC.

**Keywords** Triple-negative Breast Cancer (TNBC); TRIM37; Black Women; SNP; NRF1
**Subject Categories** Cancer; Chromatin, Transcription & Genomics; Genetics, Gene Therapy & Genetic Disease

## Introduction

Breast cancer is the second most common cancer in women, with 297,790 estimated new cases in the United States in 2023 (Siegel et al, 2023). The relatively high 5-year overall survival of 90.8%

indicates that the disease is curable. The advanced stage at the diagnosis and limited access to treatment options contribute to 43,000 annual deaths, but triple-negative breast cancer (TNBC) remains the major cause of mortality in breast cancer patients. TNBCs constitute 12–17% of all breast cancer subtypes, affecting younger women more frequently than patients in other age groups (Almansour, 2022). Histologically, TNBCs are characterized by less than 1% cellular expression of progesterone and estrogen receptors and 0 and 1+ expression of human growth factor receptor 2. Clinically, TNBC tumors are associated with later stage at diagnosis, increased mortality, recurrence, metastatic patterns, and poor prognosis (Dent et al, 2007). Given the nonavailability of targeted therapeutic options and the highly invasive nature of tumors, TNBC is usually managed by conventional chemotherapy, which is often accompanied by resistance, relapse, and severe side effects.

TNBC clinical outcomes are shaped by lifestyle, genetics, epidemiologic, and cultural factors; however, the disease etiology might also depend on biogeographical ancestry. Globally, TNBC mortality and prevalence are highest among women with African ancestry (Jiagge et al, 2016; Martini et al, 2022b; Newman et al, 2019) who belong to geographically diversified regions, such as Europe, South Africa, and admixed African American populations in the United States (Morris et al, 2007; Rapiti et al, 2017). Whether earlier onset or advanced stage at the time of diagnosis, an aggressive tumor phenotype is a characteristic feature of TNBC in Black women (BW) (Garlapati et al, 2019; Newman and Kaljee, 2017; Siddharth and Sharma, 2018). Consequentially, a 5-year survival rate for TNBC in BW is only 14% compared to 36% in women from other racial categories (Siddharth and Sharma, 2018). We argue that genetic drivers may predispose BW to aggressive TNBC. To this point, several groups utilized transcriptomic profiling of tumor tissues that identified racially and ethnically segregating genetic features. A racial stratification of genome-wide association study (GWAS) data for women with African and European ancestry identified a common risk variant at the TERT-CLPTM1L locus, which increased genetic predisposition to TNBC (Haiman et al, 2011). Martini et al linked racial identity-specific immune response signatures to outcome disparities in tumors

[1]Department of Medical Microbiology and Immunology, University of California Davis School of Medicine, Davis, CA 95616, USA. [2]Department of Biochemistry and Molecular Genetics, University of Virginia School of Medicine, Charlottesville, VA 22908, USA. [3]UC Davis Bioinformatics Core, University of California at Davis, Davis, CA 95817, USA. [4]Department of Public Health Sciences, University of California Davis, Davis, CA 95817, USA. [5]Present address: UC Davis Comprehensive Cancer Center, University of California Davis, Davis, CA 95817, USA. [6]These authors contributed equally: Rachisan Djiake Tihagam, Song Lou. ✉E-mail: sbbhatnagar@ucdavis.edu

targeted with immunotherapies (Martini et al, 2022a). Another study linked genes, such as *ACOX2*, *CRYBB2*, and *MUC1*, to increased luminal A and basal-like breast cancer risk and survival disparity (Parada et al, 2017). A comparative transcriptomic analysis of TNBC cellular models identified differences in GLI1 and Notch1 pathways predisposing African American women to aggressive growth and metastatic TNBC progression (Siddharth et al, 2021). While these studies highlighted inherent genetic and epigenetic differences that vary by racial identity or ethnicity, the functional association of the gene with breast cancer risk or racial disparity is lacking. Accordingly, these reports underscore the need for mechanistic studies to demonstrate the contribution of genetic factors in breast cancer onset and progression. The outcomes from such studies could inform the methods for early detection, intervention, and prevention of TNBC.

An important benefit of understanding the earliest changes of cancer initiation is the potential to identify biomarkers for breast cancer susceptibility, provide health benefits to the general population, and opportunities for cancer prevention. Over the last decade, greater than 100 cancer predisposition genes conferring a higher risk have been identified through genome-wide mutational analysis, linkage analysis, and candidate gene analysis (Rahman, 2014). For example, the genomic loci harboring well-characterized breast cancer genes, such as BRCA1, TP53, BRIP1, and PTEN, are associated with breast cancer risk (Breast Cancer Association et al, 2021). Another longitudinal study identified an aberrant transcriptional regulation of 69 genes in the normal, cancer-free breast tissue of women at high risk of breast cancer (Marino et al, 2022). Together, these studies provided critical insight into the genetic and epigenetic changes that predispose to breast cancer and accompany the early stages of carcinogenesis.

Our previous studies comprehensively described tripartite motif-containing protein 37 (TRIM37) function as a novel breast cancer oncoprotein (Bhatnagar et al, 2014; Bhatnagar and Green, 2015). *TRIM37* is highly expressed in breast cancer tissue and is associated with poor overall patient survival (Bhatnagar et al, 2014). TRIM37 represses gene transcription by mono-ubiquitinating histone H2A at Lys119 (H2Aub) at various genomic loci, including tumor suppressors and pro-apoptotic genes (Bhatnagar et al, 2014). *TRIM37* overexpression renders RAS-transformed, premalignant breast cells tumorigenic, and inhibition of TRIM37 function reduces tumor growth in xenograft and patient-derived mouse models (Bhatnagar et al, 2014). We also recently demonstrated that *TRIM37* drives aggressive TNBC biology through a transcriptional program favoring resistance to chemotherapy and metastasis (Przanowski et al, 2020). However, in contrast to established tumors, the high-resolution functional characterization of TRIM37 during the initial steps of neoplastic transformations before disease manifests is still limited.

Here, we describe the mechanism by which TRIM37 associates with breast cancer risk and aggressive phenotype in BW. We show that *TRIM37* upregulation in BW breast epithelial cellular systems triggers transcriptional reprogramming, favoring loss of identity and acquisition of stemness and mesenchymal phenotypes. Using comprehensive genomic and functional analysis, we identify an ancestry-specific, non-coding regulatory variant rs57141087 predominant in BW that modulates *TRIM37* levels by increasing enhancer–promoter contact via NRF1. We show that *TRIM37* overexpression in the early stages confers neoplastic

transformations, accelerates tumorigenesis, and drives cells into a fate of malignancy. Thus, we provide functional proof for the effects of *TRIM37* in TNBC etiology that are determined by the SNP rs57141087 in BW.

# Results

## African ancestry influences *TRIM37* expression in TNBC and cancer-free breast tissue

Almost 30% of all breast cancer cases in BW are predominantly TNBC (Siddharth and Sharma, 2018). We compared the TNBC incidence and survival rates in BW and White women (WW), excluding Hispanics, by interrogating the Surveillance, Epidemiology, and End Results Program (SEER) for the years 2010 through 2020. The longitudinal analysis over ten years revealed that approximately two-fold higher number of BW was diagnosed with TNBC than WW (Fig. 1A). Similar analysis across different age groups (15–39, 40–74, 75+ ages) showed significantly higher TNBC incidence (Fig. 1B) and mortality rate (Fig. 1C) in BW than WW over ten years (2010–2020). These results sharply contrast the observed decrease in breast cancer incidence in recent years (Fig. EV1A), indicating that BW remains susceptible to aggressive TNBC phenotype despite improved treatment and access to health care.

Previous studies have identified intrinsic molecular differences in TNBC tumors accounting for aggressive progression in BW compared to WW (Siddharth et al, 2021). We, therefore, queried *TRIM37* expression in different stages of TNBC stratified by self-reported BW and WW using datasets available through GSE142731 (Data ref: Saleh et al, 2021), PRJNA704957 (Data ref: NCBI Sequence Read Archive PRJNA704957, 2021), and TCGA (Data ref: The Cancer Genome Atlas Program (TCGA-Breast)). A total of 319 patients were included in the analysis, and the tumor characteristics are described in Table EV1. Interestingly, the meta-analysis revealed ~1.63-fold higher *TRIM37* expression in early histological Stage I TNBC tumors from BW than in WW (Fig. 1D), which was not the case for Stage II–IV (Fig. 1E). Univariate analysis confirmed the association between *TRIM37* expression in the Stage I TNBC tumors with racial identity ($P = 0.0174$, Fig. EV1B). Next, we assessed to what extent the early-stage differences in *TRIM37* expression could explain the disparity in the overall survival of TNBC patients. Indeed, BW with TNBC tumors expressing high *TRIM37* showed poor overall survival, with a median survival of ~114 months, relative to WW, with a median survival of ~245 months ($P = 0.0011$; Fig. 1F). Notably, no significant differences in overall survival were observed for low *TRIM37*-expressing TNBC tumors from BW and WW (Fig. EV1C).

Next, to determine whether there is an independent contribution of *TRIM37* to the neoplastic transformation in BW, we interrogated *TRIM37* expression in normal, cancer-free breast tissue. Out of 549 samples analyzed, 119 were BW, 430 were WW, 115 were below or equal to the age of 35, and 434 were above the age of 35 (Table EV1). Surprisingly, the meta-analysis revealed significantly higher *TRIM37* levels in the breast tissue of BW relative to WW (Fig. 1G). No difference in *TRIM37* levels was observed when women were stratified by age (Fig. EV1D). Using the $\chi^2$ test, we find that higher *TRIM37* levels in the breast tissue

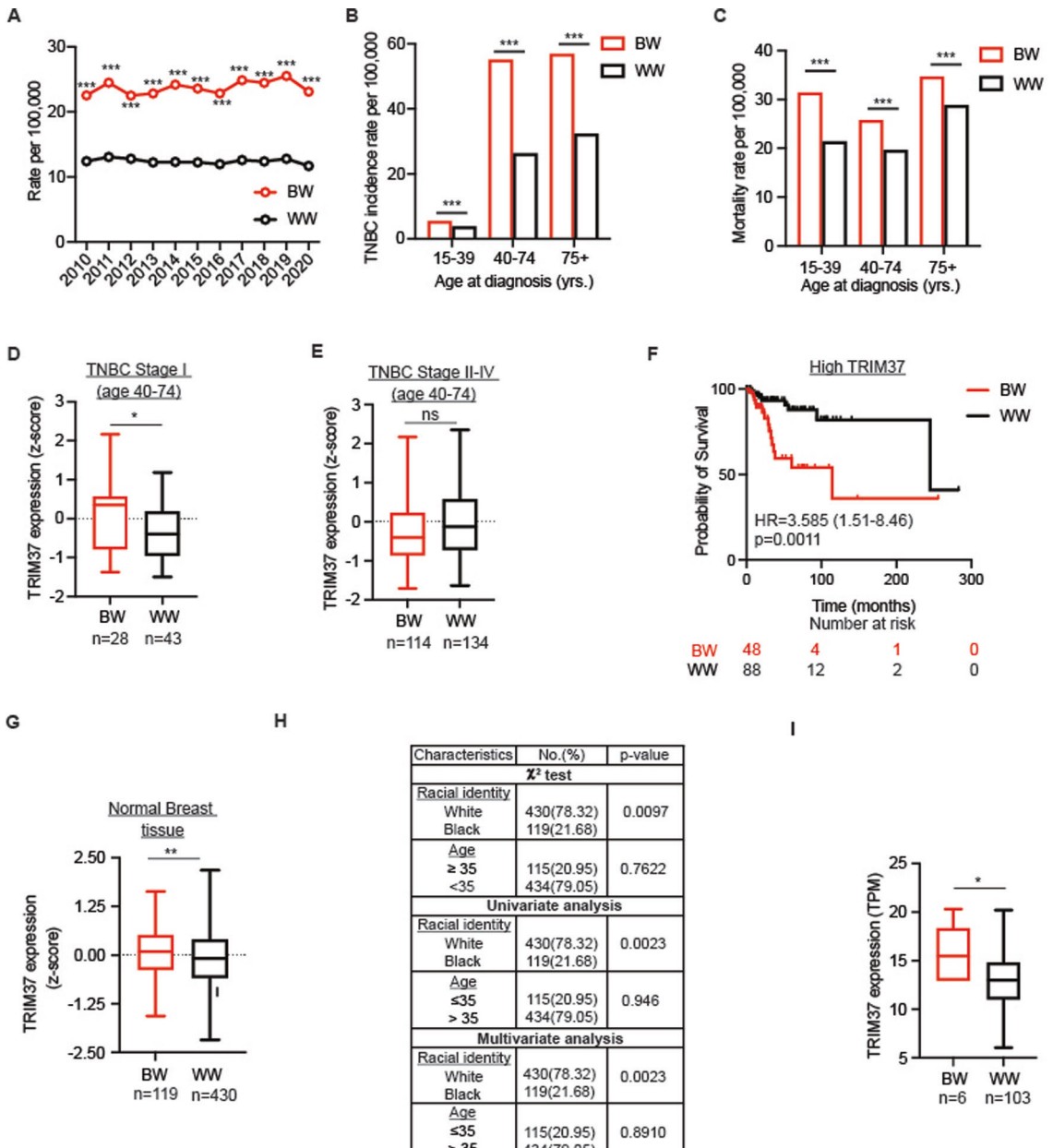

Figure 1.    Preferential upregulation of *TRIM37* in early-stage TNBC tumor and cancer-free breast tissue of Black women (BW).

(A) The incidence rate of TNBC in female BW and White women (WW), excluding Hispanics, from 2010 to 2020 Surveillance, Epidemiology, and End Results (SEER) data. *P* value for years 2010–2020 ***$P < 2.2 \times 10^{-16}$, proportion test. (B, C) TNBC incidence (B) and mortality rate (C) per 100,000 for BW and WW TNBC patients stratified by age groups 15–39, 40–74, and above 75 years of age from 2010 to 2020. *P* values for the incidence and mortality rates in the different groups are ***$<2.2 \times 10^{-16}$, proportion test. (D, E) Box plot for racial identity-specific *TRIM37* expression in Stage I (D; $n = 71$) and Stage (II–IV; $n = 248$) TNBC patients (GSE142731, Data ref: Saleh et al, 2021; PRJNA704957, Data ref: NCBI Sequence Read Archive PRJNA704957, 2021, and TCGA, Data ref: The Cancer Genome Atlas Program (TCGA-Breast). Stage I, *$P = 0.029$, and Stage II–IV, $^{ns}P = 0.058$, unpaired *t* test. The boxed areas span the first to the third quartile, with the central line representing the median expression changes for each group. Outliers from the boxplots are not displayed. The whiskers represent the 15th and 85th percentiles. (F) Kaplan–Meier survival curve showing overall survival for the BW and WW TNBC patients with high *TRIM37* expression (GSE39004; Data ref: Tang et al, 2018", TCGA, Data ref: The Cancer Genome Atlas Program (TCGA-Breast), and GSE18229, Data ref: Prat et al, 2010). **$P = 0.0011$, log-rank test. The number of surviving patients at 0, 100, 200, and 300-month time points is indicated below the graph. (G) Box plot for *TRIM37* expression in the normal breast tissue of BW ($n = 119$) and WW ($n = 430$) (GSE164641, Data ref: Marino et al, 2022; GSE111601, Data ref: Sun et al, 2018; GTEX, Data ref: The Genotype-Tissue Expression (GTEx), and TCGA, Data ref: The Cancer Genome Atlas Program (TCGA-Breast). **$P = 0.0029$, unpaired *t* test. The boxed areas span the first to the third quartile, with the central line representing the median expression changes for each group. Outliers from the boxplots are not displayed. The whiskers represent the 15th and 85th percentiles. (H) Association of *TRIM37* expression in the cancer-free breast tissue with racial identity and age by chi-squared, univariate, and multivariate analyses. (I) Box plot of *TRIM37* transcript level in the normal breast tissue of BW ($n = 6$) and WW ($n = 103$) ancestry confirmed by admixture analysis (TCGA, Data ref: The Cancer Genome Atlas Program (TCGA-Breast)). *$P = 0.049$, Wilcoxon test. The boxed areas span the first to the third quartile, with the central line representing the median expression changes for each group. Outliers from the boxplots are not displayed. The whiskers represent the 15th and 85th percentiles. (J) Box plot for *TRIM37* transcript levels in breast tissues from women at average risk ($n = 106$) or high risk ($n = 72$) of developing breast cancer (GSE164641, Data ref: Marino et al, 2022). **$P = 0.0022$, unpaired *t* test. The boxed areas span the first to the third quartile, with the central line representing the median expression changes for each group. Outliers from the boxplots are not displayed. The whiskers represent the 15th and 85th percentiles. (K) Association of *TRIM37* expression in breast tissues from women at average ($n = 106$) or high risk ($n = 72$) of developing breast cancer with clinicopathological variables (breast cancer risk, age, racial identity, BMI, menopausal status, and parity) by chi-squared, univariate, and multivariate analyses. Source data are available online for this figure.

were associated with BW (Fig. 1H). Univariate analysis demonstrated a significant association between *TRIM37* expression and racial identity but not in age (Fig. 1H). Further, in the multivariate linear regression test, relationships were examined between the *TRIM37* expression on one hand and the racial identity and age on the other. The test demonstrated that higher *TRIM37* levels were more likely to be associated with BW than WW ($P = 0.00233$, Fig. 1H). Other factors, such as age, did not show a significant association with *TRIM37* levels (Fig. 1H).

We note that the genomic complexities of admixed populations may obscure the race-specific molecular differences in self-reported ethnicity. We, therefore, estimated the genomic ancestry for each sample in the TCGA cohort, which indicated greater than 80% African ancestry in the six self-reported BW samples and greater than 70% European ancestry in 103 of self-reported WW (Fig. EV1E). To investigate the ancestry-specific *TRIM37* expression, we isolated analyses to samples with greater than 70% African or European ancestry. We consistently find that the breast tissue of BW expresses significantly higher *TRIM37* than WW (Fig. 1I). The univariate analysis confirmed a significant association of *TRIM37* with ancestry ($P = 0.050$). These results revealed African ancestry-specific, distinct *TRIM37* expression in normal, cancer-free breast tissue.

Given that TRIM37 is associated with genomic instability, tumor progression, and poor prognosis (Bhatnagar et al, 2014; Przanowski et al, 2020), we hypothesized that *TRIM37* levels in breast tissue could likely inform the clinical outcome. We, therefore, examined *TRIM37* expression in cancer-free breast tissue from women at higher lifetime risk of developing breast cancer. Of 178 healthy, cancer-free breast tissue from women of median age ~40 years, 72 were grouped into a higher lifetime risk of developing breast cancer based on the Tyrer-Cuzik model (score ≥20%) (Marino et al, 2022). Out of 178 samples, 114 were premenopausal, 64 were post-menopausal women, 135 had carried at least one pregnancy, and 68 had a BMI in a healthy weight range (Table EV1). Thirty women were Black, and 148 identified as White.

Notably, *TRIM37* was expressed significantly higher in the breast tissue of women allocated to a high-risk group ($n = 72$) than in the average-risk group ($n = 106$, Fig. 1J). Consistent with previous results, we find higher *TRIM37* levels in BW relative to WW (Fig. EV1F). No significant differences in *TRIM37* levels were identified when samples were stratified by age, BMI, parity, and menstrual status (Fig. EV1G–J).

Next, using the $\chi^2$ test, we asked whether *TRIM37* expression is associated with age, menopause status, breast cancer risk, racial identity, parity status, and BMI. The tests proved that breast cancer risk was significantly higher in samples with high *TRIM37* expression (Fig. 1K). To better understand the features of young women's breast tissue predictive of clinical outcome, the univariate and multivariate analyses combining clinicopathologic variables with *TRIM37* expression value were performed. The univariate analysis confirmed the significant association between *TRIM37* and breast cancer risk ($P = 0.00217$) and obesity ($P = 0.0355$, Fig. 1K). We noted near-significant associations of *TRIM37* expression with parity ($P = 0.0565$) and racial identity ($P = 0.09851$), in part due to a smaller sample size (Fig. 1K). In multivariate analyses, *TRIM37* correlated significantly with breast cancer risk ($P = 0.0005$), BMI ($P = 0.003$), and near significantly with racial identity ($P = 0.0568$; Fig. 1K). These results raised a possibility that early-stage high *TRIM37* observed in the breast tissue of BW could give cancer cells a "head start," impacting the disease trajectory and outcome.

## African ancestry influences *TRIM37*-associated transcriptional signatures in normal, cancer-free breast tissue

*TRIM37* promotes tumorigenesis through its association with the polycomb proteins, which are repressive complexes for chromatin-based gene regulation (Bhatnagar et al, 2014). Thus, *TRIM37* levels are expected to drive distinct gene expression changes referred to as TRIM37-associated transcriptional signatures (TRIM37-TS). We rationalized that the TRIM37-TS (Bhatnagar et al, 2014;

Przanowski et al, 2020) could better predict TNBC onset and phenotype than TRIM37 alone. To test this idea, we investigated transcriptional profiles of the paraffin-archived normal breast tissue samples that were obtained from parous, premenopausal WW, and BW (The Susan G. Komen Tissue Bank; Indiana University Simon Comprehensive Cancer Center). These women were cancer-free, and donors were recruited under a protocol approved by the Indiana University Institutional Review Board (#101103097) and completed questionnaires with reproductive histories, demographic information, early life, and lifestyle habits (Table EV2).

The differential gene expression (DEG) analysis revealed 2714 upregulated and 2239 downregulated genes at an FDR cutoff of <0.1 in BW compared with WW breast tissues (Figs. 2A and EV2A). Interestingly, among the 4953 DEGs, 609 were previously identified as TRIM37-TS (GSE136617, Data ref: Przanowski et al, 2020), revealing TRIM37-dependent molecular alterations in BW (Fig. 2B). Hierarchical clustering showed separation of transcriptomic profiles by ancestry into two distinct racial identity-specific clusters (Fig. 2C). To explore TRIM37-driven functional pathways in BW, we performed Gene Set Enrichment Analysis (GSEA) for upregulated genes, which identified the cancer stem cell (CSC) and epithelial to mesenchymal transitional (EMT) among the enriched gene sets (Figs. EV2B and 2D). The integration of CSC and EMT gene signatures showed a strong association with poor survival for TNBC patients with a hazard ratio of 1.88 ($P = 0.033$) and 1.9 ($P = 0.031$), respectively (Fig. 2E).

To experimentally validate ancestry-specific TRIM37-TS, we stably over-expressed *TRIM37* in immortalized breast epithelial cells derived from healthy women of African ancestry (KTB51 and KTB39; Fig. EV2C). The ancestry of these cell lines was previously estimated using a panel of 41-ancestry-informative SNPs (Kumar et al, 2018). RNA-seq analysis revealed significant gene expression changes induced by *TRIM37* overexpression in KTB51 (Figs. 2F and EV2D) and KTB39 (Figs. 2H and EV2F) cells relative to control cells. As expected, GSEA of the DEGs showed a positive enrichment of CSC and EMT pathways in *TRIM37* overexpressing KTB51 (Figs. EV2E and 2G) and KTB39 cells (Figs. EV2G and 2I). Together, these results suggest that a higher level of *TRIM37* primes normal breast tissue for neoplastic and premalignant changes.

## A reporter-based screen identifies risk variant rs57141087 affecting TRIM37 promoter–enhancer interactions dominant in BW

We were surprised that *TRIM37* was expressed at a higher level in the breast tissue of healthy BW than WW. To further investigate this finding, we annotated the single-nucleotide polymorphisms (SNPs) associated with TRIM37 for genomic coordinates (59,108,921–58,982,138 Mbp) available through the NHGRI (http://www.ncbi.nlm.nih.gov/snp/). For quality control, SNPs that deviated from Hardy–Weinberg equilibrium were excluded from further analyses. Of the 370 TRIM37-associated SNPs, we identified 154 variants predominant in BW with minor allele frequency (MAF) greater than or equal to 0.0003 (Fig. EV3A, Dataset EV1). Previous studies have shown that SNP location in promoters dictates gene expression and cancer susceptibility (Deng et al, 2017; Fagny et al, 2020). We, therefore, focused on nine SNPs located in the ~2 Kbp upstream of the transcription start site (TSS), including rs150880035 (C > T), rs57141087 (A > G), rs1029035382 (C > T),

rs914034546 (G > A), rs1359804461 (A > C), rs904164828 (G > T), rs572032837 (T > G), rs568274755 (G > A), and rs1042165062 (G > C) (Fig. 3A). Figure 3B summarizes allelic frequencies in BW and WW for SNPs distributed in the TRIM37 gene upstream region. The linkage disequilibrium (LD) between SNPs located in Chromosome 17 (59,108,921–58,982,138 Mbp) indicated a block of strong LD ($r^2 \geq 0.8$) residing within ~2 Kbp of the 5′ promoter and intron 1 for (i) rs8082544 with rs6503893, rs6503894, rs11868348, rs8067264, rs7503190, rs8081967, rs8182277, rs12948746, rs8081297, rs7502208, rs8072712, rs6416932rs114844088, rs144783389, and rs148648311; (ii) rs57141087 with rs142125398, rs6503895, rs8071291, rs75148295, rs73993836, rs57755522, rs8065740 and rs7224809, (iii) rs150880035 with rs115329428 and rs140241165 (Fig. EV3B).

To systematically identify functional SNPs among the variants distributed in the 5′ gene upstream region of TRIM37, we carried out a small-scale candidate-based screen using a reporter-based promoter assay in HEK293T, MCF10a, and MDA MB 231 cells. Specifically, we cloned a 960-bp fragment (Chr17: 59,107,405-59,106,446) harboring either the reference or risk allele for each indicated SNP. As expected, we observed strong luciferase activity compared to the empty control (Fig. EV3C), confirming the regulatory activity of the 5′ gene upstream region of TRIM37. Among the SNP candidates, the risk allele for rs57141087 showed the maximum increase in promoter activity relative to the reference allele in the HEK293T, MCF10a, and MDA MB 231 cells (Fig. 3C), raising a possibility of genotype-dependent control of TRIM37 promoter activity.

We next interrogated the functional locus of rs57141087 to understand its association with TRIM37-driven breast cancer risk. The cis-expression quantitative trait loci (eQTL) analysis using 64 normal primary breast samples available through GTEx showed the strongest association between rs57141087 and *TRIM37* among all genes in the 126,784 bp window (Chr17:59,108,921–58,982,138; Fig. 3D). We found that the homozygous (G/G) or heterozygous (A/G) risk allele of rs57141087 strongly associated with *TRIM37* expression relative to the homozygous (A/A) reference allele at rs57141087 (Fig. 3E). Furthermore, our analysis of previously published GWAS studies of breast cancer, including 2860 cases and 16,262 controls of African ancestry (Jia et al, 2024), identified a significant association between rs57141087 and TNBC risk at $P = 0.02$, with odd ratios of 1.096 (Fig. EV3D).

We next sought to understand the underlying mechanism by which rs57141087 modulates *TRIM37* expression. rs57141087 is located 805 bp upstream of the TRIM37 TSS on chromosome 17 (Fig. 3F), which is highly enriched with enhancer marks, including histone H3 Lysine 4 mono-methylation (H3K4me1), histone H3 Lysine 27 acetylation (H3K27ac), deoxyribonuclease I (DNaseI) hypersensitive sites and histone H3 Lysine 4 trimethylation (H3K4me3) (Fig. 3G). We, therefore, hypothesized that the risk variant of rs57141087 could positively regulate *TRIM37* expression by modulating enhancer activity. A luciferase-based reporter assay confirmed an enhancer activity of ~350 bp fragment containing rs57141087 upstream of the TRIM37 minimal promoter (Fig. EV3E). Enhancer sequence with the risk allele at rs57141087 exhibited significantly stronger luciferase activity than that with the reference allele in HEK293T cells (Fig. 3H). The inclusion of an additional 556 bp upstream (Chr17: 59,107,406–59,107,961) did not accentuate the promoter activity relative to 353 bp fragment

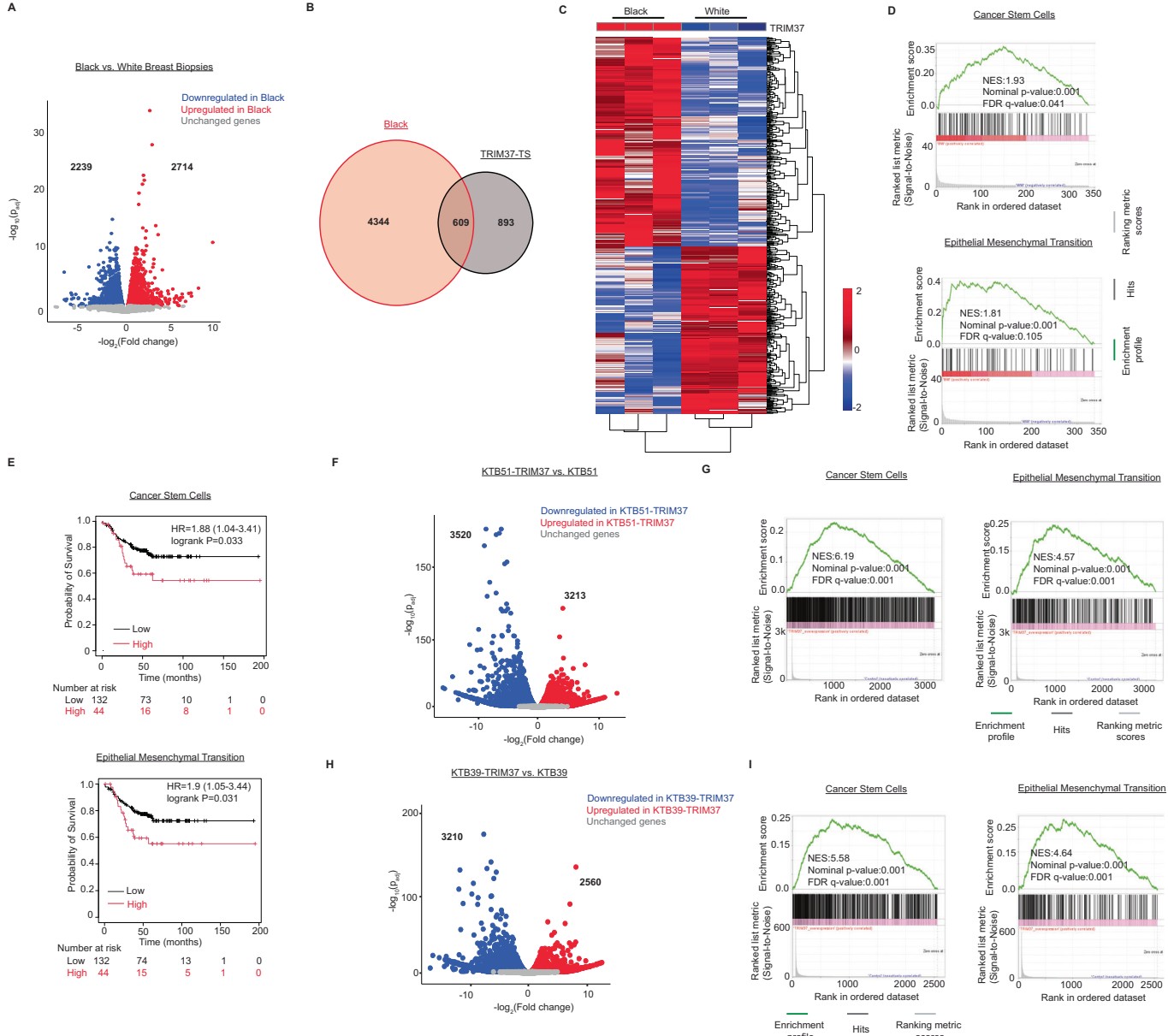

**Figure 2. African ancestry influences the *TRIM37*-associated transcriptional signatures in normal, cancer-free breast tissue.**

(A) Volcano plot illustrates differential gene expression in the cancer-free paraffin-archived breast tissue samples of parous and premenopausal and WW women ($n = 3$ biological replicates; ages <45). Differences in gene expression between BW and WW were tested using DESeq2 from Bioconductor. FDR < 0.1. *Red* are significantly upregulated genes ($n = 2714$), *blue* are significantly downregulated genes ($n = 2239$), and *grey* are genes not significantly changed ($n = 11,448$). (B) Venn diagram showing the overlap between differentially expressed genes in breast tissue from BW identified in (A) and *TRIM37*-regulated genes (GSE136617, Data ref: Przanowski et al, 2020). (C) Hierarchical clustering of median-centered *TRIM37*-TS in cancer-free BW and WW breast tissue ($n = 3$ biological replicates per group). Each colored line in the dendrogram identifies a different gene. (D) Enrichment plots for cancer stem cells (top) and EMT (bottom) gene signatures enriched in cancer-free BW ($n = 3$ biological replicates) breast tissue identified through GSEA analysis of RNA-seq data. Nominal *P* values were computed using the *t* test. (E) Kaplan–Meier analysis of survival in patients with low (*black*) or high (*red*) *TRIM37*-regulated CSC (top) and EMT (bottom) gene signatures enriched in cancer-free BW ($n = 3$ biological replicates) breast tissue identified in (B). For CSC signatures, *$P = 0.033$; for EMT signatures, *$P = 0.031$, log-rank test. The number of surviving patients at 0, 50, 100, and 150-month time points is indicated below the graph. (F) Volcano plot illustrates differential gene expression in control and *TRIM37* overexpressing KTB51 breast epithelial cells ($n = 2$ biological replicates per group). Differential gene expression between the groups was analyzed using DESeq2 from Bioconductor. FDR < 0.05. *Red* are significantly upregulated genes ($n = 3213$), *blue* are significantly downregulated genes ($n = 3520$), and *grey* are genes not significantly changed ($n = 7113$). (G) Enrichment plots for cancer stem cells (left) and EMT (right) gene signatures identified through GSEA analysis of RNA-seq data for *TRIM37* overexpressing KTB51 cells. Nominal *P* values were computed using a *t* test. (H) Volcano plot illustrates differential gene expression in control ($n = 2$ biological replicates) and *TRIM37* overexpressing ($n = 2$ biological replicates) KTB39 breast epithelial cells. Differential gene expression between the groups was analyzed using DESeq2 from Bioconductor. FDR < 0.05. *Red* are significantly upregulated genes ($n = 2560$), *blue* are significantly downregulated genes ($n = 3210$) and *grey* are genes not significantly changed ($n = 7769$). (I) Enrichment plots for cancer stem cells (left) and EMT (right) gene signatures identified through GSEA analysis of RNA-seq data for *TRIM37* overexpressing KTB39 cells. Nominal *P* values were computed using a *t* test.

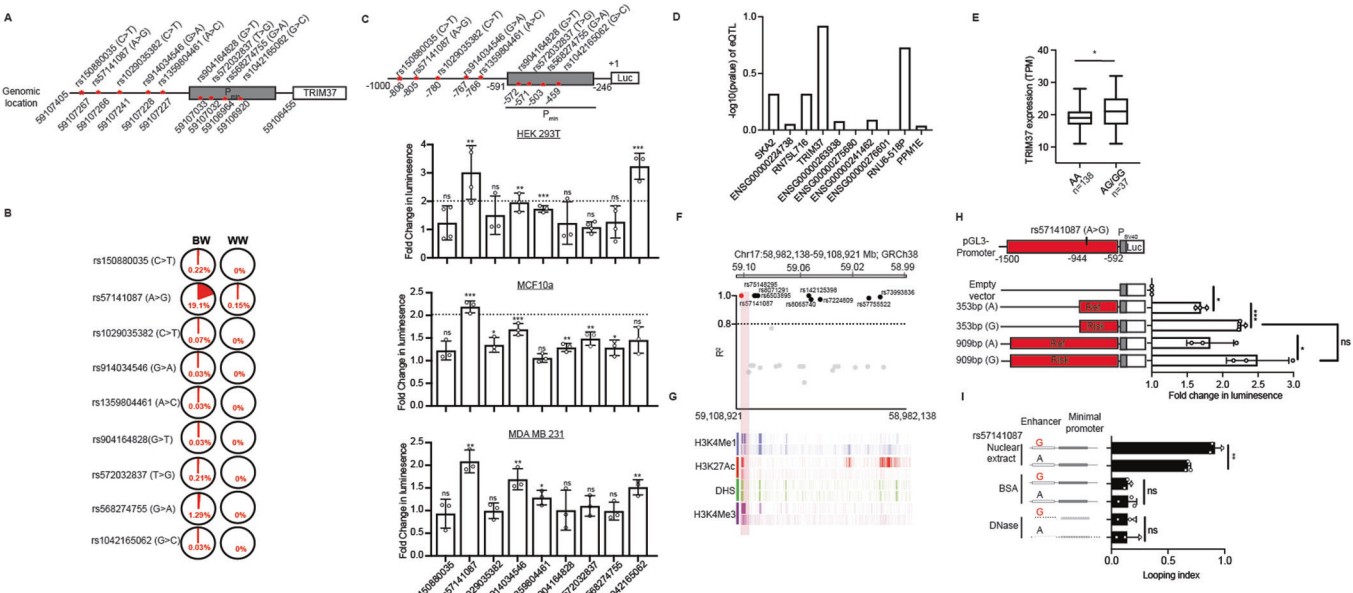

**Figure 3. rs57141087 modulates enhancer activity to regulate *TRIM37* expression.**

(A, B) A schematic view of the (Chr17: 59,107,405-59,106,446) region (A) and pie charts (B) showing the distribution of risk (red) and reference (black) alleles in BW and WW from the NCBI SNP database. The percentage of each haplotype in Black and White populations is indicated. (C) Luciferase reporter assays measure the promoter performance of the promoter harboring risk alleles for different SNPs (top) in HEK293T, MCF10a, and MDA MB 231 cells. The data are normalized to the reference allele and presented as a fold change in luminescence. For HEK293T; rs150880035 $^{ns}P = 0.463$, rs57141087, $**P = 0.005$, rs1029035382, $^{ns}P = 0.268$, rs914034546, $**P = 0.007$, rs1359804461, $***P = 0.0003$, rs904164828, $^{ns}P = 0.621$, rs572032837, $^{ns}P = 0.398$, rs568274755, $^{ns}P = 0.383$, rs1042165062, $***p = 0.001$; For MCF10a rs150880035, $^{ns}P = 0.133$, rs57141087, $***P = 9.4 \times 10^{-5}$, rs1029035382, $*P = 0.021$, rs914034546, $***P = 7.72 \times 10^{-3}$, rs1359804461, $^{ns}P = 0.330$, rs904164828, $**P = 0.005$, rs568274755, $*P = 0.045$, rs1042165062, $**P = 0.005$; For MDA MB 231; rs150880035, $^{ns}P = 0.731$, rs57141087, $**P = 0.002$, rs1029035382, $^{ns}P = 0.974$, rs914034546, $**P = 0.007$, rs1359804461, $*P = 0.036$, rs904164828, $^{ns}P = 0.974$, rs572032837, $^{ns}P = 0.469$, rs568274755, $^{ns}P = 0.928$, rs1042165062, $**P = 0.006$, unpaired *t* test. Data are mean ± SD of biological replicates, $n \geq 3$/group. (D) rs57141087 eQTL analysis of genes within 126,784-bp (Chr17:59,108,921–58,982,138) window using data for breast tissue from GTEx. (E) eQTL analysis demonstrating the correlation between rs57141087 genotype and *TRIM37* expression in samples from 1000 genome RNA-seq dataset ($n = 175$, Yoruba and UTAH population, Data ref: Lappalainen et al, 2013). $*P = 0.041$, *t* test. The boxed areas span the first to the third quartile, with the central line representing the median expression changes for each group. Outliers from the boxplots are not displayed. The whiskers represent the 15th and 85th percentiles. (F) The LocusZoom plot of LD for rs57141087 and its LD-SNPs in African (Yoruba, Luhya, Gambia, Mende, Esan, Americans of African Ancestry, and African Caribbean) population from the 1000 Genome project data. (G) The epigenetic heatmap for H3K4me1, H3K27Ac, DHSs, and H3K4me3 for the (Chr17:59,108,921–58,982,138) region harboring rs57141087. The data was derived from the Cistrome Data Browser. (H) Luciferase reporter assays measuring the enhancer activity of DNA fragment (Chr17: 59,107,961–59,107,053) harboring risk alleles for rs57141087 in HEK293T cells. The data are normalized to the vector only set to 1 and presented as a fold change in luminescence. For 353 bp, $***P = 4.8*10^{-4}$, 909 bp, $*P = 0.015$, 353 bp vs. 909 bp, $^{ns}P = 0.418$, unpaired *t* test. Data are mean±SD of biological replicates, $n = 3$/group. (I) Looping assay measures the promoter–enhancer interactions for in vitro synthesized DNA fragments (Chr17: 59,107,405-59,106,446) with risk or reference alleles. For Nuclear extract, $**P = 0.003$, BSA, $^{ns}P = 0.947$, $^{ns}$DNase, $P = 0.965$, unpaired *t* test. Data are mean±SD of biological replicates, $n = 3$/group. Source data are available online for this figure.

(Fig. 3H), indicating that the enhancer activity is limited to the 59,107,053–59,107,405 regions in the TRIM37 5' regulatory region.

Previous studies have identified looping between distant chromatin segments as an important feature of enhancer function during active transcription (Denker and de Laat, 2016; Jin et al, 2013; Ong and Corces, 2011; Spurrell et al, 2016; Zhang et al, 2013). We, therefore, measured the TRIM37 enhancer–promoter contact in the context of rs57141087 through a cell-free looping assay and quantitated by looping index (Panigrahi et al, 2018). As shown in Fig. 3I, we observed a significantly higher looping index for the construct harboring risk allele relative to the reference allele at rs57141087. Substituting a nuclear extract with BSA inhibited looping, indicating that promoter–enhancer looping depends on the protein factors (Fig. 3I). A prior treatment with DNase abolished the interactions (Fig. 3I). Together, these results indicate that the risk allele of the rs57141087 favors stronger TRIM37 promoter activity by facilitating promoter–enhancer interactions.

## The risk allele rs57141087 upregulates *TRIM37* via NRF1

To determine the effect of rs57141087 at its endogenous locus, we leveraged CRISPR/Cas9 to generate a clonal KTB51 cell line harboring risk allele G instead of A allele of rs57141087, hereafter referred to as 51-G. We obtained three heterozygous clones for the risk allele of rs57141087 that were confirmed by risk SNP-specific restriction digestion (Fig. EV4A) and next-generation sequencing of the target region (Fig. 4A). To confirm the functional integrity of the locus, the PolII recruitment to TRIM37 promoter was evaluated in 51-G and parental KTB51 cells (referred to as 51-A). The chromatin immunoprecipitation (ChIP) analysis revealed significantly higher PolII binding at the TRIM37 promoter in 51-G than in 51-A (Fig. 4B).

Given that SNPs in enhancer regions can function by modulating transcription factor (TF) binding, we evaluated the protein complexes binding capabilities of TRIM37 enhancer DNA

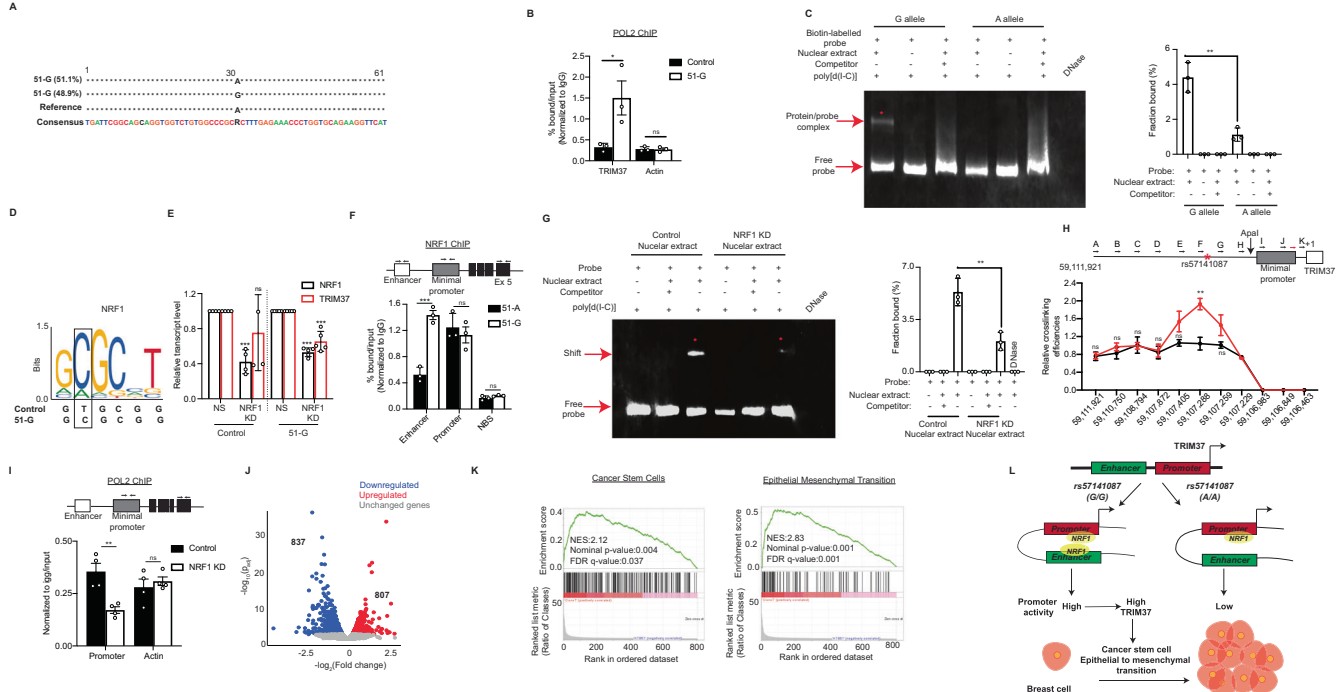

**Figure 4. rs57141087 modulates TRIM37 promoter–enhancer interactions through NRF1 binding.**

(A) A schematic summary for amplicon sequencing of A- and G allele for rs57141087 inserted through CRISPR/Cas9 technology in KTB51 breast epithelial cells. (B) ChIP assay monitoring PolII recruitment on TRIM37 promoter in KTB51 cells with A or G allele for rs57141087. Actin is used as an endogenous control. For TRIM37, *$P = 0.045$, and Actin, $^{ns}P = 0.889$, unpaired $t$ test. Data are mean ± SD of biological replicates, $n = 3$/group. (C) Immunoblot for biotin-labeled TRIM37 enhancer fragment with risk (G allele) and reference (A allele) incubated with HEK293T nuclear extract and competitor (unlabeled probe) as indicated. The bound and free DNA fragments are indicated with red arrows. The concentrations of nuclear extract (5 μg), Biotin-labeled probes (0.2 pmol), competitor (10 pmol), and poly [d(I-C)] (50 ng/μl) were indicated at the top. The representative image from one experiment is shown (left), and the shift in the band is indicated (red asterisk, *) and quantitated (right). **$P = 0.004$, unpaired $t$ test. Data are mean±SD of biological replicates, $n = 3$/group. (D) The risk G allele of rs57141087 harboring NRF1 binding motif in the enhancer region of TRIM37 (Chr17: 59,107,267–59,107,262) is shown. (E) qRT-PCR monitoring NRF1 and TRIM37 levels in KTB51 with A or G allele for rs57141087. Gapdh is used as an endogenous control. For reference allele NRF1, ***$P = 0.0001$, and TRIM37, $^{ns}P = 0.230$; For risk allele NRF1, ***$P = 2.68 \times 10^{-8}$, and TRIM37, ***$P = 0.0001$, unpaired $t$ test. Data are mean ± SD of biological replicates, $n \geq 4$/group. (F) ChIP assay monitoring NRF1 recruitment on TRIM37 locus in 51-G and 51-A cells. NRF1 non-binding site (NBS) is used as a negative control. The primers used for the qPCR analysis in the TRIM37 locus are indicated on Top. For Enhancer, ***$P = 0.0007$, Promoter, $^{ns}P = 0.544$, and NBS, $^{ns}P = 0.149$, unpaired $t$ test. Data are mean±SD of biological replicates, $n = 3$/group. (G) Immunoblot for biotin-labeled TRIM37 enhancer fragment incubated with nuclear extract from control or NRF1 knockdown HEK293T cells and unlabeled probe (competitor) as indicated. The bound and free DNA fragments are indicated with red arrows. The concentrations of nuclear extract (5 μg), Biotin-labeled probes (0.2 pmol), competitor (10 pmol), and poly [d(I-C)] (50 ng/μl) were indicated at the top. The representative image from one experiment is shown (left), and the shift in the band is indicated (red asterisk, *) and quantitated (right). **$P = 0.006$, unpaired $t$ test. Data are mean±SD of biological replicates, $n = 3$/group. (H) 3C-qPCR analysis across the ~5500-bp TRIM37 locus harboring risk and reference allele of SNP rs57141087 in 51-G and 51-A cells, respectively. The schematic shows the TRIM37 gene structure with PCR primers used for ChIP-qPCR assay (top). For (A), $^{ns}P = 0.780$, (B), $^{ns}P = 0.206$, (C), $^{ns}P = 0.918$, (D), $^{ns}P = 0.814$, (E), $^{ns}P = 0.069$; (F), **$P = 0.002$, (G), $^{ns}P = 0.104$, (H), $^{ns}P = 0.247$, and (I–K), not applicable, paired $t$ test. Data are mean ± SD of biological replicates, $n = 3$/group. (I) ChIP assay monitoring PolII recruitment on TRIM37 promoter in control and NRF1 knockdown KTB51 cells. Actin is used as an endogenous control. The primers used for the qPCR analysis in the TRIM37 locus are indicated on top. For Promoter, **$n = 0.005$, and Actin, $^{ns}n = 0.551$, unpaired $t$ test. Data are mean ± SD of biological replicates, $n = 4$/group. (J) Volcano plot illustrates differential gene expression in KTB51 with A or G allele for rs57141087 ($n = 2$ biological replicates). Differential gene expression between the groups was analyzed by DESeq2 Bioconductor. FDR < 0.05. Red are significantly upregulated genes ($n = 807$), blue are significantly downregulated genes ($n = 837$), and grey are genes not significantly changed ($n = 11,628$). (K) Enrichment plots for cancer stem cells (left) and EMT (right) gene signatures identified through GSEA analysis of RNA-seq data. Nominal $P$ values were computed using a $t$ test. (L) Graphical representation of the TRIM37 promoter regulation by the risk allele of rs57141087 and its effect on breast cancer progression. Source data are available online for this figure.

fragment in the context of rs57141087 risk and reference allele by electrophoretic mobility shift assay (EMSA). The result showed that the G allele has ~3.92-fold stronger binding capabilities than the A allele (Figs. 4C and EV4B). As expected, prior treatment of the probe with the DNase abolished the signal (Figs. 4C and EV4B). Prompted by these findings, we next scanned the rs57141087 locus for potential TF binding sites, which showed that the risk allele, but not the reference allele, of rs57141087, creates the unique binding sites for Growth factor independent-1 (GFI1), Paired Box 5 (PAX5) and Nuclear Respiratory Factor 1 (NRF1) in the enhancer region

(Figs. 4D and EV4C,D). GFI1 is a transcriptional repressor critical in myeloid and lymphoid differentiation and lymphocyte effector functions (Rathinam et al, 2005). PAX5 regulates early development and induces transcriptional signatures associated with neoplastic transformations (Jensen et al, 2007). NRF1 homodimerizes to regulate the expression of metabolic genes and nuclear genes required for DNA transcription and replication (Liu et al, 2024; Virbasius et al, 1993). Interestingly, the shRNA-mediated knockdown of PAX5 and GFI1 in 51-G and 51-A did not alter TRIM37 levels (Fig. EV3E,F), excluding their role in the

transcriptional regulation of *TRIM37* via the SNP rs57141087. In contrast, NRF1 knockdown in 51-G significantly decreased *TRIM37* levels (Fig. 4E). No significant change in *TRIM37* levels was observed in 51-A following NRF1 knockdown (Fig. 4E).

We next interrogated the physical interaction of NRF1 with the enhancer region harboring the risk allele of SNP rs57141087. The ChIP assay showed a significantly higher enrichment of NRF1 in the enhancer region for 51-G cells relative to 51-A cells (Fig. 4F). In concert, the protein binding affinities of DNA fragment harboring risk allele of rs57141087 showed a dramatic decrease when incubated with the nuclear extract from NRF1 knockdown cells relative to control cells (Figs. 4G and EV4G).

We argued that disrupting the NRF1 binding in the TRIM37 locus would interfere with promoter–enhancer interactions. To test this idea, we carried out an in vitro looping assay using a 960 bp fragment harboring either a risk or reference allele in the enhancer region and minimal promoter with wild-type or mutated NRF1 binding sites. As shown in Fig. EV4H, the loss of NRF1 binding in the TRIM37 locus significantly lowered the looping index for the construct harboring reference allele relative to the risk allele at rs57141087. As expected, substituting a nuclear extract with BSA or prior treatment with DNase abolished the interactions (Fig. EV4H). Encouraged by these results, we performed quantitative chromosome conformation capture (3C-qPCR) assays to examine rs57141087 interaction with the TRIM37 minimal promoter in vivo (Rebouissou et al, 2022). Using an anchor primer in the minimal promoter and probes in the region harboring rs57141087, we observed a strong chromatin interaction between the rs57141087 and the TRIM37 promoter in 51-G cells but not in 51-A cells (Fig. 4H). Significantly, the NRF1 knockdown is accompanied by a decreased PolII enrichment at the TRIM37 promoter relative to the control 51-G cells (Fig. 4I). Together, these results confirm a loop structure formation between the SNP rs57141087-containing enhancer region and the TRIM37 promoter, which regulates TRIM37 promoter activity.

To further explore the transcriptional effects of risk variant rs57141087, we carried out RNA-seq analysis in 51-A and 51-G. Two biological replicates were included in each group, and high correlations were observed between the replicates (Fig. EV4I,J; $r = 0.96$ for 51-A and $r = 0.95$ for 51-G). As expected, *TRIM37* was expressed at a significantly higher level in 51-G than in 51-A, as measured by qRT-PCR (Fig. EV4K). We identified 807 and 837 genes that were significantly upregulated and downregulated in 51-G, respectively (Fig. 4J). Gene Set Enrichment Analysis (GSEA) analysis identified CSC and EMT terms highly enriched in 51-G upregulated genes (Figs. EV4L and 4K) in concordance with *TRIM37* function in tumorigenesis and breast cancer progression. Together, our results indicate that risk variant rs57141087 brings enhancer in contact with TRIM37 promoter in *cis* through NRF1 binding to induce precise gene expression changes triggering neoplastic transformation and breast cancer progression (Fig. 4L).

### *TRIM37* drives oncogenic transformation in ancestry-mapped immortalized, healthy breast epithelial cells

Next, we assessed the tumorigenic capabilities of breast epithelial cells from BW in the context of *TRIM37*. To this end, we stably over-expressed *TRIM37* in KTB51 and a RAS expressing premalignant derivative of KTB51, referred to as K51t (Fig. 5A). Among the different KTB51 derivatives tested, the ectopic

expression of *TRIM37* potentiated the proliferation and colony formation in KTB51 and K51t cells (Fig. 5B,C).

Inspired by the results of the in-silico analysis (Fig. 2), we performed a series of phenotypic analyses to determine the tumorigenic and metastatic potential of *TRIM37*-transformed KTB51 cells. The wound-healing assay showed rapid wound closure within four hrs. for *TRIM37*-transformed KTB51 (K51-TRIM37) and KTB51t (K51t-TRIM37) cells relative to control cells (Figs. 5D and EV5A). A higher number of mammospheres were formed by KTB51 derivatives expressing *TRIM37*, with K51t-TRIM37 cells showing the greatest increase in self-renewal phenotype (Figs. 5E and EV5B). We next subjected mammospheres to flow cytometry for CD44/CD24 and EpCAM expression to evaluate the CSC and EMT phenotype, respectively. As expected, ~4.42- and ~1.68-fold increase in CD44$^{high}$/CD24$^{low}$ and EpCAM were observed for K51t-TRIM37 than in KTB51 cells, respectively (Figs. 5F,G and EV5C,D). In concert with FACS results, the qRT-PCR analysis of the mammospheres confirmed significantly increased expression of EMT gene signatures in KTB51 derivatives (Fig. 5H). In parallel, we generated *TRIM37*-expressing derivatives of white breast epithelial KTB37 cells (Fig. EV5E), which also showed increased cellular proliferation (Fig. EV5F,G). Consistently, we find that *TRIM37* overexpression is also sufficient to potentiate the CSC and EMT phenotype in KTB37 cells (Fig. EV5H–M), ruling out additional ancestry-specific genetic or epigenetic factors that could contribute to the tumorigenic potential of *TRIM37*.

We next directly investigated the oncogenic potential of *TRIM37*-transformed KTB51 derivatives through growth in soft agar. Notably, the ectopic expression of *TRIM37* was sufficient to transform KTB51 cells, which showed a 2.42-fold increase in colonies for KTB51-TRIM37 cells relative to KTB51 cells (Fig. 5I). The K51t-TRIM37 showed a maximum number of colonies in soft agar (Fig. 5I).

Finally, the ectopic expression of *TRIM37* enabled xenograft tumor formation in mice, with the K51t-TRIM37 tumors showing accelerated growth relative to control cells (Fig. 5J). The tumor growth was confirmed by hematoxylin and eosin (H&E) staining (Fig. 5K), the gross tumor tissue isolated postmortem (Fig. EV5N), and a high proliferative index as determined by Ki67 staining (Fig. 5L). The immunohistochemistry (IHC)-based staining for the Estrogen-α receptor (ER), progesterone-receptor (PR), and human epidermal growth factor receptor 2 (HER2) showed no significant staining in K51t-TRIM37 tumors relative to the positive control (Fig. 5M–O). Collectively, these results confirmed that high *TRIM37* levels in the early stages of transformation can inherently wire breast epithelial cells to accelerated TNBC growth and tumorigenesis.

### Context-dependent impact of oncogenic *TRIM37* expression on breast cancer onset and progression

Tumorigenesis is a multistage process that progresses through accumulating numerous somatic mutations, epigenetic aberrations, and transcriptional alterations. To experimentally address the differential effect of *TRIM37* on transformed phenotype in a cellular context, we established a reversible system with doxycycline-inducible *TRIM37* in constitutively KTB51t cells (K51t$^i$; Fig. EV6A). In K51t$^i$ cells, adding doxycycline leads to a reversible expression of *TRIM37* (Fig. 6A), and increased growth as measured through proliferation (Fig. EV6B), and colony formation assay (Fig. EV6C).

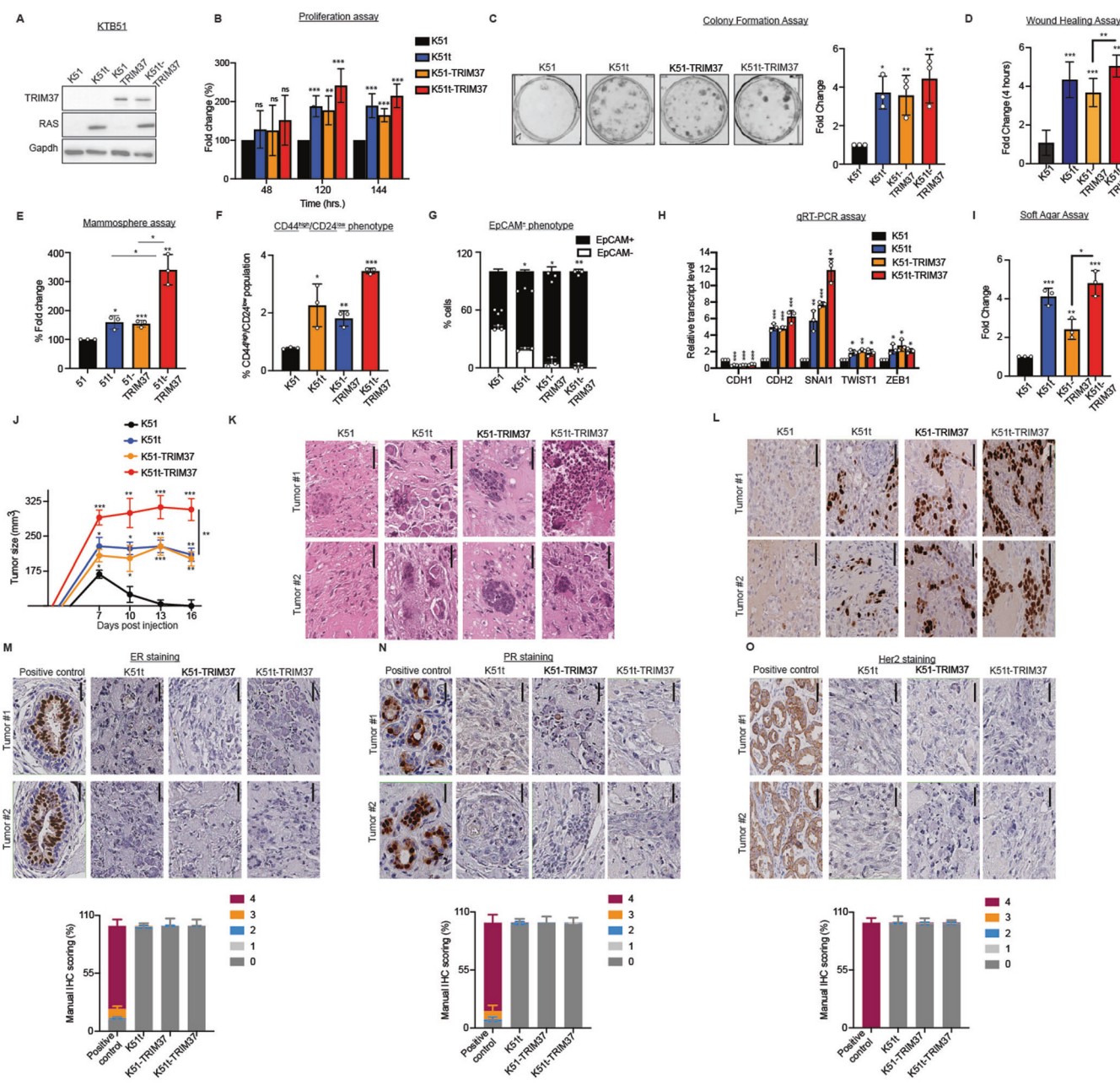

Next, we examined the impact of *TRIM37* depletion on the loss of identity and cellular reprogramming phenotype in K51t[i] cells. As expected, *TRIM37* expression in the presence of doxycycline significantly increased the number of mammospheres formed compared with the control KTB51 and uninduced K51t[i] cells (Figs. 6B and EV6D). Interestingly, doxycycline withdrawal stalled the reprogramming phenotype, and fewer or smaller mammospheres were observed (Figs. 6B and EV6D). In concert with the mammospheres assay, the FACS analysis confirmed a dramatic decrease in the CD44[high]/CD24[low] and EpCAM-positive cell population derived from mammospheres following doxycycline withdrawal (Figs. 6C,D and EV6E,F).

To explore this phenomenon in vivo, we assessed the growth of K51t[i] xenografts in NSG mice. Injected mice were fed on either a control or doxycycline-containing diet to allow for the continuous expression of *TRIM37*, and tumor growth was monitored. All the animals fed a doxycycline diet showed continued tumor growth, whereas 100% of the control animals failed to show growth of xenograft tumors (Fig. 6E). Interestingly, a subset of tumor-bearing animals, when deprived of a doxycycline diet, showed a marked reduction in the tumor growth, indicating the role of *TRIM37* in early stages of tumorigenesis (Fig. 6E). H&E and Ki67 staining revealed gross histology and the hyperproliferative cells of tumors (Fig. 6F,G). The K51t[i] xenografts from mice fed on a doxycycline diet demonstrated the absence of ER, PR, and Her2 expression relative to positive control (Fig. 6H–J). Together, these results indicate that *TRIM37* upregulation in the early stages of the disease gives the tumor a "head-start" to progress and metastasize by triggering cellular reprogramming.

**Figure 5. TRIM37 drives cancer stem cell and EMT phenotypes to enhance breast cancer initiation and growth.**

(A) Immunoblots in KTB51 and *TRIM37* derivatives of RAS-transformed KTB51 (K51t) cells. Gapdh was the loading control. (B) Relative cell growth for KTB51, K51t, K51-TRIM37, and K51t-TRIM37 cells at indicated times. For 48 h, K51t, $^{ns}P = 0.234$, K51-TRIM37, $^{ns}P = 0.413$, and K51t-TRIM37, $^{ns}P = 0.108$. For 120 h, K51t, $^{***}P = 8.96 \times 10^{-5}$, K51-TRIM37, $^{**}P = 0.002$, and K51t-TRIM37, $^{***}P = 8.64 \times 10^{-5}$. For 144 h., K51t, $^{***}P = 0.0002$, K51-TRIM37, $^{***}P = 2.61 \times 10^{-5}$, and K51t-TRIM37, $^{***}P = 3.10 \times 10^{-5}$, unpaired $t$ test. Data are mean ± SD of biological replicates, $n = 5$/group. (C) Representative bright-field images after crystal violet staining showing the growth of KTB51, K51t, K51-TRIM37, and K51t-TRIM37 cells. The colonies were quantified (right). For K51t, $^*P = 0.012$, K51-TRIM37, $^{**}P = 0.005$, and K51t-TRIM37, $P = ^{**}0.009$, unpaired $t$ test. Data are mean±SD of biological replicates, $n = 3$/group. (D) The relative migratory abilities for KTB51, K51t, K51-TRIM37, and K51t-TRIM37 cells were quantitated after 4 h. For K51t, $^{***}P = 3.3 \times 10^{-5}$, K51-TRIM37, $^{***}P = 6.44 \times 10^{-5}$, and K51t-TRIM37, $^{***}P = 5.26 \times 10^{-7}$; K51t vs. K51t-TRIM37, $^{ns}P = 0.138$, K51-TRIM37 vs. K51t-TRIM37, $^{**}P = 0.005$, unpaired $t$ test. Data are mean ± SD of biological replicates, $n = 6$/group. (E) Quantitation of solid mammospheres formed by KTB51, K51t, K51-TRIM37, and K51t-TRIM37 cells. For K51t, $^*P = 0.012$, K51-TRIM37, $^{***}P = 0.001$, and K51t-TRIM37, $^{**}P = 0.005$; K51t vs. K51t-TRIM37, $^*P = 0.024$, K51-TRIM37 vs. K51t-TRIM37, $^*P = 0.018$, unpaired $t$ test. Data are mean±SD of biological replicates, $n = 6$/group. (F, G) FACS analysis of CD24 and CD44 (F) and EpCAM (G) in KTB51, K51t, K51-TRIM37, and K51t-TRIM37 cells derived from mammospheres in (E). For CD44$^{high}$/CD24$^{low}$ in K51t, $^*P = 0.026$, K51-TRIM37, $^{**}P = 0.003$, and K51t-TRIM37, $^{***}P = 1.31 \times 10^{-6}$. For EpCAM$^+$ in K51t, $^*P = 0.011$, K51-TRIM37, $^*P = 0.011$, and K51t-TRIM37, $^{**}P = 0.003$, unpaired $t$ test. Data are mean±SD of biological replicates, $n = 3$/group. (H) qRT-PCR analysis of EMT markers in KTB51, K51t, K51-TRIM37, and K51t-TRIM37 cells derived from mammospheres in (E). Gapdh is used as an endogenous control. For CDH1 in K51t, $^{***}P = 8.08 \times 10^{-5}$, K51-TRIM37, $^{***}P = 1.70 \times 10^{-5}$, and K51t-TRIM37, $^{***}P = 0.0003$; CDH2; K51t, $^{***}P = 9.11 \times 10^{-5}$, K51-TRIM37, $^{***}P = 0.0004$, and K51t-TRIM37, $^{***}P = 0.0002$, For SNAI1 in K51t, $^{**}P = 0.003$, K51-TRIM37, $^{***}P = 2.8 \times 10^{-6}$, and K51t-TRIM37, $^{**}P = 0.005$. For TWIST1 in K51t, $^*P = 0.012$, K51-TRIM37, $^{**}P = 0.0014$, and K51t-TRIM37, $^*P = 0.022$. For ZEB1 in K51t, $^*P = 0.019$, K51-TRIM37, $^*P = 0.013$, and K51t-TRIM37, $^*P = 0.020$, unpaired $t$ test. Data are mean ± SD of biological replicates, $n = 3$/group. (I) Quantitation of colonies formed in soft agar by KTB51, K51t, K51-TRIM37, and K51t-TRIM37 cells. For K51t, $^{***}P = 0.0002$, K51-TRIM37, $^{**}P = 0.009$, and K51t-TRIM37, $^{***}P = 0.0005$; K51t vs. K51t-TRIM37, $^{ns}P = 0.201$, K51-TRIM37 vs. K51t-TRIM37, $^*P = 0.044$, unpaired $t$ test. Data are mean±SD of biological replicates, $n = 3$/group. (J) The xenograft tumor volume measurements of NSG mice injected with KTB51, K51t, K51-TRIM37, and K51t-TRIM37 cells. For 7 days, K51t, $^*P = 0.018$, K51-TRIM37, $^*P = 0.033$, and K51t-TRIM37, $^{***}P = 6.28 \times 10^{-5}$. For 10 days, K51t, $^*P = 0.011$, K51-TRIM37, $^*P = 0.039$, and K51t-TRIM37, $^{**}P = 0.007$. For 13 days, K51t, $^{***}P = 1.71 \times 10^{-5}$, K51-TRIM37, $^{***}P = 0.0002$, and K51t-TRIM37, $^{***}P = 1.42 \times 10^{-5}$. For 16 days, K51t, $^{**}P = 0.002$, K51-TRIM37, $^{**}P = 0.005$, and K51t-TRIM37, $^{***}P = 1.85 \times 10^{-5}$, K51t vs. K51t-TRIM37, $^{**}P = 0.005$, and K51-TRIM37 vs. K51t-TRIM37, $^{**}P = 0.004$, unpaired $t$ test. Data are mean ± SD of biological replicates, $n = 6$/group. (K, L) Representative images of H&E (K, ×20, scale bar, 200 μm) and Ki67 (L, ×20, scale bar, 200 μm) for the tumors isolated in (J) are shown. (M–O) Representative immunohistochemical (IHC) images of ER (M, ×20, scale bar, 200 μm), PR (N, ×20, scale bar, 200 μm), and Her2 (O, ×20, scale bar, 200 μm) for the tumors isolated in (J) are shown. The manual quantification of the IHC signal is shown for ER (M, bottom), PR (N, bottom), and Her2 (O, bottom) ($n = $ ~1200–1500 cells per group). Source data are available online for this figure.

# Discussion

The TNBC phenotype accounts for 30% of all breast cancer cases in BW (Siddharth and Sharma, 2018), whereas only 15% of WW are diagnosed with TNBC (Bowen et al, 2008). Likewise, the TNBC mortality rate is 40% higher in BW than in WW (Siegel et al, 2021). The efforts have been focused on understanding the molecular basis of racial disparity and characterizing the effects of African ancestry on breast cancer onset, progression, and clinical outcome. Significant genetic, epigenetic, and transcriptomic diversity has been identified through high-throughput genomics, proteomics, and cross-sectional imaging analysis of breast tumors from self-reported BW and WW (Huo et al, 2017; Lee et al, 2021; Marino et al, 2022; Martini et al, 2022a). For example, the mutation rate and intra-tumoral heterogeneity in breast tumors are significantly higher in BW compared to WW (Ansari-Pour et al, 2021; Parida et al, 2023). The global transcript studies have detailed over 400 DEGs between the two racial identities (Marino et al, 2022; Martini et al, 2022a). While these results are promising, there is a significant gap in our understanding of how these changes in gene expression are associated with cancer disparity and what regulates racial identity-specific gene expression. We note that the interpretation of the data is further complicated by the inherent limitations of small cohort size, with primary data from populations of European descent underscoring the need for better representations of diverse populations. Biases in analysis can also originate from the self-reported medical history data and heterogeneity of the racial identity. The lack of a diverse set of ancestry-specific cellular models and missing functional mechanisms linking genes or mutations to the unique TNBC biology in BW further present challenges in identifying and validating ancestry-specific genetic drivers that can be used as predictive biomarkers.

In this study, we used a candidate-based approach to interrogate the association of biological components to disproportionate TNBC incidence and mortality in BW. We performed the meta-analysis for gene-level associations to examine whether differences in the *TRIM37* expression would account for the more aggressive TNBC phenotype in BW. We show that *TRIM37* is expressed significantly higher in the cancer-free breast tissue from BW than in WW. We note that racial identity is a complex construct, and the use of self-reported racial identity is problematic for determining underlying ancestry-specific genetic determinants of breast cancer risk. We, therefore, estimated the global genomic ancestry for each individual in the TCGA cohort to evaluate the varying levels of admixture. Our analysis of the six samples from BW showed a high level of African ancestry (median 94.5%) with smaller contributions of European (median 5%) and Asian (median 0%) ancestry. Similarly, the WW samples showed a high level of European ancestry (median 99%). The samples from WW had a mixture of African (median 0.95%) and Asian ancestry (median 1.77%), with one sample showing greater than 30% Asian ancestry. Given that ancestral differences can impact the gene-disease associations, we believe the admixture refinement strongly supports *TRIM37* function as a genetic determinant of racial disparity in TNBC patients.

Multiple SNPs have been associated with breast cancer prevalence, as well as clinicopathologic status (Lilyquist et al, 2018; Middha et al, 2023; Stolarova et al, 2023). Besides *BRCA1* and *BRCA2* mutations that markedly increase breast cancer risk (Antoniou et al, 2010), hundreds of low- and moderate-risk susceptibility variants have been identified, including caspase-8 (rs2293554, rs6723097, (Park et al, 2016)), TIMP-2 (rs7501477 (Peterson et al, 2009)), and FSCN1 (rs56156320, rs3801004, (Wang et al, 2017)). In addition, SNPs play a critical role in phenotypic variation and disease susceptibility through effects on gene expression, such as PDE4DIP (rs12124527, (Kim et al, 2012)) and FTO (rs9939609, (Karra et al, 2013)). Interestingly, polymorphic variants in the tripartite motif-containing (TRIM) superfamily members have been linked to several human diseases, highlighting their clinical significance. For example, a recent study identified the

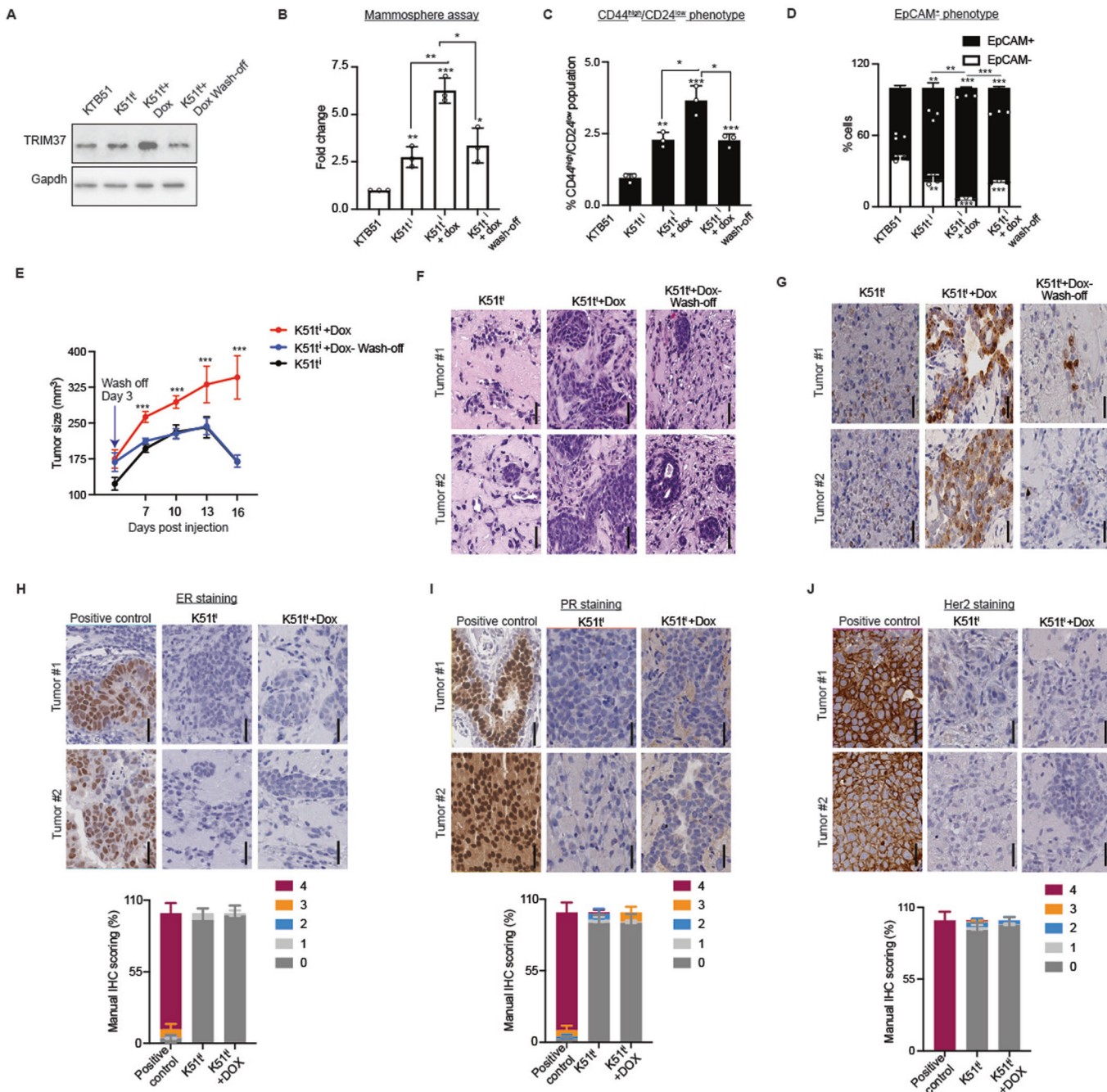

breast cancer risk locus SNP rs4971059 that regulates the TRIM46-HDAC1 axis to modulate genes involved in DNA replication and repair (Zhang et al, 2021). Notably, TRIM37 polymorphic variants have previously been associated with leukemia (rs11656413, (Hess et al, 2017)), schizophrenia ((rs4968363, rs2877926, (Hall et al, 2020)), and inflammatory bowel (Rivas et al, 2011).

Previous GWAS have identified multiple breast cancer suscept-ibility variants (Easton et al, 2007; Ferreira et al, 2019), but ancestry-specific variants associated with breast cancer progression and aggressiveness are rare. Here, we identified a risk variant of rs57141087 in the 17q23 region that associates with *TRIM37* expression through modulating promoter–enhancer interactions. We note that rs57141087 has not been previously identified as a causal variant for breast cancer, nor is it in strong LD with previously described risk SNPs. However, the discovery of all risk-associated variants cannot be based on statistical analyses (van de Bunt et al, 2015). Furthermore, the underrepresentation of African ancestry and predominance of data collected from populations of European ancestry biases analyses and interpretation of the results. Significantly, our analysis of GWAS data from samples of African ancestry (Fig. EV3D), complemented by comprehensive functional analysis, showed a significant association of rs57141087 with TNBC risk.

Our sequence analysis identified a unique site for NRF1 in the enhancer region of TRIM37 harboring risk variant rs57141087.

**Figure 6. Early-stage expression of *TRIM37* affects breast cancer initiation and progression.**

(A) Immunoblot analysis in K51t$^i$ cells following treatment with doxycycline (dox) 48 h. post-induction or removal after 3 days (Wash-off). Gapdh was the loading control. (B) Quantitation of solid mammospheres formed by K51t$^i$ in the presence of dox. For a subset of induced K51t$^i$ cells, dox was removed after 3 days of treatment (wash-off). For K51t$^i$, **$P = 0.005$, K51t$^i$+dox, ***$P = 0.0002$, K51t$^i$+wash-off, *$P = 0.011$, K51t$^i$ vs. K51t$^i$+dox, **$P = 0.002$, and K51t$^i$+Dox vs. K51t$^i$+wash-off, *$P = 0.012$, unpaired $t$ test. Data are mean±SD of biological replicates, $n = 3$/group. (C, D) FACS analysis of K51t$^i$ mammospheres derived in (B) for CD24 and CD44 (C) and EpCAM (D) in the presence of dox or wash-off. For CD44$^{high}$/CD24$^{low}$ in K51t$^i$, **$P = 0.001$, K51t$^i$+dox, ***$P = 0.0009$, K51t$^i$+wash-off, ***$P = 0.00098$, K51t$^i$ vs. K51t$^i$+dox, *$P = 0.014$, K51t$^i$+dox vs. K51t$^i$+wash-off, $P = 0.012$. For EpCAM$^+$ in K51t$^i$, **$P = 0.003$, K51t$^i$+dox, ***$P = 1.2 \times 10^{-5}$, K51t$^i$+wash-off, ***$P = 0.0001$, K51t$^i$ vs. K51t$^i$+dox, **$P = 0.004$, K51t$^i$+dox vs. K51t$^i$+wash-off, ***$P = 6.65 \times 10^{-5}$, unpaired $t$ test. Data are mean±SD of biological replicates, $n = 3$/group. (E) The xenograft tumor volume measurements of NSG mice injected with K51t$^i$ and fed on either a control or dox-enriched diet. A subset of animals was deprived of dox induction after 4 days (Wash-off). For 7 days, K51t$^i$ vs. K51t$^i$+dox, ***$P = 3.74 \times 10^{-7}$, and K51t$^i$+dox vs. K51t$^i$+wash-off, ***$P = 2.22 \times 10^{-6}$. For 10 days, K51t$^i$ vs. K51t$^i$+dox, ***$P = 1.28 \times 10^{-5}$, and K51t$^i$+dox vs. K51t$^i$+wash-off, ***$P = 2.16 \times 10^{-6}$. For 13 days, K51t$^i$ vs. K51t$^i$+dox, ***$P = 9.4 \times 10^{-5}$, and K51t$^i$+dox vs. K51t$^i$+wash-off, ***$P = 4.31 \times 10^{-5}$. For 16 days, K51t$^i$ vs. K51t$^i$+dox, ***$P = 1.22 \times 10^{-7}$, and K51t$^i$+dox vs. K51t$^i$+wash-off, ***$P = 3.74 \times 10^{-7}$, unpaired $t$ test. Data are mean ± SD of biological replicates, $n = 6$/group. (F, G) Representative images of H&E (F, ×20, scale bar, 200 µm) and Ki67 (G, ×20, scale bar, 200 µm) for the tumors isolated in (E) are shown. (H–J) Representative immunohistochemical (IHC) images of ER (H, ×20, scale bar, 200 µm), PR (I, ×20, scale bar, 200 µm), and Her2 (J, ×20, scale bar, 200 µm) for the tumors isolated in (E) are shown. The manual quantification of the IHC signal is shown for ER (H, bottom), PR (I, bottom), and Her2 (J, bottom) ($n = $ ~1200 cells per group. Source data are available online for this figure.

Mechanistically, NRF1 is a transcription factor that functions as a homodimer to adopt a U-turn conformation, facilitating the interaction between the two TGCGC motifs in the GCGCATGCGC consensus sequence (Liu et al, 2024). NRF1 activity is significantly associated with breast cancer and patient survival (Falco et al, 2016). Interestingly, the transcription factor target enrichment analysis in the TCGA cohort identified increased NRF1 activity in BW with TNBC subtypes (Ramos et al, 2020). We found that NRF1 bound to risk variant rs57141087 modulates TRIM37 promoter activity in an allele-specific manner. Here, we show that the additional NRF1 binding in the enhancer region facilitates promoter–enhancer interaction through looping to increase TRIM37 promoter activity.

Despite making progress in early diagnosis, optimized management, and breast cancer treatment, our understanding of cancer causation is limited. Detecting driver events in cancer initiation is pivotal for developing new predictive prognostic biomarkers. We find that TRIM37 could contribute to disease susceptibility through its role in the early phase of carcinogenesis. Here, we uncovered that higher *TRIM37* expression in the normal, cancer-free breast tissue of BW favors transcriptional activation sufficient to induce transformation in immortalized breast epithelial cells.

# Methods

## Reagents and tools table

| Reagent/resource | Reference or source | Identifier or catalog number |
|---|---|---|
| **Experimental models** | | |
| KTB39 (Homo sapiens) | Harikrishna Nakshatri, Indiana University | |
| KTB37 (Homo sapiens) | Harikrishna Nakshatri, Indiana University | |
| KTB51 (Homo sapiens) | Harikrishna Nakshatri, Indiana University | |
| MDA MB-231 (Homo sapiens) | Bhatnagar et al, 2014 | |
| MCF10A (Homo sapiens) | Bhatnagar et al, 2014 | |
| HEK293 (Homo sapiens) | Bhatnagar et al, 2014 | |
| NOD.Cg-Prkdcscid Il2rgtm1Wjl/SzJ (M. musculus) | Jackson Laboratory | Strain #:005557 |
| Cancer-free paraffin-archived breast tissue | Natascia Marino (The Susan G. Komen Tissue Bank; Indiana University Simon Comprehensive Cancer Center) | |
| **Recombinant DNA** | | |
| TRIM37 (NM_015294) Human Tagged Lenti ORF Clone | Origene | Cat# RC224302L4 |
| pBabe H-Ras | Addgene | Cat# 18744 |
| pTRE-2 puro | Lynn Chamberlain | |
| pLenti CMV rtTA3 Blast (w756-1) | Addgene | Cat# 26429 |
| pGL3-basic | Promega | Cat# E1751 |
| pGL3-promoter vector | Promega | Cat# E1761 |

| Reagent/resource | Reference or source | Identifier or catalog number |
|---|---|---|
| pBABE-puro | Addgene | Cat# 1764 |
| LentiCRISPRv2 | Addgene | Cat# 52961 |
| **Antibodies** | | |
| TRIM37 (D7U2L) | Cell Signaling | Cat# 96167 |
| Ras (27H5) | Cell Signaling | Cat# 3339 |
| Peroxidase AffiniPure™ Goat Anti-Rabbit IgG (H + L) | Jackson ImmunoResearch | Cat# 111-035-144 |
| Peroxidase AffiniPure™ Goat Anti-Mouse IgG (H + L) | Jackson ImmunoResearch | Cat# 115-035-166 |
| PE anti-human CD24 | Biolegend | Cat# 311105 |
| Pacific Blue™ anti-human CD44 | Biolegend | Cat# 338823 |
| Purified anti-human CD326 (EpCAM) | Biolegend | Cat# 324201 |
| PE Goat anti-mouse IgG | Biolegend | Cat# 405307 |
| Anti-RNA polymerase II | Abcam | Cat# ab24758 |
| **Oligonucleotides and other sequence-based reagents** | | |
| **SNPs mutagenesis primers** | | |
| rs150880035 | This study | 5'-GTTTCTCAAAaTGCGGGCCAC-3' 5'-CCTGGTGCAGAAGGTTCATG-3' |
| rs57141087 | This study | 5'-TTTCTCAAAGcGCGGGCCACA-3' 5'-CCCTGGTGCAGAAGGTTC-3' |
| rs1029035382 | This study | 5'-ACCTGCTGCCaAATCACACGATG-3' 5'-GGTCTGTGGCCCGCACTT-3' |
| rs914034546 | This study | 5'-TCACACGATGtTGGGTGAAAACG-3' 5'-TTCGGCAGCAGGTGGTCT-3' |
| rs1359804461 | This study | 5'-CACACGATGCgGGGTGAAAAC-3' 5'-ATTCGGCAGCAGGTGGTC-3' |
| rs904164828 | This study | 5'-GCCCCAAGCTaAGGCCGCCTG-3' 5'-TCTGCTACGCCACCCACG-3' |
| rs572032837 | This study | 5'-CCCCAAGCTCcGGCCGCCTGC-3' 5'-CTCTGCTACGCCACCCACG-3' |
| rs568274755 | This study | 5'-GTCTCGTATGtGCCCCGCAAC-3' 5'-TTGGCGACTCGCTGCCTC-3' |
| rs1042165062 | This study | 5'-CGCTCCTCCCgTCCTTTCCCC-3' 5'-TGGAGCGATCGCCTAGTTC-3' |
| **NRF1 mutagenesis** | | |
| NRF1 | This study | 5'-GTTCCATAGGTTATTTCCCGCGTCACGTGG-3' 5'-GACGGTGGAGTTCAGCG-3' |
| NRF1 | This study | 5'-GCGTCCTGCGATTTTTCGCTTCGCAGC-3' 5'-TGTGGGAGGGGAAAGGAG-3' |
| NRF1 | This study | 5'-GCTCAGGCCGTTATTTCAGGCTAGGGCCG-3' 5'-TTGGGGCTCTGCTACGCG-3' |
| **qRT-PCR** | | |
| GAPDH | This study | 5'-TGCACCACCAACTGCTTAGC-3' 5'-GGCATGGACTGTGGTCATGAG-3' |
| TRIM37 | This study | 5'-AACAGAGCGTGGAGAGCATT-3' 5'-CTTCTGCCCAACGACAATTT-3' |
| NRF1 | This study | 5'-TCAGCAAACGCAAACACAGG-3' 5'-AAGTGAGACAGTGCCATCAGG-3' |
| PAX5 | This study | 5'-ATGGATGCACACGGATGTTC-3' 5'-TCAACGGCTTGTGTCCTTTG-3' |

| Reagent/resource | Reference or source | Identifier or catalog number |
| --- | --- | --- |
| GFI1 | This study | 5'-AAAGCAAGAAGGCTCACAGC-3'<br>5'-GCATTTGAAGTGCTGTCTGC-3' |
| CDH1(E-cadherin) | This study | 5'-ACGCATTGCCACATACACTC-3'<br>5'-TCGGGCTTGTTGTCATTCTG-3' |
| CDH2(N-cadherin) | This study | 5'-TGTCCTTACTGTTGCTGCAG-3'<br>5'-TGTAACAGACACGGTTGCAG-3' |
| SNAIL | This study | 5'-TCCAGAGTTTACCTTCCAGCAG-3'<br>5'-ACAGAGTCCCAGATGAGCATTG-3' |
| TWIST | This study | 5'-TGGCCTGCAAAACCATAGTC-3'<br>5'-TGCATTTTACCATGGGTCCTC-3' |
| ZEB1 | This study | 5'-ATAGCAAGGAGTGGAGCATAGG-3'<br>5'-AAAGCCACATCAGCAACAGC-3' |
| Looping_enh | This study | 5'-AGAAATGACACCTCAGTTTG-3'<br>5'-GTGGTTCTCTCTGGAAGGAA-3' |
| Looping_prom | This study | 5'-CCCCTGACGTGGGCGCCGGG-3'<br>5'-GCCGCTGGCGACCCGCAGGC-3' |
| **ChIP** | | |
| TRIM37 | This study | 5'-AATGAGAAATGACACCTCAGTTTGG-3'<br>5'-GGTTTGTAGGACTTGCGTTCC-3' |
| TRIM37 | This study | 5'-ACTTAACTACGGGTGTGGCTC-3'<br>5'-CTGGCGTACTGGTGGAAAGC-3' |
| TRIM37 | This study | 5'-GCGCGCGCCCGCGTCACGTGGGC-3'<br>5'-TCCTTTCTCCCGGCTCAG-3' |
| TRIM37_ex5 | This study | 5'-AATTGTCGTTGGGCAGAAGA-3'<br>5'-ACAAAGGCAAAATCCATTCAATATTGA-3' |
| TRIM37_enh | This study | 5'-ACCATGAACCTTCTGCACCA-3'<br>5'-GAAAGCGCAGACTTCAGAGC-3' |
| TRIM37_MP | This study | 5'-CAGCGTCCTGCGCGCATGCGCTT-3'<br>5'-GCCCACGTGACGCGGGCGCGCGC-3' |
| TRIM37_MP | This study | 5'-CCTCCTTTCCCCTCCCACA-3'<br>5'-GAAGAAGGTGCCGCAGAGAA-3' |
| **3C assay** | | |
| A | This study | 5'-AAGAATGTTTATGTACATTTC-3' |
| B | This study | 5'-GTGGTACGCTCTCAGCTCACT-3' |
| C | This study | 5'-CATGCCCAGCTAGATATATAGTCAT-3' |
| D | This study | 5'-TAATTCTTTTAAGTAAAATT-3' |
| E | This study | 5'-AATGAGAAATGACACCTCAGTTTGG-3' |
| F | This study | 5'-TCTGCACCAGGGTTTCTCAAA-3' |
| G | This study | 5'-ACCTGCTGCCGAATCACACGAT-3' |
| H | This study | 5'-GCTGGGTGAAAACGCGGATGG-3' |
| E | This study | 5'-CGTAGCAGAGCCCCAAGCTCA-3' |
| J | This study | 5'-AGTCGCCAAGTCTCGTATGC-3' |
| K | This study | 5'-GCCCCGCCGAGAGCCGGAGGC-3' |
| Anchor | This study | 5'-CTCTGGTTACGGTTGGTGTTT-3' |
| **Cloning** | | |
| TRIM37 | This study | 5'-GGTACCAATGAGAAATGACACCTCAGTTTGG-3'<br>5'-GCTAGCCGCTCTGTTCATCCATTGCC-3' |
| TRIM37 | This study | 5'-CGGGGTACCTAATTCTTTTAAGTAAAATT-3'<br>5'-CCCGCTAGCCCACCCACGCCCACC-3' |
| TRIM37 | This study | 5'-GGTACCAATGAGAAATGACACCTCAGTTTGG-3'<br>5'-CCCGCTAGCCCACCCACGCCCACC-3' |

| Reagent/resource | Reference or source | Identifier or catalog number |
|---|---|---|
| EMSA | This study | 5′-GGTACCAATGAGAAATGACACCTCAGTTTGG-3′ 5′-ATTCGGCAGCAGGTGGTC-3′ |
| sgRNA | This study | 5′-CACCGGCATCGTGTGATTCGGCAGC-3′ 5′-AAACGCTGCCGAATCACACGATGCC-3′ |
| **shRNA** | | |
| NRF1_shRNA#1 | UMass Chan Medical School RNAi Core Facility | TRCN0000016903 |
| NRF1_shRNA#2 | UMass Chan Medical School RNAi Core Facility | TRCN0000016905 |
| GFI1_shRNA#1 | UMass Chan Medical School RNAi Core Facility | TRCN0000020464 |
| PAX5_shRNA#1 | UMass Chan Medical School RNAi Core Facility | TRCN0000016060 |
| PAX5_shRNA#2 | UMass Chan Medical School RNAi Core Facility | TRCN0000016062 |
| **Chemicals, enzymes, and other reagents** | | |
| KPNI | New England Biolabs | Cat# R3142 |
| NHEI | New England Biolabs | Cat# R3131 |
| BAMHI | New England Biolabs | Cat# R0136S |
| Apal | New England Biolabs | Cat# R0114S |
| Xcml | New England Biolabs | Cat# R0533S |
| Rodent Diet (2018, 625 Doxycycline) | ENVIGO | Cat# TD.01306 |
| T4 DNA ligase | Thermo Scientific | Cat# EL0011 |
| Q5 Site-Directed Mutagenesis Kit | New England Biolabs | Cat# E0554S |
| **Software** | | |
| R-studio | https://posit.co/download/rstudio-desktop/ | |
| Galaxy | https://usegalaxy.org/ | |
| Kaplan–Meier plotter | https://kmplot.com/analysis/ | |
| **Other** | | |
| Illumina Novaseq Platform | Novogene | |
| Quantstudio 3 | Applied Biosystems | |
| ChemiDoc™ Imaging System | Bio-Rad | |

## Cell lines and reagents

MDA MB 231, MCF10A, and HEK293T cells were cultured as previously described (Bhatnagar et al, 2014). KTB51, KTB37, and KTB39 cells (Harikrishna Nakshatri, Indiana University) were maintained in a mixture of DMEM/F-12 and DMEM (3:1) supplemented with 5% fetal bovine serum (FBS, Avantor), insulin (5 μg/ml, Gibco), EGF (20 ng/ml, Invitrogen), hydrocortisone (0.4 μg/ml, Invitrogen), adenine phosphate salt (24 μg/ml, Invitrogen), and Rock inhibitor (5 μg/ml, Invitrogen) (Kumar et al, 2018). All cells were kept under the same conditions at 37 °C and 5% $CO_2$. Cells cultured at the same time were pooled together and seeded after counting in a 6-well or 10-cm dish. Following this, cells were subjected, in a random order, to treatment with shRNA, sgRNA, plasmids, or the control. Cells were routinely tested for mycoplasma using the PlasmoTest kit (Invitrogen) (Przanowski et al, 2020).

## Animal care

NOD.Cg-Prkdcscid Il2rgtm1Wjl/SzJ (Jackson Laboratory) mice were housed in a specialized pathogen-free facility accredited by the American Association of Laboratory Animal Care (Przanowski et al, 2020). All animal studies and protocols (#22405 and #22406)

were approved by the guidance of the University of California, Davis Animal Care, and Use Committee (IACUC).

## TRIM37 and RAS overexpression

To overexpress TRIM37 and RAS, cells were seeded in a six-well plate to achieve 60–80% confluency and subsequently transduced with 200–500 μl lentiviral particles carrying the TRIM37 Lenti ORF clone, mGFP-tagged (Origene), pBabe Ras[mut] (Addgene), or both in 2 ml of suitable medium supplemented with ~6–10 μg/ml polybrene. For selection, the media was replaced with puromycin (1 μg/ml), and stable cells were maintained.

## Doxycycline-mediated activation of TRIM37

A TRIM37 cDNA clone (Origene) was subcloned into the vector pTRE-2 using BamH1 sites, and insertion was confirmed by Sanger sequencing. The TRIM37-expressing vector and pTET-ON (Addgene) were co-transfected into cells using Effectene reagent (Qiagen). After 24 h, the media containing the appropriate selection agents were replaced. Stable cells were induced with 500 ng/mL doxycycline (Dox) at various time points.

## RNA interference

To achieve stable shRNA knockdowns, cells were seeded as mentioned above and transduced with 200–500 µl of lentiviral particles expressing shRNAs (obtained from Open Biosystems/Thermo Scientific through the UMMS RNAi Core Facility, listed in the reagents and tools table) or sgRNA in a total volume of 2 ml of appropriate media supplemented with ~6–10 µg/ml polybrene. Media was replaced after overnight incubation, and cells were subjected to puromycin selection (1 µg/ml) for at least 3 days, as described previously (Przanowski et al, 2020).

## Immunoblotting

Cell lysates were prepared by lysing the cell pellet in RIPA buffer supplemented with one mM Dithiothreitol and 1× protease inhibitor. Immunoblots were probed using antibodies against Gapdh (Thermofisher), TRIM37 (Cell Signaling), and RAS (Cell Signaling).

## Quantitative RT-PCR (qRT-PCR)

Total RNA was extracted, and qRT-PCR was performed as previously described (Przanowski et al, 2018), with gene expression normalized to GAPDH. Primers used are listed in the reagents and tools table. Each sample was analyzed at least three independent times, and results from at least three different biological triplicates are presented.

## Proliferation assay

Briefly, cells were seeded at a density of $2 \times 10^4$ cells/ml in a 12-well plate. Cells were trypsinized and resuspended in 1 ml of complete media. Ten microliters of the resuspended cells were mixed with 10 µl of trypan blue (1:1). The number of viable cells was determined after 48, 96, 120, and 144 h using the Countess 3 Cell Counter (Invitrogen).

## Wound-healing assay

The cell migration was evaluated by seeding cells in a 48-well plate at a density of $2 \times 10^6$ cells/ml. After 24 h of incubation, culture inserts were removed using sterile pipetting tips, and the cell monolayer was gently washed with PBS to remove any cell debris and unattached cells. Fresh medium was then added to the wells. Plates were photographed immediately, and cell migration was monitored and recorded over a period of 4 h.

## Colony formation assay

Cells were seeded in six-well plates and cultured until visible colonies were formed. Colonies were fixed with 95% ethanol and 5% glacial acetic acid at 37 °C. After 30 min, colonies were washed twice with PBS and stained with 0.5% Crystal Violet dissolved in 2% ethanol and 98% PBS. Cells were imaged using a ChemiDoc Imaging System (Bio-Rad), and colonies were counted.

## Mammosphere assay

Five hundred cells were seeded in ultra-low-adherence 24-well plates (Corning) and cultured for 10 days. Images were captured with a ×10 microscope, and mammospheres were counted. For qRT-PCR and flow cytometry, mammospheres were dissociated with 1× trypsin and resuspended in the appropriate media.

## Flow cytometry

For CD24/CD44 staining, dissociated mammospheres were washed sequentially with PBS and FACS buffer and then resuspended in 100 µl of FACS buffer. Fluorescent-conjugated antibodies, including PE-conjugated anti-CD24 and Pacific, Blue-conjugated anti-CD44 antibodies (BD Biosciences, San Jose, CA, USA), were added. Cells were incubated for 45 min at 4 °C. Flow cytometry analysis was performed using a flow cytometer, and FlowJo software was used for data analysis.

For EpCAM staining, cells were incubated with a non-conjugated primary antibody against EpCAM (BD Biosciences, San Jose, CA, USA) for 45 min at 4 °C. After washing three times, the cells were incubated with a PE-conjugated secondary antibody for an additional 45 min at 4 °C. Flow cytometry analysis was performed as described above.

## NSG tumor xenograft with 17β-estradiol tablet

Age-matched female NSG mice were subcutaneously injected with one estradiol tablet (0.18 mg/pellet, Fisher Scientific) and 10 million of the indicated cell lines. For the doxycycline-inducible TRIM37 overexpression experiments, NSG mice were fed with a doxycycline diet (ENVIGO, 625 Doxycycline).

## Ki67, hematoxylin, and eosin staining

Tumors were excised and then fixed in 4% paraformaldehyde (PFA). Tumors were sectioned and stained with Ki67, H&E, and antibodies against ER, PR, and Her2. The in situ hybridization (ISH) Scoring Algorithm was used to score ER, PR, and Her2 (Thomsen et al, 2020).

## Vector construction

A 960-bp fragment (chr17: 59,106,446–59,107,405, hg38) of TRIM37 5'UTR was amplified from MDA MB 231 and cloned into the KpnI and NheI sites of pGL3-Basic (Promega) vector. In addition, 353-bp (chr17: 59,107,053–59,107,405, hg38) and 909-bp (chr17: 59,107,053–59,107,961, hg38) fragments of TRIM37 5'UTR were also amplified from MDA MB 231 and inserted into the KpnI and NheI sites of the pGL3-promoter (Promega) vector. Risk alleles of individual SNPs were introduced using the Q5 site-directed mutagenesis kit (New England Biolabs), with all constructs confirmed for correct variant incorporation by Sanger sequencing.

## Transfection and luciferase assay

Cells were plated 1 day before transfection for luciferase assay. Cells were transiently transfected with pBABE-puro (Addgene) combined with pGL3-basic (Promega) or pGL3-basic containing the 960-bp of TRIM37 5'UTR, harboring the reference or risk allele for each indicated SNP. For enhancer activity assay, cells were transfected with pBABE-puro (Addgene) along with pGL3 promoter (Promega) or pGL3-promoter vector containing a 353-

bp or 909-bp of TRIM37 5'UTR harboring the reference or risk allele for rs57141087 SNP. The transfection was performed using Effectene reagent (Qiagen) according to the manufacturer's protocols. Media was replaced after 48 h of incubation, and cells were subjected to puromycin selection (1 μg/ml) until control cells were dead. Cells were then harvested, and luciferase assay (Luciferase Assay System, Promega) was performed according to the manufacturer's instructions.

## Single-stranded oligodeoxynucleotides (ssODNs) mediated knock-in using CRISPR/Cas9

sgRNAs targeting sequences adjacent to the rs57141087 region were designed using TargetFinder and cloned into the LentiCRISPRv2 vector. KTB51 cells were transduced with lentiCRISPRv2-sgRNA virus for 24 h, followed by transfection with ssODN (Integrated DNA Technologies). Subsequently, all clones were validated using Next Generation Sequencing Amplicon-EZ and risk SNP-specific restriction digestion with the HhaI enzyme.

## Directed-ChIP assay

ChIP assays were performed as described previously (Przanowski et al, 2020). Antibodies against POLII (Abcam), NRF1 (Cell Signaling), and IgG (as a negative control) were used. Primer sequences for amplifying ChIP products are listed in the reagents and tools table. Each ChIP experiment was carried out at least three times independently.

## Nuclear extract preparation

Nuclear extract was prepared from either normal HEK293T cells or HEK293T NRF1 knockdown cells. The cell pellet was swollen in 200 μl of ice-cold NAR A buffer for 10 min. Centrifugation was performed to collect the nuclear pellet at 2000 rpm for 2 min at room temperature. Subsequently, the nuclear pellet was lysed in an ice-cold NAR C buffer with vigorous shaking for 25 min. Protein concentration was determined using Pierce's BCA protein assay kit (Thermo Fisher Scientific). The nuclear extracts were stored at −80 °C until further use.

## Looping assay

The looping assay was performed, and qRT-PCR results were analyzed as described previously (Panigrahi et al, 2018). The beads were digested with XcmI (New England Biolabs) at 37 °C for 5 h.

## Chromatin conformation capture (3 C) with qRT-PCR analysis

3 C analysis was performed according to previously established methods (Rebouissou et al, 2022). Following preparation of cell nuclei and formaldehyde cross-linking, samples were digested with ApaI (New England Biolabs) overnight at 37 °C. Ligation products were quantified using qRT-PCR assay. To determine primer efficiencies, a serial dilution of BAC clone (BACPAC Resources Center at BACPAC Genomics, Inc.) was digested with ApaI and ligated with T4 DNA Ligase. Primer sequences are listed in the reagents and tools table.

## Electrophoretic mobility shift assay (EMSA)

Consensus NRF1 DNA binding sites in the TRIM37 gene locus were identified using the UCSC Genome Browser and the Predicting Regulatory Functional Effect by Approximate P-value Estimation (PERFECTOS-APE) software. NRF1 binding site was prepared by constructing complementary oligonucleotides with the candidate binding site—GCGCGG—in the middle. A mutant version of this site was prepared by exchanging the conserved C and T in each NRF1 consensus binding site, and the primers used are listed in the reagents and tools table. EMSA was performed as described previously (Hua et al, 2018). Biotin-labeled probes were detected using the LightShift Chemiluminescent EMSA kit (Thermo Scientific) according to the manufacturer's instructions.

## Data analysis

### Data source
Transcriptomic data for TNBC were downloaded from The Cancer Genome Atlas (TCGA), Yale University (PRJNA704957), and the University of Alabama (GSE142731). Transcriptomic data from women at high risk of developing breast cancer were retrieved from GEO under the accession number GSE164641. RNA-seq data for normal breast samples was downloaded from TCGA, and genomic data were used for admixture analysis. Transcriptomic data for cancer-free breast tissue were downloaded from TCGA, GTEx, and GEO accession numbers (GSE164641, GSE111601, and GSE277029). Survival analysis data for TNBC patients were extracted from TCGA, GSEA39004, and GSE18229. Genomic and transcriptomic datasets for the Yoruba and UTAH populations (Accession: E-GEUV-1) from the 1000 Genomes were used for the eQTL analysis.

### SEER data analysis
TNBC incidence and mortality rate statistics were obtained from the Surveillance, Epidemiology, and End Results database (Surveillance, Epidemiology, and End Results Program, cancer.gov). GraphPad Prism was used to create graphical representations of trends in TNBC incidence, survival, and mortality rates from 2010 to 2020, stratified by age and racial identity. *P* value significance was calculated using the proportion test.

### TRIM37 expression analysis
Multiple TNBC mRNA datasets were downloaded from TCGA, GSE142731, and PRJNA704957. GSE164641 provided mRNA data for normal breast samples from women with an average risk or high risk of developing breast cancer. In addition, mRNA datasets from normal breast tissue were downloaded from GTEx (phs000424.v9.p2), TCGA, GSE164641, GSE111601, and GSE277029. These datasets were merged into a single meta-dataset using a cross-platform meta-analysis approach, as previously described (Tihagam and Bhatnagar, 2023). Boxplots were used to show the *TRIM37* expression difference between Black and White or average-risk and high-risk women. Statistical analysis was performed using the *t* test. All analyses were carried out using R-studio packages.

### Multivariate/univariate analyses
To investigate the relationship between *TRIM37* and clinicopathological factors (racial identity, age, BMI, cancer risk, menopausal state,

parity, and tumor stages), we employed a multivariate or univariate linear regression approach using the R regression function lm.

### Chi-square statistics

The chi-square test was used to evaluate the association between *TRIM37* expression and clinicopathological variables. The test was run using GraphPad Prism.

### Survival analysis

The association between high- or low *TRIM37* expression and overall survival in Black and White TNBC patients was examined using survival data from TCGA, GSE39004, and GSE18229. The receiver operating characteristic (ROC) plot was used to determine the optimal cutoff value for dividing TNBC patients into "high" or "low" TRIM37-expressing groups. Cox proportional hazards regression was used to perform survival analysis, and the results were visualized using Kaplan–Meier survival plots.

### Admixture analysis

Ancestry estimation was determined using the Admixture v1.3.0 software (Alexander et al, 2009). Admixture was run using the imputed TCGA breast SNP variants and RNA-seq data from three estimated populations (Black, White, and Asian). The correspondence between the columns and the population was identified using the originally reported racial identity information from three estimated populations (Black, White, and Asian). The analysis results in a proportional breakdown of each sample into three super-populations (African, White, and Asian).

### Differential expression gene

RNA was isolated from cancer-free paraffin-archived breast tissue samples of parous and premenopausal Black ($n = 3$) and White ($n = 3$) (provided by Natascia Marino) with the AllPrep DNA/RNA FFPE Kit (Qiagen). RNA-seq was performed by Novogene Genome Sequencing Company using the Illumina Novoseq platform with a paired-end 150 bp sequencing strategy. RNA-Seq data was aligned to the human genome assembly GRCm38/hg38 using the HISAT2 software package. Gene quantification and count data were generated using Featurecounts. DEG analysis between the three Black and three White samples was conducted using R-packages Deseq2 and edgeR. Differentially expressed genes were overlaid with TRIM37-TS genes (Przanowski et al, 2020). Gene set enrichment analysis to identify pathways enriched in TRIM37-TS upregulated in Black women was performed through the GSEA 4.3.2 software. Additionally, the association between EMT and CSC hallmark genes and recurrence-free survival (RFS) for patients with TNBC was examined using the Kaplan–Meier plotter (Gyorffy, 2024). EMT and CSC signature genes were divided into high and low based on the upper quartile.

### Functional annotation of SNPs

Genotype and RNA-seq data from the Genotype-Tissue Expression (GTEx, phs000424.v9.p2) project were used to calculate the normalized effect size (NES) for the association between TRIM37 expression and the expression quantitative trait loci (eQTL) status of the SNPs. Linkage disequilibrium (LD) was calculated between all SNPs (Chr17:59,108,921–58,982,138 Mb) in the African population based on the 1000 Genomes Project data using the LDmatrix Tool (Machiela and Chanock, 2015). LD-SNPs were defined based on a $r^2$ value greater than 0.8. In addition, mapped read files of epigenetic enhancer marks,

notably H3K4me1, H3K27ac, DHS, and H3K4me3 of breast cell lines, were accessed through the Cistrome database. All peaks were identified via WashU Epigenome Browser hg38, and rs57141087 location overlapped with epigenetic enhancer marks.

### Transcriptional factors binding site prediction

The TF binding sites for the region harboring rs57141087 were predicted using the UCSC Genome Browser and the Predicting Regulatory Functional Effect by Approximate $P$ value Estimation (PERFECTOS-APE) software, in conjunction with the transcription factor binding sites (TFBS) motif collection from JASPAR Parameters for prediction included a $P$ value < 0.05 and a fold change >2.

### RNA-seq analysis

RNA was isolated from control and TRIM37 overexpressing cells; Control and KTB51-G clone with RNeasy Mini Kit (Qiagen). RNA was also isolated from Formalin-Fixed Paraffin-Embedded normal breast tissue (provided by Natascia Marino) with AllPrep DNA/RNA FFPE Kit (Qiagen). RNA-seq was performed by Novogene Genome Sequencing Company using the Illumina Novoseq platform with a paired-end 150 bp sequencing strategy. RNA-Seq data was aligned to the human genome assembly GRCm38/hg38 using the HISAT2 software package. Using the DESeq2 R package, the differential expression of genes in the control and experimental group was determined. GSEA 4.3.2 software was used for gene set enrichment analysis for all genes. The volcano plot and heatmap were drawn using R.

### Imputation analysis

Ungenotyped variants were imputed using Beagle v5.2 for TCGA SNP array data. Beagle is based on identity by descent (IBD) and uses a hidden Markov model to account for uncertainty in inferring IBD (Browning et al, 2018). The imputation used the 1000 Genomes biallelic SNV called GRCh38, released by EMBL-EBI in December 2018, as the reference panel (Lowy-Gallego et al, 2019). HapMap genetic maps in PLINK format for GRCh38 were input to Beagle to estimate the recombination rate.

### Breast cancer risk analysis

TNBC risk association with the risk allele of rs57141087 was calculated using Genome-wide association studies (GWAS) data for African ancestry (Jia et al, 2024). Summary-level statistics data for TNBC (GCST90296722) were downloaded from the GWAS Catalog. The odds ratio (OR) was calculated from the beta coefficient using the following equation: beta = ln(OR).

### Statistical analysis

All experiments were performed at least in triplicate, and the results presented are the mean of at least three different biological replicates. The comparisons between the two groups were made by unpaired $t$ test; comparisons between multiple treatment groups were made by one-way or two-way ANOVA with indicated multiple comparisons post hoc tests. The overall survival of high- or low TRIM37 was calculated using the log-rank test. The difference in incidence and mortality rates between Black and White TNBC patients was assessed using the two-proportions z-test.

## Data availability

All transcriptomic data generated is available through the Gene Expression Omnibus (GEO) under the accession number GSE277029. The image source data can be accessed through BioImage Archive submission can be accessed with accession number S-BIAD1356: https://www.ebi.ac.uk/biostudies/bioimages/studies/S-BIAD1356?key=793d7905-9d4a-4556-869b-743bec729267.

The source data of this paper are collected in the following database record: biostudies:S-SCDT-10_1038-S44319-024-00331-2.

## Peer review information

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

## Acknowledgements

The authors thank Michael R. Green (University of Massachusetts School of Medicine), Harikrishna Nakshatri (Indiana University School of Medicine), and Natascia Marino (Indiana University School of Medicine) for providing reagents and cells. Ani W. Manichaikul (University of Virginia School of Medicine) and Pankaj Kumar (University of Virginia Bioinformatic Core) for their scientific support. Samples from the Susan G. Komen Tissue Bank at the IU Simon Comprehensive Cancer Center were used in this study. We thank contributors, including Indiana University, who collected samples used in this study, as well as donors and their families whose help and participation made this work possible. We also thank the University of California Davis, Comprehensive Cancer Center, Center for Genomic Pathology Lab, Flow Cytometry Core (NCI P30CA093373), and The Office of Animal Welfare at UC Davis. The authors thank the University of Virginia Comprehensive Cancer Center, the Biomedical Sciences Graduate Program (BIMS), and the Summer Research Internship Program (SRIP) programs. The Genotype-Tissue Expression (GTEx) Project was supported by the Common Fund of the Office of the Director of the National Institutes of Health and by NCI, NHGRI, NHLBI, NIDA, NIMH, and NINDS. The data used for the analyses described in this manuscript were obtained from dbGaP accession number phs000424.v9.p2 on 03/28/2022. The results shown in this manuscript are in part based upon data generated by the TCGA Research Network: https://www.cancer.gov/tcga. We acknowledge Jayanti Neupane and Klaudia Braczyk for their technical assistance. LF is the Placer Breast Cancer Endowed Chair. SB was supported by the Metavivor Foundation and the National Cancer Institute (NCI R01CA248930). RDT was supported by the NCI Diversity supplement (NCI R01CA248930-S1) and the UVA Cancer Training Grant.

## Author contributions

**Rachisan Djiake Tihagam**: Data curation; Software; Formal analysis; Validation; Investigation; Methodology; Writing—original draft. **Song Lou**: Validation; Investigation; Visualization. **Yuanji Zhao**: Investigation. **Kammi Song-Yan Liu**: Investigation. **Arjun Tushir Singh**: Software; Investigation; Visualization. **Bon Il Koo**: Investigation. **Piotr Przanowski**: Investigation. **Jie Li**: Supervision; Validation. **Xiaosong Huang**: Investigation. **Hong Li**: Formal analysis. **Jogender Tushir-Singh**: Formal analysis. **Laura Fejerman**: Investigation; Writing—review and editing. **Sanchita Bhatnagar**: Conceptualization; Resources; Data curation; Supervision; Funding acquisition; Investigation; Methodology; Writing—original draft; Project administration; Writing—review and editing.

Source data underlying figure panels in this paper may have individual authorship assigned. Where available, figure panel/source data authorship is listed in the following database record: biostudies:S-SCDT-10_1038-S44319-024-00331-2.

## Disclosure and competing interests statement

The authors declare no competing interests.

# Expanded View Figures

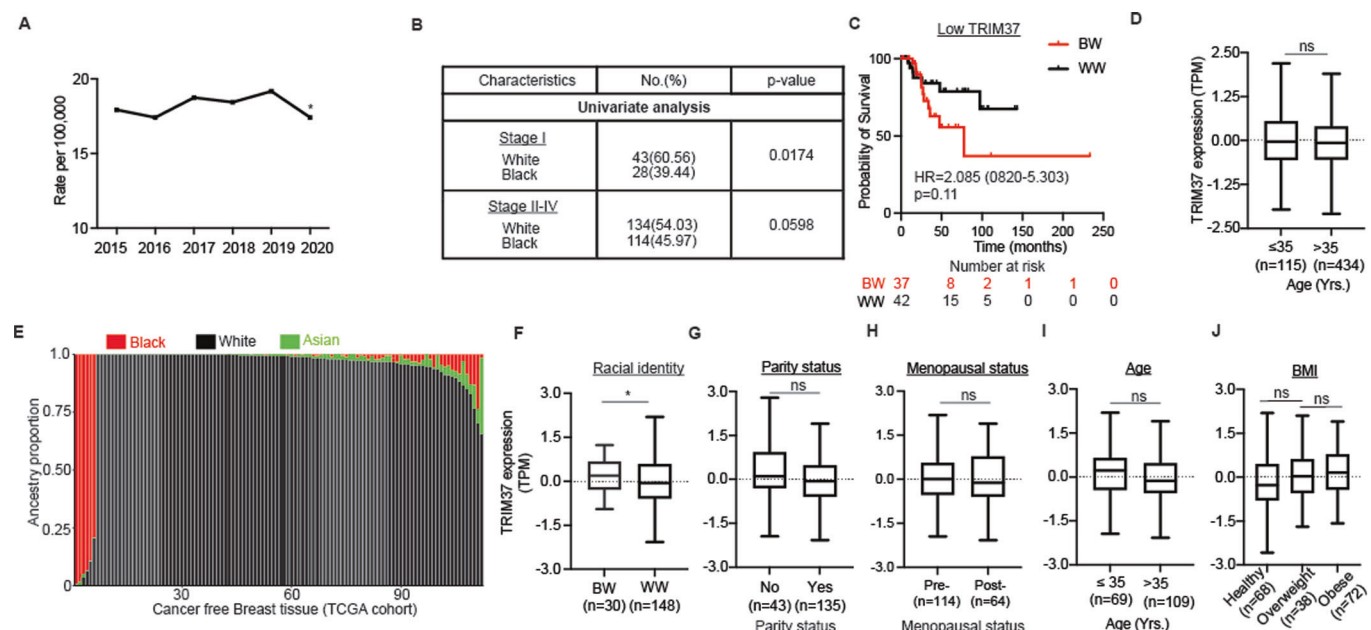

**Figure EV1. Ancestry-specific differences in TNBC survival and incidence.**

(A) Breast cancer incidence rate per 100,000 in women, excluding Hispanics, from 2015-2020 Surveillance, Epidemiology, and End Results (SEER) data. *$P = 0.038$, proportional test. The number of samples is indicated. (B) Univariate analyses of relationships between *TRIM37* and racial identity in Stage I ($n = 71$) and Stage II–IV ($n = 248$) TNBC patients (GSE142731, Data ref: Saleh et al, 2021; PRJNA704957, Data ref: NCBI Sequence Read Archive PRJNA704957, 2021, and TCGA, Data ref: The Cancer Genome Atlas Program (TCGA-Breast)) (C) Kaplan–Meier survival curve showing overall survival for the BW and WW TNBC patients with low *TRIM37* expression (GSE39004, Data ref: Tang et al, 2018; TCGA, Data ref: The Cancer Genome Atlas Program (TCGA-Breast), and GSE18229, Data ref: Prat et al, 2010). The number of surviving patients at 0, 50, 100, 150, 200, and 250-month time points is indicated below the graph. $^{ns}P = 0.11$, log-rank test. The number of surviving patients at 0, 50, 100, 150, 200, and 250-month time points are indicated. (D) Box plot for *TRIM37* expression in the normal breast tissue by age, below or equal to 35 ($n = 115$) vs. greater than 35 years ($n = 434$) (GSE164641, Data ref: Marino et al, 2022"; GSE111601, Data ref: Sun et al, 2018; GTEX, Data ref: The Genotype-Tissue Expression (GTEx), and TCGA, Data ref: The Cancer Genome Atlas Program (TCGA-Breast)). $^{ns}P = 0.95$, t test. The boxed areas span the first to the third quartile, with the central line representing the median expression changes for each group. Outliers from the boxplots are not displayed. The whiskers represent the 15th and 85th percentiles. (E) Estimated genetic ancestry distribution for self-reported BW ($n = 6$) and WW ($n = 103$) in the TCGA cohort (Data ref: The Cancer Genome Atlas Program (TCGA-Breast)). Each column represents an individual in the cohort, and the estimated proportion of African, European, and Asian ancestry is shown on the y-axis. (F–J) Box plot for *TRIM37* expression in the breast tissue from women stratified by racial identity (Black vs. White), parity (No vs. yes), menopausal status (Pre-vs. Post), age (below or equal to 35 vs. greater than 35 years) and BMI (Healthy vs. overweight vs. obese) (GSE164641, Data ref: Marino et al, 2022). For race, *$P = 0.05$, parity, $^{ns}P = 0.087$, menopausal status, $^{ns}P = 0.52$, age, $^{ns}P = 0.3$, and BMI, $^{ns}P = 0.17$, t test. The number of samples is indicated. The boxed areas span the first to the third quartile, with the central line representing the median expression changes for each group. Outliers from the boxplots are not displayed. The whiskers represent the 15th and 85th percentiles.

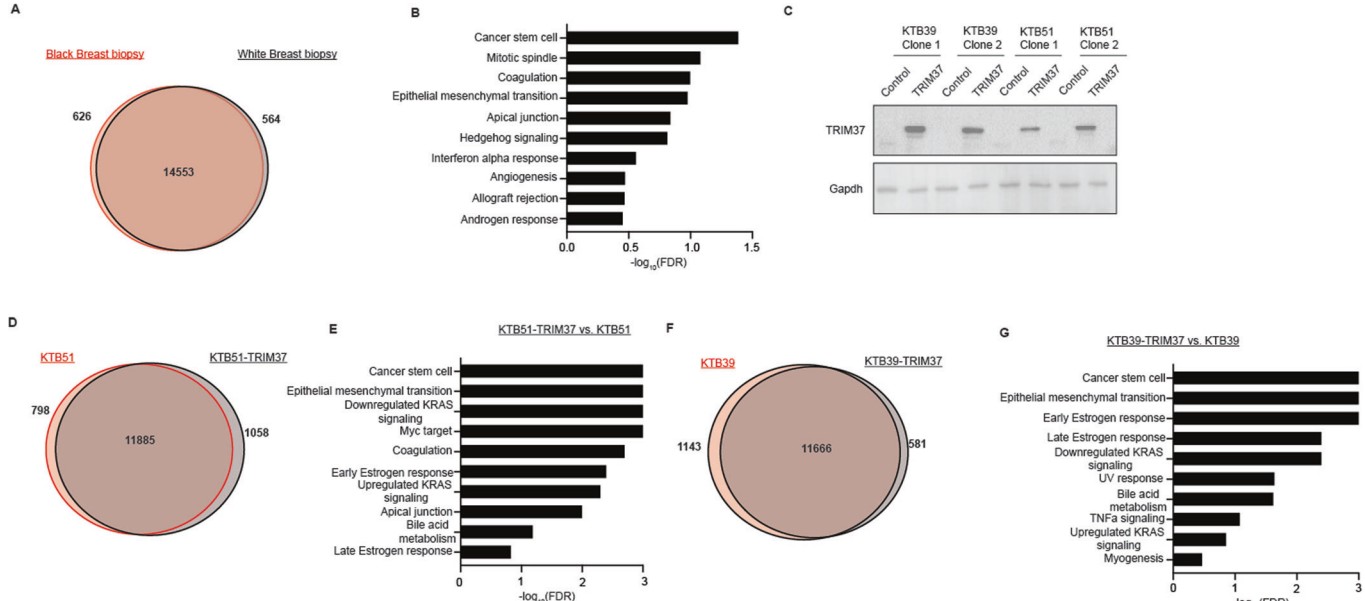

**Figure EV2. TRIM37 drives cancer stemness and epithelial to mesenchymal transition phenotype in Black breast epithelial cells.**

(A) Venn diagram showing the overlap between DEGs in the cancer-free breast tissue samples of parous and premenopausal BW ($n = 3$; ages <45) and WW ($n = 3$; ages <45). (B) The top ten pathways enriched in the cancer-free breast tissue samples of BW relative to WW identified by GSEA are shown. (C) Immunoblot analysis of control and *TRIM37* overexpressing KTB39 and KTB51 cells. Two clones for each cell line are shown. Gapdh was used as a loading control. (D) Venn diagram showing the overlap between differentially expressed genes in control and *TRIM37* overexpressing KTB51 cells. (E) The top ten pathways significantly enriched in *TRIM37* overexpressing KTB51 cells relative to control cells identified by GSEA are shown. (F) Venn diagram showing the overlap between differentially expressed genes in control and *TRIM37* overexpressing KTB39 cells. (G) The top ten pathways enriched in *TRIM37* overexpressing KTB39 cells relative to control cells identified by GSEA are shown.

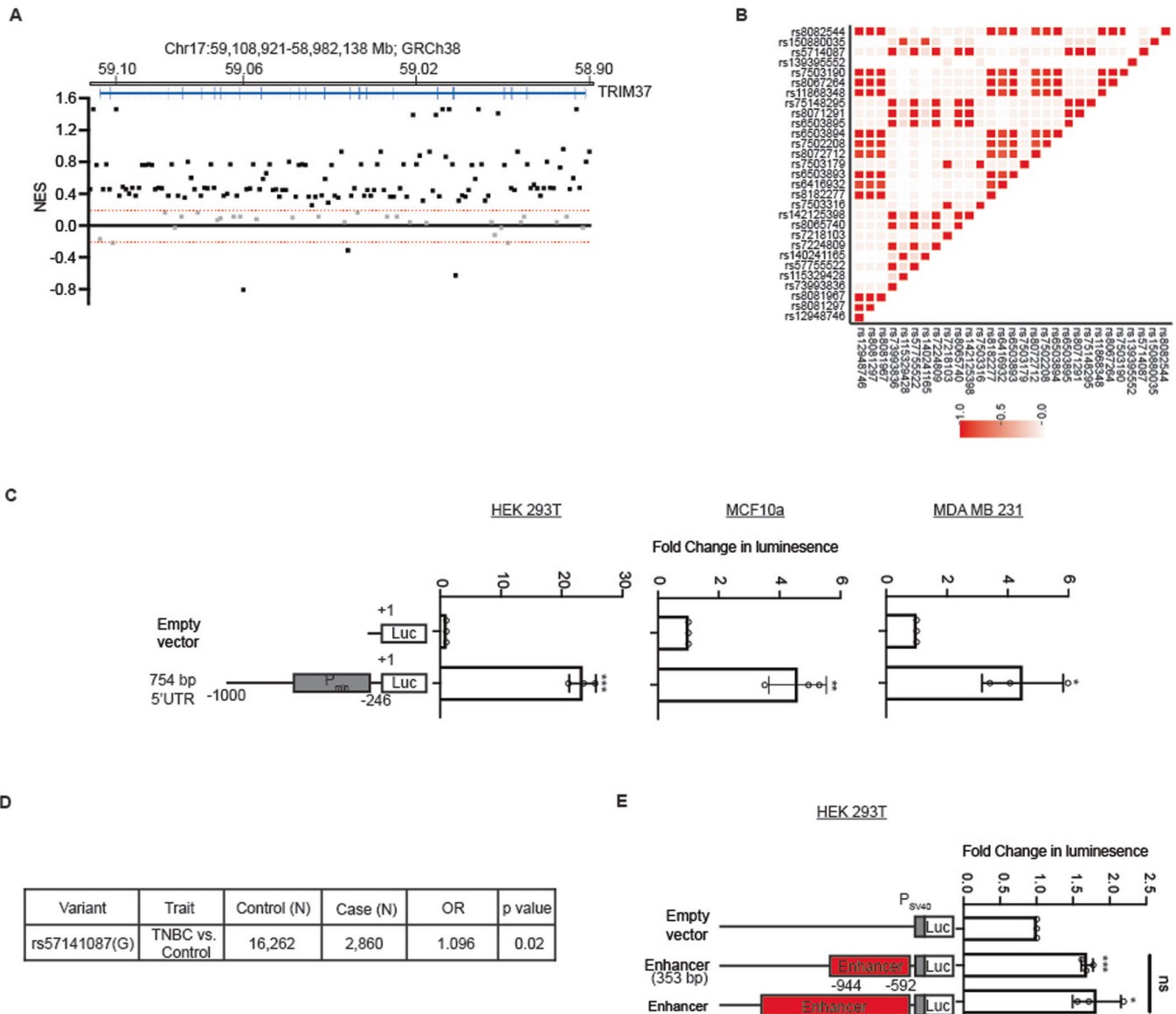

**Figure EV3. rs57141087 increases *TRIM37* expression through modulation of enhancer function.**

(A) Distribution of SNPs across the TRIM37 gene (Chr17:59,108,921–58,982,138 Mb) and their association with *TRIM37* expression in the breast tissue from GTEx. (B) Linkage disequilibrium ($r^2$) heatmap of SNPs in TRIM37 gene in African (Yoruba, Luhya, Gambia, Mende, Esan, Americans of African Ancestry, African Caribbean) population from the 1000 Genome project data. (C) Luciferase reporter assays measure the promoter performance of the TRIM37 minimal promoter (Chr17: 59,107,052–59,106,707) in HEK293T, MCF10a, and MDA MB 231 cells. For HEK293T, ***$P = 6.05 \times 10^{-5}$, MCF10a, **$P = 0.008$, and MDA MB 231, *$P = 0.012$, unpaired t test. Data are mean±SD of biological replicates, n = 3/group. (D) TNBC risk association with the risk allele of rs57141087 using GWAS data for African ancestry (Jia et al, 2024). (E) Luciferase reporter assays measuring the enhancer activity of DNA fragments (Chr17: 59,107,961–59,107,053) in HEK293T cells. The data is normalized to the vector only set to 1 and presented as a fold change in luminescence. of the different fragments in HEK293T. For 353 bp, ***$P = 0.0001$, 909 bp, *$P = 0.012$, and 353 bp vs. 909 bp, $P = 0.539$, unpaired t test. Data are mean±SD of biological replicates, n = 3/group.

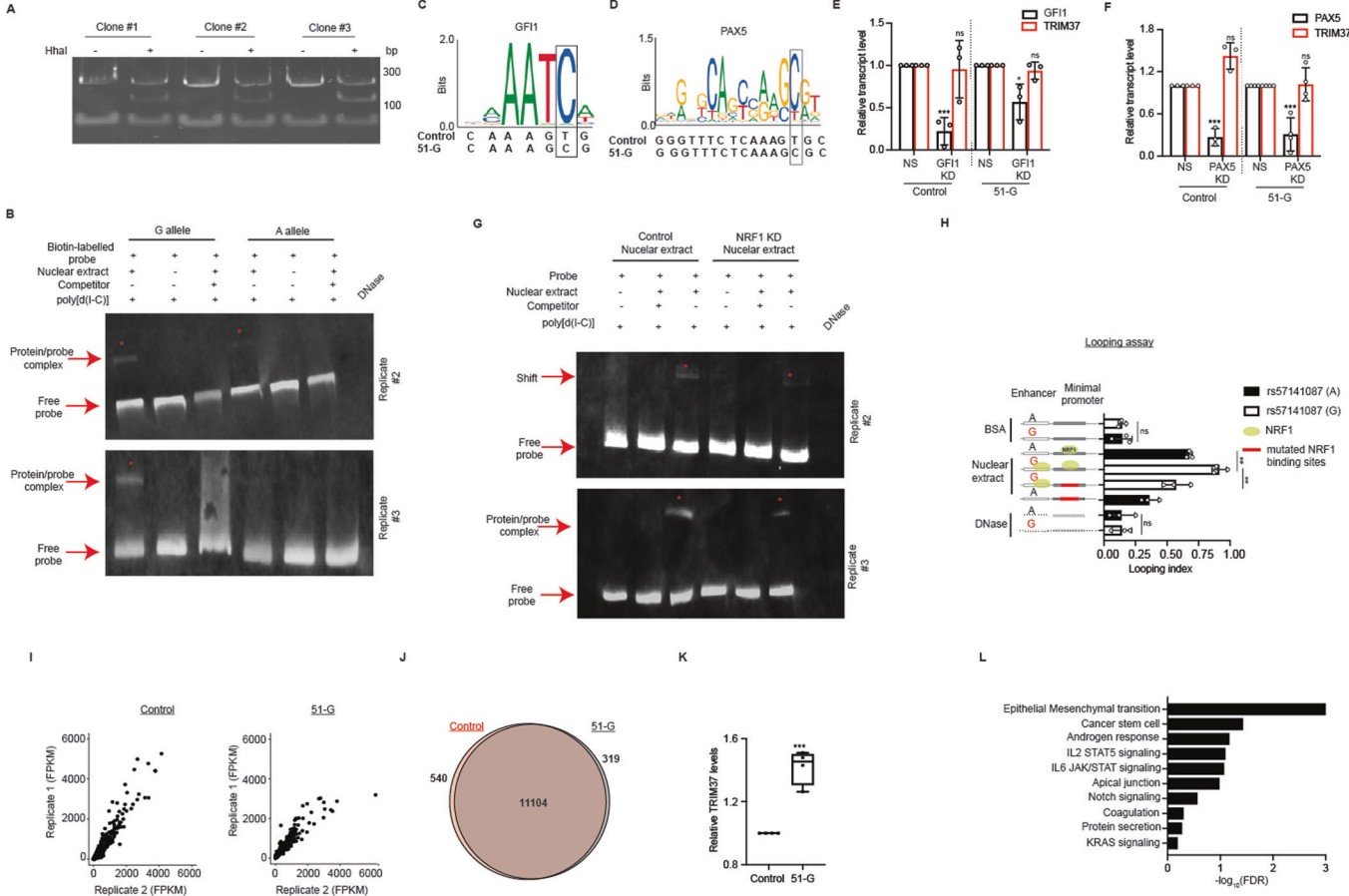

**Figure EV4. NRF1 binds uniquely to the enhancer region harboring the rs57141087 risk allele.**

(**A**) Representative polyacrylamide gel indicates a distinct HhaI digestion pattern for KTB51 clones harboring rs57141087 reference (A) and risk (G) alleles. The data for three different clones is shown. (**B**) Immunoblot for biotin-labeled TRIM37 enhancer fragment with risk (G allele) and reference (A allele) incubated with HEK293T nuclear extract and competitor (unlabeled probe) as indicated. The bound and free DNA fragments are indicated with red arrows, and the shift in the band is indicated (red asterisk, *). The concentrations of nuclear extract (5 µg), Biotin-labeled probes (0.2 pmol), competitor (10 pmol), and poly [d(I-C)] (50 ng/µl) were indicated at the *Top*. The representative images from two different experiments (Replicate #1–2) are shown. (**C, D**) The risk G allele of rs57141087 harboring GFI1 (**C**) and PAX5 (**D**) binding motif in the enhancer region of TRIM37 (Chr17: 59,107,267–59,107,262) is shown. (**E, F**) qRT-PCR monitoring *TRIM37* levels in GFI1 (**E**) and PAX5 (**F**) knockdown in KTB51 cells with A or G allele for rs57141087. Gapdh is used as an endogenous control. For GFI1 knockdown, GFI1 (A), ***P = 0.001, TRIM37 (A), $^{ns}P = 0.831$, GFI1 (G), *P = 0.024, and TRIM37 (G), $^{ns}P = 0.345$. For PAX5 knockdown, PAX5 (A), ***P = 0.0005, TRIM37 (A), $^{ns}P = 0.187$, PAX5 (G), ***P = 0.001, and TRIM37 (G), $^{ns}P = 0.874$, unpaired *t* test. Data are mean±SD of biological replicates, *n* ≥3/group. (**G**) Immunoblot for biotin-labeled TRIM37 enhancer fragment incubated with nuclear extract from control or NRF1 knockdown HEK293T cells and unlabeled probe (competitor) as indicated. The bound and free DNA fragments are indicated with red arrows, and the shift in the band is indicated (red asterisk, *). The concentrations of nuclear extract (5 µg), Biotin-labeled probes (0.2 pmol), competitor (10 pmol), and poly [d(I-C)] (50 ng/µl) were indicated at the *Top*. The representative images from two different experiments (Replicate #1–2) are shown. (**H**) A looping assay measuring the promoter–enhancer interactions for in vitro synthesized DNA fragments (Chr17: 59,107,405-59,106,446) with risk or reference alleles or mutated NRF1 binding motifs in the promoter was used. For BSA, $^{ns}P = 0.947$, NRF1, **P = 0.003, NRF1 vs mutated NRF1, **P = 0.009, and DNase, $^{ns}P = 0.965$, unpaired *t* test. Data are mean±SD of biological replicates, *n* = 3/group. (**I**) Correlation plots of the FPKM read from RNA-seq analysis for each replicate are shown for control (*Left*) and 51-G (*Right*) cells. (**J**) Venn diagram showing the overlap between differentially expressed genes in control and 51-G cells. (**K**) qRT-PCR analysis monitoring *TRIM37* in control and 51-G cells. Gapdh is used as an endogenous control. ***P = 0.0003, unpaired t test. Data are mean ± SD of biological replicates, *n* = 4/group. The boxed areas span the first to the third quartile. The whiskers represent the 15th and 85th percentiles. (**L**) The top ten pathways enriched in 51-G relative to control cells identified by GSEA are shown.

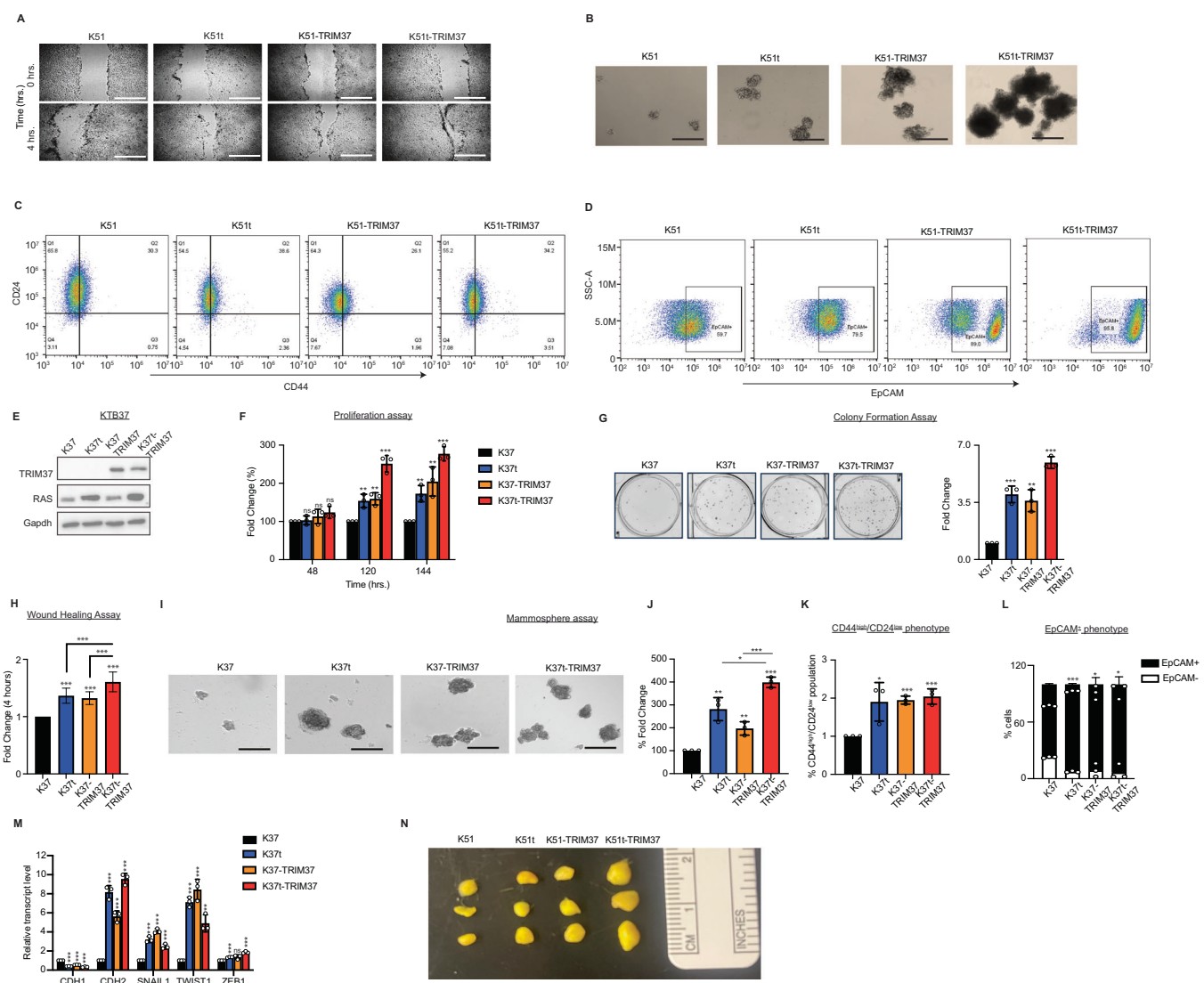

**Figure EV5. TRIM37-directs neoplastic transformations in immortalized epithelial cells.**

(A) Representative phase contrast images of relative migratory abilities for KTB51, K51t, K51-TRIM37, and K51t-TRIM37 after 4 h. (10X, scale bar, 300 μm). (B) Representative images of solid mammospheres formed by KTB51, K51t, K51-TRIM37, and K51t-TRIM37 cells. (10X, scale bar, 300 μm). (C, D) Representative FACS plot showing gating strategy and distribution of stained population for CD24 and CD44 (C) and EpCAM (D) in KTB51, K51t, K51-TRIM37, and K51t-TRIM37 cells. (E) Immunoblots in KTB37 and *TRIM37* derivatives of RAS-transformed KTB37(K37t) cells. Gapdh was the loading control. (F) Relative cell growth for KTB37 and *TRIM37* derivatives of RAS-transformed KTB37 (K37t-TRIM37) cells at indicated times. For 48 h., K37t, $^{ns}P = 0.655$, K37-TRIM37, $^{ns}P = 0.287$, and K37t-TRIM37, $^{ns}P = 0.058$. For 120 h., K37t, $^{**}P = 0.006$, K37-TRIM37, $^{**}P = 0.003$, and K37t-TRIM37, $^{***}P = 0.0003$. For 144 h. K37t, $^{**}P = 0.004$, K37-TRIM37, $^{**}P = 0.009$, and K37t-TRIM37, $^{***}P = 0.0002$, unpaired *t* test. Data are mean ± SD of biological replicates, $n = 3$/group. (G) Representative bright-field images after crystal violet staining show the growth of KTB37, K37t, K37-TRIM37, and K37t-TRIM37 cells. The colonies were quantified (*Right*). For K37t, $^{***}P = 0.0005$, K37-TRIM37, $^{**}P = 0.003$, and K37t-TRIM37, $^{***}P = 1.86*10^{-5}$, unpaired *t* test. Data are mean ± SD of biological replicates, $n = 3$/group. (H) The relative migratory abilities for KTB37, K37t, K37-TRIM37, and K37t-TRIM37 cells were quantitated after 4 h. For K37t, $^{***}P = 1.32*10^{-13}$, K37-TRIM37, $^{***}P = 3.39*10^{-14}$, and K37t-TRIM37, $^{***}P = 1.72*10^{-16}$; K37t vs. K37t-TRIM37, $^{***}P = 4.13*10^{-5}$, K37-TRIM37 vs. K37t-TRIM37, $^{***}P = 1.28*10^{-6}$, unpaired *t* test. Data are mean±SD of biological replicates, $n = 6$/group. (I, J) Representative images (10X, scale bar, 300 μm) (I) and quantitation (J) of solid mammospheres formed by KTB37, K37t, K37-TRIM37, and K37t-TRIM37 cells. The colonies were quantified. For K37t, $^{**}P = 0.003$, K37-TRIM37, $^{**}P = 0.004$, and K37t-TRIM37, $^{***}P = 2.15*10^{-5}$; K37t vs. K37t-TRIM37, $^*P = 0.021$, K37-TRIM37 vs. K37t-TRIM37, $^{***}P = 0.0007$, unpaired *t* test. Data are mean±SD of biological replicates, $n = 3$/group. (K, L) FACS analysis of CD24 and CD44 (K) and EpCAM (L) in KTB37, K37t, K37-TRIM37, and K37t-TRIM37 cells derived from mammospheres in (I, J). For CD44$^{high}$/CD24$^{low}$ in K37t, $^*P = 0.036$, K37-TRIM37, $^{***}P = 0.0001$, and K37t-TRIM37, $^{***}P = 0.0008$. For EpCAM$^+$ in K37t, $^{***}P = 1.99*10^{-5}$, K37-TRIM37, $^*P = 0.028$, and K37t-TRIM37, $^*P = 0.024$, unpaired *t* test. Data are mean±SD of biological replicates, $n = 3$/group. (M) qRT-PCR analysis of EMT markers in KTB37, K37t, K37-TRIM37, and K37t-TRIM37 cells derived from mammospheres in (I, J). Gapdh is used as an endogenous control. For CDH1 in K37t, $^{***}P = 2.03*10^{-7}$, K37-TRIM37, $^{***}P = 1.01*10^{-6}$, and K37t-TRIM37, $^{***}P = 1.58*10^{-5}$. For CDH2 in K37t, $^{***}P = 5.52*10^{-5}$, K37-TRIM37, $^{***}P = 0.0001$, and K37t-TRIM37, $^{***}P = 1.38*10^{-5}$. For SNAI1 in K37t, $^{***}P = 0.0002$, K37-TRIM37, $^{***}P = 2.56*10^{-5}$, and K37t-TRIM37, $^{***}P = 0.0006$. For TWIST1 in K37t, $^{***}P = 3.08*10^{-5}$, K37-TRIM37, $^{***}P = 0.0003$, and K37t-TRIM37, $^{**}P = 0.002$. For ZEB1 in K37t, $^{***}P = 0.0001$, K37-TRIM37, $^{ns}P = 0.051$, and K37t-TRIM37, $^{***}P = 0.0002$, unpaired *t* test. Data are mean±SD of biological replicates, $n = 3$/group. (N) Representative bright-field images showing gross histology of xenografts harvested from NSG mice injected with, K51t, K51-TRIM37, and K51t-TRIM37 cells.

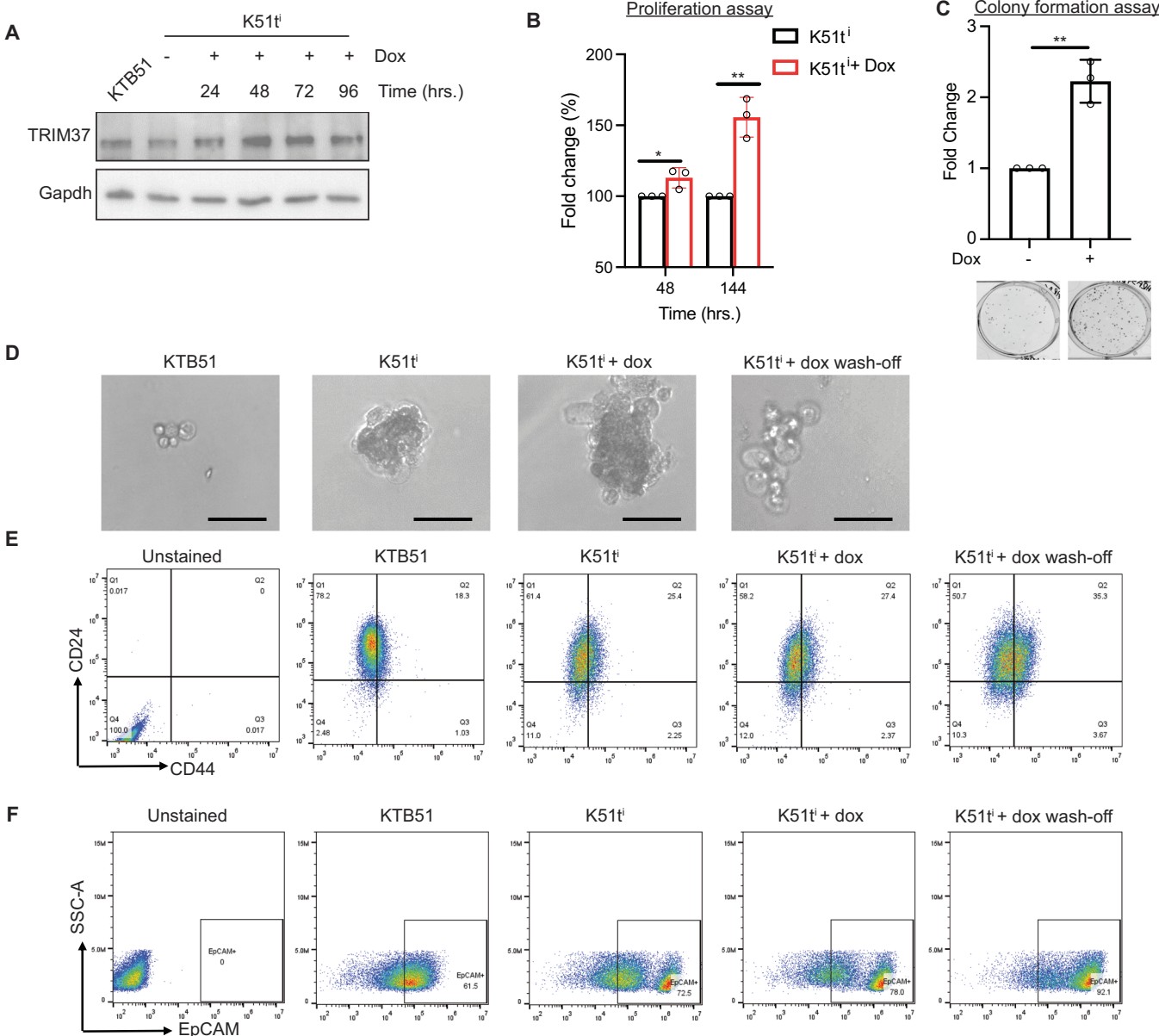

**Figure EV6. TRIM37 is required for breast cancer onset.**

(A) Immunoblot analysis in K51t$^i$ cells following doxycycline (dox) treatment for 24, 48, 72 and 96 h. post-induction. Gapdh was the loading control. (B) Relative cell growth for K51t$^i$ and dox-induced K51t$^i$ at indicated times. For 48 h, *P = 0.034 and 144 h, **P = 0.002, unpaired t test. Data are mean±SD of biological replicates, n = 3/group. (C) The colony formation assay quantitating the growth of K51t$^i$ and dox-induced K51t$^i$. Representative bright-field images after crystal violet staining are shown (*Bottom*). For K51t$^i$ vs. K51t$^i$+dox, **P = 0.003, unpaired t test. Data are mean±SD of biological replicates, n = 3/group. (D) Representative images of solid mammospheres formed by K51t$^i$ and dox-induced K51t$^i$ at indicated times. (×10, scale bar, 300 μm). (E, F) Representative FACS plot showing gating strategy and distribution of stained population for CD24 and CD44 (E) and EpCAM (F) in K51t$^i$ and dox-induced K51t$^i$ at indicated times.

