## [Peer Review File · EMBO Reports]

The TRIM37 variant rs57141087 contributes to triple-negative breast cancer outcomes in Black women

Sanchita Bhatnagar, Rachisan Tihagam, Song Lou, Yuanji Zhao, Kammi Liu, Arjun Singh, Bon Koo, Piotr Przanowski, Jie Li, Xiaosong Huang, Laura Fejerman, Jogender Singh, and Hong Li

Corresponding author(s): Sanchita Bhatnagar (sbbhatnagar@ucdavis.edu)

Review Timeline:

Submission Date:	1st May 24
Editorial Decision:	21st Jun 24
Revision Received:	12th Sep 24
Editorial Decision:	25th Oct 24
Revision Received:	7th Nov 24
Accepted:	12th Nov 24

Editor: Achim Breiling

Transaction Report:

Dear Dr. Bhatnagar,

Thank you for the submission of your manuscript to EMBO reports. I have now received the reports from the three referees that were asked to evaluate your study, which can be found at the end of this email. As you will see, the referees have several comments, concerns, and suggestions, indicating that a major revision of the manuscript is necessary to allow publication of the study in EMBO reports. As the reports are below, and all the concerns need to be addressed, I will not detail them further here.

Given the constructive referee comments, I would like to invite you to revise your manuscript with the understanding that the concerns of the referees must be addressed in the revised manuscript and in a detailed point-by-point response. Acceptance of your manuscript will depend on a positive outcome of a second round of review. It is EMBO reports policy to allow a single round of revision only and acceptance of the manuscript will therefore depend on the completeness of your responses included in the next, final version of the manuscript.

1) a .docx formatted version of the final manuscript text (including legends for main figures, EV figures and tables), but without the figures included. Figure legends should be compiled at the end of the manuscript text.

2) individual production quality figure files as .eps, .tif, .jpg (one file per figure), of main figures and EV figures. Please upload these as separate, individual files upon re-submission.

4) a complete author checklist, which you can download from our author guidelines

(<https://www.embopress.org/page/journal/14693178/authorguide>). Please insert page numbers in the checklist to indicate where the requested information can be found in the manuscript. The completed author checklist will also be part of the RPF.

5) that primary datasets produced in this study (e.g. RNA-seq, ChIP-seq, structural and array data) are deposited in an appropriate public database. If no primary datasets have been deposited, please also state this in a dedicated section (e.g. 'No primary datasets have been generated and deposited'), see below.

The accession numbers and database should be listed in a formal "Data Availability" section (placed after Materials & Methods) that follows the model below. This is now mandatory (like the COI statement). Please note that the Data Availability Section is restricted to new primary data that are part of this study. This section is mandatory. As indicated above, if no primary datasets have been deposited, please state this in this section

Data availability

8) Regarding data quantification and statistics, please make sure that the number "n" for how many independent experiments were performed, their nature (biological versus technical replicates), the bars and error bars (e.g. SEM, SD) and the test used to calculate p-values is indicated in the respective figure legends (also for EV figures and all those in an Appendix). Please also check that all the p-values are explained in the legend, and that these fit to those shown in the figure. Please provide statistical testing where applicable. Please avoid the phrase 'independent experiment', but clearly state if these were biological or technical replicates. Please also indicate (e.g. with n.s.) if testing was performed, but the differences are not significant. In case n=2, please show the data as separate datapoints without error bars and statistics. See also: <http://www.embopress.org/page/journal/14693178/authorguide#statisticalanalysis>

9) Please also note our reference format:

10) We updated our journal's competing interests policy in January 2022 and request authors to consider both actual and perceived competing interests. Please review the policy <https://www.embopress.org/competing-interests> and update your competing interests if necessary. Please name this section 'Disclosure and Competing Interests Statement' and put it after the Acknowledgements section.

11) We now use CRediT to specify the contributions of each author in the journal submission system. CRediT replaces the author contribution section. Please use the free text box to provide more detailed descriptions and do NOT provide your final manuscript text file with an author contributions section. See also our guide to authors: <https://www.embopress.org/page/journal/14693178/authorguide#authorshippinguidelines>

12) We would encourage you to use 'Structured Methods', our new Materials and Methods format. According to this format, the Materials and Methods section should include a Reagents and Tools Table (listing key reagents, experimental models, software, and relevant equipment and including their sources and relevant identifiers), uploaded as separate file, followed by a Methods and Protocols section in which we encourage the authors to describe their methods using a step-by-step protocol format with bullet points, to facilitate the adoption of the methodologies across labs. More information on how to adhere to this format as well as downloadable templates (.doc) for the Reagents and Tools Table can be found in our author guidelines (section 'Structured Methods'):

Please add 5 keywords to the manuscript text and order the manuscript sections like this, using these names:
Title page - Abstract - Keywords - Introduction - Results - Discussion - Methods - Data availability section - Acknowledgements - Disclosure and Competing Interests Statement - References - Figure legends - Expanded View Figure legends

I look forward to seeing a revised version of your manuscript when it is ready. Please let me know if you have questions or comments regarding the revision.

Yours sincerely,

Referee #1:

This is a well-performed study to demonstrate a risk variant rs57141087 leading to higher TRIM37 expression in BW, potentially contributing to breast cancer initiation and progression. I only have several comments and suggestions:

1. In the gain-of-function studies, the authors ectopically expressed TRIM37 only in Black breast epithelial cells (KTB51 and KTB39); they also performed a lot of mechanistic studies and in vivo experiments using KTB51 or its subline. It is unclear if they were to perform gain-of-function studies in a WW breast epithelial cell line (e.g., KTB36), what would be the outcome? In another word, whether increasing the level of TRIM37 is sufficient for these in vitro and in vivo phenotypes (if so, then one would expect to see similar results when overexpressing TRIM37 in a WW cell line, such as KTB36), or whether these TRIM37-related oncogenic changes require additional (cooperating) genetic or epigenetic event(s) from a BW background (i.e., they could only see these oncogenic changes when using Black breast epithelial cells)?

2. In the in vivo xenograft experiments, it is unclear whether the tumors developed from TRIM37-overexpressing KTB51t cells are triple-negative breast tumors? The author should further characterize these tumors and determine their subtype. Without such data, the main conclusion from this work, as reflected in the title (...rs57141087 contributes to triple-negative breast cancer onset and progression ...), lacks sufficient experimental support.

Minor points:

1. In the EMSA experiment shown in Fig. 4C, it is unclear what was exactly being measured here?
2. TRIM37 is amplified in ~9-10% of breast cancers (due to chromosome 17q23 amplification). Is there any difference between BW and WW in terms of its amplification status?
3. A similar study about higher expression of another TRIM family member, TRIM46, in breast cancer driven by SNP rs4971059 was published previously (PMID: 34459501). The authors may want to quote this relevant work.
4. Compared to Fig. 6K, the Ki67 staining shown in Fig. 5K is unconvincing.

Referee #2:

The manuscript titled "A TRIM37 risk variant rs57141087 contributes to triple-negative breast cancer onset and progression in Black women" explores the genetic basis for the higher incidence and aggressiveness of triple-negative breast cancer (TNBC) in Black women. The study identifies a risk variant, rs57141087, in the TRIM37 gene that is prevalent in Black women and elucidates its role in TNBC onset and progression. The manuscript addresses a critical gap in understanding the genetic factors contributing to racial disparities in TNBC. The identification of TRIM37 as a risk factor specific to Black women is novel and significant, adding valuable insights into personalized medicine and potential targeted interventions. The study employs a robust methodology, including meta-analysis of gene-level associations, expression profiling using TCGA datasets, and functional assays (e.g., luciferase reporter assays, ChIP, and RNA-seq). These methods are appropriate and well-executed, providing strong evidence for the conclusions drawn. I only have minor recommendations for improvement before this very important manuscript can be published.

Minor recommendations:

- 1) Ensure that all figures and tables are clearly labeled and referenced in the text. Some figure legends could provide more detail to aid in understanding without referring back to the text. Figure 2 is low resolution clearly visible. Font size is too small to

read.

2) Discuss the limitations of the study, such as potential biases in the cohort selection, limitations of self-reported race, and the need for larger sample sizes.

3) Additional recommended changes are as following. Scale bar for data presented in supplementary Figure 5 and 6 is missing. For the data compared with non-significant results should be labeled as ns. The manuscript should be checked throughout for proper statistical analysis and comparison between the groups should be consistently presented.

Referee #3:

Major concerns:

1. The authors do not provide a reference or any evidence that rs57141087 is associated with breast cancer risk. According to the largest breast cancer GWAS published to date (Michailidou et al, Nature, 2017), there is no evidence that any SNP near TRIM37 falls within a breast cancer risk signal. It is possible that this study did not contain high enough sample numbers of African American ancestry to identify rs57141087. The authors need to provide evidence that the SNP is associated with breast cancer risk.

2. The authors compare the expression of a normal breast cell line from BW and WW. (Fig 2 A-E) and suggest that the differences in expression are attributed to the ancestry of the cell line. This approach is fundamentally flawed - the change of expression could be due to a number of reasons that have nothing to do with ancestry. For example i) There a multiple normal breast epithelial cell types eg. Luminal progenitor vs luminal mature vs basal cells. The cell lines were made by infecting normal epithelial cells (a pool of multiple epithelial cell types) with hTERT. It is possible one cell line is derived from basal epithelial cells and the other luminal epithelial. ii) infection with hTERT is not sufficient to immortalize cell alone. Other genetic alterations must occur and these are likely different between cell lines. iii) difference in parity between individuals from which the cells were derived. Pregnancy and menopause significantly affects breast cancer risk and gene expression. Were the individuals from which the cell lines were of similar parity or menopause status.

3. The association between rs57141087 with expression of TRIM37 in Supplementary Fig3A - what cell type was this eQTL found in? The authors should show that the eQTL signal found in breast tissue (Fig 3D) colocalizes with the breast cancer risk signal (assuming there is one). E.g. using a COLOC or something similar.

4. Hi-C generally doesn't have the resolution to decipher specific chromatin looping between promoters and enhancers above background levels (caused by random collisions). This is especially difficult when the 2 sites are in close genomic distance as is the TRIM37 promoter and the region containing rs57141087. The authors need to show that the interaction signal observed is above background levels. This should be done by comparing the interaction signal between TRIM37 and the SNP region to other nearby restriction fragments. E.g. using chromosome conformation capture and qPCR. Fig 3H. The authors show that the region containing rs57141087 interacts with the TRIM37 promoter. It is not clear if all chromatin interactions in the regions are shown or only those between those 2 sites?

5. Fig 4I. The chromatin looping assay is performed using DNA made in bacteria which doesn't model human chromatin eg. Not bound by histones etc. The standard assay for measuring chromatin interaction frequency in mammalian cell lines is chromosome conformation capture (also known as 3C). The authors should repeat these results in a more biologically relevant system. As mentioned above, chromatin interaction frequency should always be measured relative to background chromatin interactions.

6. Fig 4C. EMSAs are not very quantitative. Please show biological replicates as Supplementary Figures.

Minor concerns:

Page10 "Given that the SNP location determines gene expression and cancer susceptibility (23,24), ...". It's not clear what the authors mean here. I can't find any reference to TRIM37 in ref 23 or 24?

A detailed point-by-point response to the reviewers' concerns, additional experiments, and textual revisions.

Referee #1:

This is a well-performed study to demonstrate a risk variant rs57141087 leading to higher TRIM37 expression in BW, potentially contributing to breast cancer initiation and progression. I only have several comments and suggestions:

1. In the gain-of-function studies, the authors ectopically expressed TRIM37 only in Black breast epithelial cells (KTB51 and KTB39); they also performed a lot of mechanistic studies and in vivo experiments using KTB51 or its subline. It is unclear if they were to perform gain-of-function studies in a WW breast epithelial cell line (e.g., KTB36), what would be the outcome? In another word, whether increasing the level of TRIM37 is sufficient for these in vitro and in vivo phenotypes (if so, then one would expect to see similar results when overexpressing TRIM37 in a WW cell line, such as KTB36), or whether these TRIM37-related oncogenic changes require additional (cooperating) genetic or epigenetic event(s) from a BW background (i.e., they could only see these oncogenic changes when using Black breast epithelial cells)?

We thank the reviewer for an insightful comment. We would like to point out that our previous studies indicated that higher TRIM37 levels are sufficient to drive TRIM37-associated oncogenic events in cellular and *in vivo* models that were derived primarily from White women (Bhatnagar *et al*, 2014; Bhatnagar & Green, 2015; Przanowski *et al*, 2020). Briefly, the ectopic expression of TRIM37 transformed MCF10AT cells, a pre-malignant MCF10A derivative stably expressing activated HRAS, enabled growth in soft agar and tumor formation in mice. Ectopic expression of TRIM37 also transformed HMLER and BPLER cells, immortalized human breast epithelial cell lines derived from White women patients. Thus, while the contribution of race-associated genetic and epigenetic components to TRIM37 function cannot be ruled out, we expect that TRIM37 over-expression will drive tumorigenesis and metastatic events in an ancestry-independent manner.

Nonetheless, to address reviewers' concerns, we have now included new results from experiments assessing the tumorigenic and metastatic potential of TRIM37 expressing derivatives of breast epithelial KTB37 cells derived from White women (**Figure EV5**).

In summary, we stably over-expressed TRIM37 in KTB37 and a RAS expressing premalignant derivative of KTB37, referred to as K37t (*Rebuttal Fig.1A*). Among the different KTB37 derivatives tested, the ectopic expression of TRIM37 potentiated the proliferation and colony formation in KTB37 and K37t cells (*Rebuttal Fig.1B-C*). Next, we performed a series of phenotypic analyses to determine the tumorigenic and metastatic potential of TRIM37-transformed KTB37 cells as described for KTB51, a breast epithelial cell line derived from Black women (**Figure 5**). The scratch-migration assay showed rapid wound closure within four hrs. for TRIM37-transformed KTB37 (K37-TRIM37) and K37t (K37t-TRIM37) cells relative to control cells (*Rebuttal Fig.1D*).

A higher number of mammospheres were formed by KTB37 derivatives expressing TRIM37, with K37t-TRIM37 cells showing the greatest increase in self-renewal phenotype (*Rebuttal Fig.1E*). We next subjected mammospheres to flow cytometry for CD44/CD24 and EpCAM expression to evaluate the CSC and EMT phenotypes respectively. As expected, ~2.05- and ~1.21-fold increase in CD44^{high}/CD24^{low} and EpCAM were observed for K37t-TRIM37 cells, respectively (*Rebuttal Fig.1F-G*). In concert with FACS results, the qRT-PCR analysis of the mammospheres confirmed significantly increased expression of EMT gene signatures in KTB37 derivatives (*Rebuttal Fig.1H*).

Based on the results from the phenotypic assays for KTB37 (*Rebuttal Fig.1*) and KTB51 (**Figure 5**), we conclude that TRIM37 over-expression is sufficient to drive tumorigenesis and metastatic events in immortalized breast epithelial cells.

Rebuttal Figure 1. *TRIM37*-directs neoplastic transformations in immortalized epithelial cells derived from White women. (A) Immunoblots in KTB37 and *TRIM37* derivatives of RAS-transformed KTB37(K37t) cells. Gapdh was the loading control. (B) Relative cell growth for KTB37 and *TRIM37* derivatives of RAS-transformed KTB37 (K37t-*TRIM37*) cells at indicated times. (C) Representative bright-field images after crystal violet staining show the growth of KTB37, K37t, K37-*TRIM37*, and K37t-*TRIM37* cells. The colonies were quantified (*Right*). (D) The relative migratory abilities for KTB37, K37t, K37-*TRIM37*, and K37t-*TRIM37* cells were quantitated after 4 hours. (E) Representative images (*Left*) and quantitation (*Right*) of solid mammospheres formed by KTB37, K37t, K37-*TRIM37*, and K37t-*TRIM37* cells. The colonies were quantified (10X, Scale Bar, 300 μ m). (F-G) FACS analysis of CD24 and CD44 (F) and EpCAM (G) in KTB37, K37t, K37-*TRIM37*, and K37t-*TRIM37* cells derived from mammospheres in (E). (H) qRT-PCR analysis of EMT markers in KTB37, K37t, K37-*TRIM37*, and K37t-*TRIM37* cells derived from mammospheres in (E). Gapdh is used as an endogenous control. Error bars indicate SD and range of at least three biological replicates. *, $p < 0.05$; **, $p < 0.01$, *** $p < 0.001$, ns, not significant.

2. In the *in vivo* xenograft experiments, it is unclear whether the tumors developed from *TRIM37*-overexpressing KTB51t cells are triple-negative breast tumors? The author should further characterize these tumors and determine their subtype. Without such data, the main conclusion from this work, as reflected in the title (...rs57141087 contributes to triple-negative breast cancer onset and progression ...), lacks sufficient experimental support.

We thank the reviewer for the insightful comment. To address the reviewer's concerns, we have further characterized the mouse xenografts derived from KTB51 derivatives for the expression of the Estrogen- α (ER), progesterone receptor (PR), and human epidermal growth factor receptor 2 (HER2). The results from the analysis are now included in the revised **Fig.5** and **Fig.6**.

Briefly, we performed the immunohistochemistry (IHC) based staining for the ER, PR, and HER2 in xenografts derived from mice harboring K51t-*TRIM37*. Our analysis showed no significant staining for ER, PR, and Her2 in K51t-*TRIM37* tumors relative to the positive control (*Rebuttal Fig.2A-C*). We quantitated the ER, PR, or Her2 positive cells for each tumor by counting at least ~25 cells in 20 different fields per section. Consistently, the KTB51tⁱ xenografts from mice fed on a doxycycline diet demonstrated the absence of ER, PR, and Her2 expression relative to the positive control (*Rebuttal Fig.2D-F*).

Collectively, these results confirmed that high *TRIM37* levels in the early stages of transformation can inherently wire breast epithelial cells to accelerated TNBC growth and tumorigenesis.

Rebuttal Figure 2. *TRIM37*-directs neoplastic transformations in immortalized epithelial cells derived from Black women. (A-C) Representative immunohistochemical (IHC) images of ER (A, 20X, Scale Bar, 200 μ m), PR (B, 20X, Scale Bar, 200 μ m), and Her2 (C, 20X, Scale Bar, 200 μ m) for the tumors isolated from the xenograft tumor volume measurements of NSG mice injected with K51t, K51-*TRIM37*, and K51t-*TRIM37* cells (n=6 per group). The manual quantification of the IHC signal is shown for ER (*Bottom*), PR (*Bottom*), and Her2 (*Bottom*) (n~1200-1500 cells per group). (D-F) Representative immunohistochemical (IHC) images of ER (D, 20X, Scale Bar, 200 μ m), PR (E, 20X, Scale Bar, 200 μ m), and Her2 (F, 20X, Scale Bar, 200 μ m) for the tumors isolated from NSG mice injected with KTB51tⁱ and fed on control or dox-enriched diet (n=6 per group). The manual quantification of the IHC signal is shown for ER (*Bottom*), PR (*Bottom*), and Her2 (*Bottom*) (n~1200 cells per group). Error bars indicate SD and range of at least three biological replicates.

Minor points:

1. *In the EMSA experiment shown in Fig. 4C, it is unclear what was exactly being measured here?*

We apologize for the oversight, and we have now revised the text to include the experimental details and conclusion from the experiments (**Pg.13-14, Fig.4C and 4G**). We have also revised the figure legend to explain the experimental design. As recommended by Reviewer 3, we have also included results from the three replicates to support our conclusion (**Fig.EV4B and EV4G**).

Briefly, we evaluated the protein complexes binding capabilities of TRIM37 enhancer DNA fragment in the context of rs57141087 risk and reference allele by electrophoretic mobility shift assay (EMSA). The result showed that the G allele has ~3.92-fold stronger binding capabilities than the A allele (*Rebuttal Fig.3A*).

We next interrogated the physical interaction of NRF1 with the enhancer region harboring the risk allele of SNP rs57141087. The ChIP assay showed a significantly higher enrichment of NRF1 in the enhancer region for 51-G cells relative to 51-A cells (Fig.4F). In concert, the protein binding affinities of DNA fragment harboring risk allele of rs57141087 showed a dramatic decrease when incubated with the nuclear extract from NRF1 knockdown cells relative to control cells (*Rebuttal Fig.3B*).

Rebuttal Figure 3. rs57141087 modulates TRIM37 promoter-enhancer interactions through NRF1 binding. (A) Immunoblot for biotin-labeled TRIM37 enhancer fragment with risk (G allele) and reference (A allele) incubated with HEK293T nuclear extract and competitor (unlabeled probe) as indicated. The bound and free DNA fragments are indicated with red arrows. The concentrations of nuclear extract (5 μ g), Biotin-labeled probes (0.2 pmol), competitor (10 pmol), and poly [d(I-C)] (50 ng/ μ l) were indicated at the *Top*. The representative images from three experiments are shown (*Top, Replicate #1-3*), and the % shift in the band is quantitated (*Bottom*). (B) Immunoblot for biotin-labeled TRIM37 enhancer fragment incubated with nuclear extract from control or NRF1 knockdown HEK293T cells and competitor (unlabeled probe) as indicated. The bound and free DNA fragments are indicated with red arrows. The concentrations of nuclear extract (5 μ g), Biotin-labeled probes (0.2 pmol), competitor (10 pmol), and poly [d(I-C)] (50 ng/ μ l) were indicated at the *Top*. The representative images from three experiments are shown (*Replicate #1-3, Top*), and the % shift in the band is quantitated from three experiments (*Bottom*). Error bars indicate SD and range of at least three biological replicates. *, $p < 0.05$; **, $p < 0.01$, *** $p < 0.001$, ns, not significant.

A

	G allele			A allele			
Biotin-labelled probe	+	+	+	+	+	+	DNase
Nuclear extract	+	-	+	+	-	+	
Competitor	-	-	+	-	-	+	
poly[d(I-C)]	+	+	+	+	+	+	

B

	Control Nuclear extract			NRF1 KD Nuclear extract			
Probe	+	+	+	+	+	+	DNase
Nuclear extract	-	+	+	-	+	+	
Competitor	-	+	-	-	+	-	
poly[d(I-C)]	+	+	+	+	+	+	

2. *TRIM37 is amplified in ~9-10% of breast cancers (due to chromosome 17q23 amplification). Is there any difference between BW and WW in terms of its amplification status?*

We agree with the reviewer's comment that the 17q23 region is amplified in breast cancer (Bhatnagar *et al.*, 2014). To address this comment, we analyzed the TRIM37 copy number in 169 TNBC samples available through TCGA. Notably, there are no significant differences in the TRIM37 copy number between Black (46.03%) and White (43.40%) TNBC women patients (*Rebuttal Table I*).

The results from the analysis are now included in **Appendix Table SI**.

Rebuttal Table I. Analysis of 17q23 copy number in TNBC patients.

TRIM37 copy number in TNBC samples (n=169)			
Parameter	Number of samples	Black women	White women
Deletion	90	33 (52.38%)	57 (53.77%)
Amplification	75	29 (46.03%)	46 (43.40%)
No change	4	1 (1.59%)	3 (2.83%)

3. *A similar study about higher expression of another TRIM family member, TRIM46, in breast cancer driven by SNP rs4971059 was published previously (PMID: 34459501). The authors may want to quote this relevant work.*

We apologize for the oversight and thank the reviewer for the comment. We have revised the text and appropriately cited an important study in the Discussion section (**Pg.19-20**).

Revised Discussion:

“SNP often influences breast cancer prevalence and is associated with susceptibility to breast cancer, as well as clinicopathologic status (Lilyquist *et al.*, 2018; Middha *et al.*, 2023; Stolarova *et al.*, 2023). Besides BRCA1 and BRCA2 mutations that markedly increase breast cancer risk (Antoniou *et al.*, 2010), several low- and moderate-risk susceptibility variants have been identified, including caspase-8 (rs2293554, rs6723097, (Park *et al.*, 2016)), TIMP-2 (rs7501477, (Peterson *et al.*, 2009)), and FSCN1 (rs56156320, rs3801004, (Wang *et al.*, 2017)). Additionally, SNP plays a critical role in phenotypic variation and disease susceptibility through effects on gene expression, such as PDE4DIP (rs12124527, (Kim *et al.*, 2012)), and FTO (rs9939609, (Karra *et al.*, 2013)). Interestingly, polymorphic variants in the tripartite motif-containing (TRIM) superfamily members have been linked to several human diseases, highlighting their clinical significance. For example, a recent study identified the breast cancer risk locus SNP rs4971059 that regulates the TRIM46-HDAC1 axis to modulate genes involved in DNA replication and repair (Zhang *et al.*, 2021). Notably, TRIM37 polymorphic variants have previously been associated with acute myeloid leukemia (rs11656413, (Hess *et al.*, 2017), schizophrenia (rs4968363, rs2877926, (Hall *et al.*, 2020)) and inflammatory bowel (Rivas *et al.*, 2011).”

4. *Compared to Fig. 6K, the Ki67 staining shown in Fig. 5K is unconvincing.*

To address reviewers' concerns, we have improved the Ki67 staining for the xenograft tumors harvested from the NSG mice injected with KTB51 and TRIM37 derivatives of RAS-transformed KTB51 cells (*Rebuttal Fig.4*). The new results are included in the revised **Fig.5L**.

Rebuttal Figure 4. Representative images of Ki67 (K, 40X, Scale Bar, 60μm) for the tumors isolated from NSG mice injected with KTB51 and TRIM37 derivatives of RAS-transformed KTB51 cells.

Referee #2:

The manuscript titled "A TRIM37 risk variant rs57141087 contributes to triple-negative breast cancer onset and progression in Black women" explores the genetic basis for the higher incidence and aggressiveness of triple-negative breast cancer (TNBC) in Black women. The study identifies a risk variant, rs57141087, in the TRIM37 gene that is prevalent in Black women and elucidates its role in TNBC onset and progression. The manuscript addresses a critical gap in understanding the genetic factors contributing to racial disparities in TNBC. The identification of TRIM37 as a risk factor specific to Black women is novel and significant, adding valuable insights into personalized medicine and potential targeted interventions. The study employs a robust methodology, including meta-analysis of gene-level associations, expression profiling using TCGA datasets, and functional assays (e.g., luciferase reporter assays, ChIP, and RNA-seq). These methods are appropriate and well-executed, providing strong evidence for the conclusions drawn. I only have minor recommendations for improvement before this very important manuscript can be published.

Minor recommendations:

- 1) *Ensure that all figures and tables are clearly labeled and referenced in the text. Some figure legends could provide more detail to aid in understanding without referring back to the text. Figure 2 is low resolution clearly visible. Font size is too small to read.*

We apologize for the oversight. We have revised the manuscript to label and reference all the figures and tables in the text correctly labeled. As recommended by the reviewer, we have also replaced all the panels in Fig. 2 with high-resolution images. We have included additional experimental details for all the figure legends.

- 2) *Discuss the limitations of the study, such as potential biases in the cohort selection, limitations of self-reported race, and the need for larger sample sizes.*

As recommended by the reviewer, we have included the limitations and potential biases in the Discussion section (**Pg.18**).

- 3) *Additional recommended changes are as follows: The scale bar for the data presented in supplementary Figures 5 and 6 is missing. The data compared with non-significant results should be labeled as ns. The manuscript should be checked throughout for proper statistical analysis, and the comparison between the groups should be consistently presented.*

As recommended by the reviewer, we have included scale bars in Extended Version Figures 5 and 6 and appropriately indicated the results from the statistical analyses in all the figures.

Referee #3:

Major concerns:

1. *The authors do not provide a reference or any evidence that rs57141087 is associated with breast cancer risk. According to the largest breast cancer GWAS published to date (Michailidou et al, Nature, 2017), there is no evidence that any SNP near TRIM37 falls within a breast cancer risk signal. It is possible that this study did not contain high enough sample numbers of African American ancestry to identify rs57141087. The authors need to provide evidence that the SNP is associated with breast cancer risk.*

We interpret that the reviewer refers to GWAS studies where SNP rs57141087 was not considered the dominant causal variant for breast cancer. We would like to point out that not all causal variants can be accounted for based on the statistical analyses (van de Bunt et al, 2015) (van de Bunt *et al.*, 2015). Furthermore, the underrepresentation of African ancestry and predominance of data collected from the White populations biases analyses and interpretation of the data. Notably, Michailidou et al. and Zhang et al. included datasets that are predominantly samples from European and Asian ancestry, and no samples from African ancestry are included (Michailidou *et al*, 2017; Zhang *et al*, 2020). As such, our analysis of these datasets showed no association for SNP rs57141087 with breast cancer or TNBC risk.

In contrast, our analysis of previously published GWAS studies of breast cancer, including 2,860 cases and 16,262 controls of African ancestry (Jia *et al*, 2024), identified a significant association between rs57141087 and TNBC risk at $p=0.02$, with odd ratios of 1.096 (*Rebuttal Table II*). The rs57141087 also correlated with breast cancer risk at $p=0.029$, with odd ratios of 1.047 (*Rebuttal Table II*). Furthermore, we would like to point out that we include a comprehensive functional analysis of rs57141087 to demonstrate its association with breast cancer risk (**Figure 4**).

We thank the reviewer for the insightful comment that helped us strengthen our conclusion regarding SNP rs57141087 as a new breast cancer risk variant. The revised manuscript includes the new data (**Fig.EV3D**).

Rebuttal Table II.

Table for referee with unpublished data and its description has been removed upon request by the authors.

2. The authors compare the expression of a normal breast cell line from BW and WW. (Fig 2 A-E) and suggest that the differences in expression are attributed to the ancestry of the cell line. This approach is fundamentally flawed - the change of expression could be due to a number of reasons that have nothing to do with ancestry. For example i) There are multiple normal breast epithelial cell types eg. Luminal progenitor vs luminal mature vs basal cells. The cell lines were made by infecting normal epithelial cells (a pool of multiple epithelial cell types) with hTERT. It is possible one cell line is derived from basal epithelial cells and the other luminal epithelial. ii) infection with hTERT is not sufficient to immortalize cell alone. Other genetic alterations must occur and these are likely different between cell lines. iii) difference in parity between individuals from which the cells were derived. Pregnancy and menopause significantly affects breast cancer risk and gene expression. Were the individuals from which the cell lines were of similar parity or menopause status.

We agree with the reviewer's concern regarding the use of immortalized cell lines and their genetic differences due to parity and menopause status. We would like to thank the reviewer for his constructive comments, which have enabled us to strengthen our conclusion further.

To address reviewers' concerns, we utilized paraffin-archived normal breast tissue samples that were obtained from parous, pre-menopausal white and black women (The Susan G. Komen Tissue Bank; Indiana University Simon Comprehensive Cancer Center). These women were cancer-free, and donors were recruited under a protocol approved by the Indiana University Institutional Review Board (#101103097) and completed questionnaires with reproductive histories, demographic information, early life, and lifestyle habits (*Rebuttal Table III*).

Rebuttal Table III: Characteristics of paraffin-archived normal breast tissue.

Cancer-free paraffin-archived normal breast tissue (n=6)						
Parameter	Sample #1	Sample #2	Sample #3	Sample #4	Sample #5	Sample #6
Racial identity	Black	Black	Black	White	White	White
Age (<45 Years)	Yes	Yes	Yes	Yes	Yes	Yes
Menopausal status	Pre-	Pre-	Pre-	Pre-	Pre-	Pre-
Parity	Yes	Yes	Yes	Yes	Yes	Yes
Diagnosed with BC	No	No	No	No	No	No
Ever smoked	Yes	No	No	Yes	No	No
Test Positive for Genetic Risk	No	No	No	N/A	No	No
BRCA1	No	No	No	No	No	No
BRCA2	No	No	No	No	No	No
Blood Relatives Cancer	N/A	No	No	Yes	No	Yes
Height (ft)	5	5	5	5	5	5

The differential gene expression (DEG) analysis revealed 2714 upregulated and 2239 downregulated genes at an FDR cut-off of <0.1 in Black compared with White breast tissue (*Rebuttal Fig.5A*). Interestingly, among the 4953 DEGs, 609 were previously identified as TRIM37-TS (GSE136617), revealing TRIM37-dependent molecular alterations in BW (*Rebuttal*

Fig.5B). Hierarchical clustering showed separation of transcriptomic profiles by ancestry into two distinct race-specific clusters (*Rebuttal Fig.5C*). To explore TRIM37-driven functional pathways in BW, we performed Gene Set Enrichment Analysis (GSEA) for upregulated genes, which identified the cancer stem cell (CSC) and epithelial to mesenchymal transitional (EMT) among the enriched gene sets (*Rebuttal Fig.5D*). The integration of CSC and EMT gene signatures showed a strong association between high CSC and EMT gene signatures and poor survival for TNBC patients with a hazard ratio of 1.88 ($p=0.033$) and 1.9 ($p=0.031$), respectively (*Rebuttal Fig.5E*).

The results from the analysis are now included in the revised **Figure 2**.

Rebuttal Figure 5. African ancestry influences the TRIM37-associated transcriptional signatures in normal, cancer-free breast tissue. (A) Volcano plot illustrates differential gene expression in the cancer-free paraffin-archived breast tissue samples of parous and premenopausal BW ($n=3$; ages <45) and WW ($n=3$; ages <45) women. *Red* are significantly upregulated genes ($n=2,714$), *Blue* are significantly downregulated genes ($n=2,239$), and *Grey* are genes not significantly changed ($n=11,448$). FDR < 0.1. (B) Venn diagram showing the overlap between differentially expressed genes in breast tissue from BW identified in (A) and TRIM37-regulated genes (GSE136617). (C) Hierarchical clustering of median-centered TRIM37-regulated genes in cancer-free BW ($n=3$) and WW ($n=3$) breast tissue. Each colored line in the dendrogram identifies a different gene. (D) Enrichment plots for cancer stem cells (*Top*) and EMT (*Bottom*) gene signatures enriched in cancer-free BW ($n=3$) breast tissue identified through GSEA analysis of RNA-seq data. (E) Kaplan-Meier analysis of survival in patients with low (*Black*) or high (*Red*) TRIM37-regulated CSC (*Top*) and EMT (*Bottom*) gene signatures enriched in cancer-free BW ($n=3$) breast tissue identified in (B). The number of surviving patients at 0, 50, 100, and 150-month time points is indicated below the graph.

A

B

C

D

E

3. The association between rs57141087 with expression of TRIM37 in Supplementary Fig3A - what cell type was this eQTL found in? The authors should show that the eQTL signal found in breast tissue (Fig 3D) colocalizes with the breast cancer risk signal (assuming there is one). E.g. using a COLOC or something similar.

We apologize for missing the details about the source tissue and have now revised the figure legend to include the information. As recommended by the reviewer, we analyzed data from the GWAS studies of breast cancer, including 2,860 TNBC cases and 16,262 controls of African ancestry (Jia *et al.*, 2024). Our analysis of the available datasets identified a significant association between rs57141087 and TNBC risk at $p=0.02$, with odd ratios of 1.096 (*Rebuttal Table II*). The rs57141087 also correlated with breast cancer risk at $p=0.029$, with odd ratios of 1.047 (*Rebuttal Table II*). Further, we would like to point out that we include a comprehensive functional analysis of rs57141087 to demonstrate its association with breast cancer risk (**Figure 4**).

We thank the reviewer for the insightful comment that helped us strengthen our conclusion regarding SNP rs57141087 as a new breast cancer risk variant. The revised manuscript includes the new data (**Fig.EV3D**).

Rebuttal Table II.

Table for referee with unpublished data and its description has been removed upon request by the authors.

4. Hi-C generally doesn't have the resolution to decipher specific chromatin looping between promoters and enhancers above background levels (caused by random collisions). This is especially difficult when the 2 sites are in close genomic distance as is the TRIM37 promoter and the region containing rs57141087. The authors need to show that the interaction signal observed is above background levels. This should be done by comparing the interaction signal between TRIM37 and the SNP region to other nearby restriction fragments. E.g. using chromosome conformation capture and qPCR. Fig 3H. The authors show that the region containing rs57141087 interacts with the TRIM37 promoter. It is not clear if all chromatin interactions in the regions are shown or only those between those 2 sites?

We thank the reviewer for the insightful comment. As recommended by the reviewer, the Hi-C data was replaced by the quantitative chromosome conformation capture (3C-qPCR) assay. Using 3C-qPCR, we directly examined rs57141087 interaction with the TRIM37 minimal promoter *in vivo* (Rebouissou et al, 2022). Using an anchor primer in the minimal promoter and probes in the region harboring rs57141087, we observed a strong chromatin interaction between the rs57141087 and the TRIM37 promoter in 51-G cells but not in 51-A cells (*Rebuttal Fig.6*).

Results from 3C-qPCR are consistent with the *in vitro* looping assay examining the TRIM37 promoter-enhancer interactions. Briefly, we carried out an *in vitro* looping assay using a 960 bp fragment harboring either a risk or reference allele in the enhancer region and minimal promoter with wild-type or mutated NRF1 binding sites. As shown in **Fig.EV4H**, the loss of NRF1 binding in the TRIM37 locus significantly lowered the looping index for the construct harboring reference allele relative to the risk allele at rs57141087. As expected, substituting a nuclear extract with BSA or prior treatment with DNase abolished the interactions (**Fig.EV4H**). Together, these results confirm a loop structure formation between the SNP rs57141087-containing enhancer region and the TRIM37 promoter.

The results from the analysis are now included in the revised **Figure 4H**.

Rebuttal Figure 6. rs57141087 modulates TRIM37 promoter-enhancer interactions. 3C-qPCR analysis across the ~5,500-bp TRIM37 locus harboring risk and reference allele of SNP rs57141087 in 51-G and 51-A cells, respectively. The schematic shows the TRIM37 gene structure with PCR primers used for 3C-qPCR assay (*Top*). Error bars indicate SD and range of at least three biological replicates. *, $p < 0.05$; **, $p < 0.01$, *** $p < 0.001$, ns, not significant.

5. Fig 4I. The chromatin looping assay is performed using DNA made in bacteria which doesn't model human chromatin eg. Not bound by histones etc. The standard assay for measuring chromatin interaction frequency in mammalian cell lines is chromosome conformation capture (also known as 3C). The authors should repeat these results in a more biologically relevant system. As mentioned above, chromatin interaction frequency should always be measured relative to background chromatin interactions.

We thank the reviewer for the insightful comment, which helped us strengthen our conclusion regarding the chromatin interaction between the region harboring risk SNP rs57141087 and the TRIM37 minimal promoter.

As recommended by the reviewer, we performed quantitative chromosome conformation capture (3C-qPCR) assays to examine rs57141087 interaction with the TRIM37 minimal promoter *in vivo* (Rebouissou *et al.*, 2022). Using 3C-qPCR, we directly examined rs57141087 interaction with the TRIM37 minimal promoter *in vivo* (Rebouissou *et al.*, 2022). Using an anchor primer in the minimal promoter and probes in the region harboring rs57141087, we observed a strong chromatin interaction between the rs57141087 and the TRIM37 promoter in 51-G cells but not in 51-A cells (*Rebuttal Fig. 7*).

The results from the analysis are now included in the revised **Figure 4H**.

Rebuttal Figure 7. rs57141087 modulates TRIM37 promoter-enhancer interactions. 3C-qPCR analysis across the ~5,500-bp TRIM37 locus harboring risk and reference allele of SNP rs57141087 in 51-G and 51-A cells, respectively. The schematic shows the TRIM37 gene structure with PCR primers used for 3C-qPCR assay (*Top*). Error bars indicate SD and range of at least three biological replicates. *, $p < 0.05$; **, $p < 0.01$, *** $p < 0.001$, ns, not significant.

6. Fig 4C. EMSAs are not very quantitative. Please show biological replicates as Supplementary Figures.

As recommended by the reviewer, we have added immunoblots following EMSA from three different experiments. To further clarify our results, we have revised the text and figure legends (*Rebuttal Fig. 8*).

Rebuttal Figure 8. rs57141087 modulates TRIM37 promoter-enhancer interactions through NRF1 binding. (A) Immunoblot for biotin-labeled TRIM37 enhancer fragment with risk (G allele) and reference (A allele) incubated with HEK293T nuclear extract and competitor (unlabeled probe) as indicated. The bound and free DNA fragments are indicated with red arrows. The concentrations of nuclear extract (5 μ g), Biotin-labeled probes (0.2 pmol), competitor (10 pmol), and poly [d(I-C)] (50 ng/ μ l) were indicated at the *Top*. The representative images from three experiments are shown (*Top, Replicate #1-3*), and the % shift in the band is quantitated (*Bottom*). (B) Immunoblot for biotin-labeled TRIM37 enhancer fragment incubated with nuclear extract from control or NRF1 knockdown HEK293T cells and competitor (unlabeled probe) as indicated. The bound and free DNA fragments are indicated with red arrows. The concentrations of nuclear extract (5 μ g), Biotin-labeled probes (0.2 pmol), competitor (10 pmol), and poly [d(I-C)] (50 ng/ μ l) were indicated at the *Top*. The representative images from three experiments are shown (Replicate #1-3, *Top*), and the % shift in the band is quantitated from three experiments (*Bottom*). Error bars indicate SD and range of at least three biological replicates. *, $p < 0.05$; **, $p < 0.01$, *** $p < 0.001$, ns, not significant.

Minor concerns:

Page10 "Given that the SNP location determines gene expression and cancer susceptibility (23,24), ...". It's not clear what the authors mean here. I can't find any reference to TRIM37 in ref 23 or 24?

We apologize for the confusion. We have made textual changes to emphasize that the location of SNPs determines gene expression and cancer susceptibility (**Pg.10**).

Antoniou AC, Wang X, Fredericksen ZS, McGuffog L, Tarrell R, Sinilnikova OM, Healey S, Morrison J, Kartsonaki C, Lesnick T *et al* (2010) A locus on 19p13 modifies risk of breast cancer in BRCA1 mutation carriers and is associated with hormone receptor-negative breast cancer in the general population. *Nat Genet* 42: 885-892

Bhatnagar S, Gazin C, Chamberlain L, Ou J, Zhu X, Tushir JS, Virbasius CM, Lin L, Zhu LJ, Wajapeyee N *et al* (2014) TRIM37 is a new histone H2A ubiquitin ligase and breast cancer oncoprotein. *Nature* 516: 116-120

Bhatnagar S, Green MR (2015) TRIMming down tumor suppressors in breast cancer. *Cell Cycle* 14: 1345-1346

Hall LS, Medway CW, Pain O, Pardin AF, Rees EG, Escott-Price V, Pocklington A, Bray NJ, Holmans PA, Walters JTR *et al* (2020) A transcriptome-wide association study implicates specific pre- and post-synaptic abnormalities in schizophrenia. *Hum Mol Genet* 29: 159-167

Hess M, Lenz S, Blatte TJ, Bullinger L, Binder H (2017) Partitioned learning of deep Boltzmann machines for SNP data. *Bioinformatics* 33: 3173-3180

Jia G, Ping J, Guo X, Yang Y, Tao R, Li B, Ambs S, Barnard ME, Chen Y, Garcia-Closas M *et al* (2024) Genome-wide association analyses of breast cancer in women of African ancestry identify new susceptibility loci and improve risk prediction. *Nat Genet* 56: 819-826

Karra E, O'Daly OG, Choudhury AI, Youssef A, Millership S, Neary MT, Scott WR, Chandarana K, Manning S, Hess ME *et al* (2013) A link between FTO, ghrelin, and impaired brain food-cue responsivity. *J Clin Invest* 123: 3539-3551

Kim S, Cho H, Lee D, Webster MJ (2012) Association between SNPs and gene expression in multiple regions of the human brain. *Transl Psychiatry* 2: e113

Lilyquist J, Ruddy KJ, Vachon CM, Couch FJ (2018) Common Genetic Variation and Breast Cancer Risk-Past, Present, and Future. *Cancer Epidemiol Biomarkers Prev* 27: 380-394

Michailidou K, Lindstrom S, Dennis J, Beesley J, Hui S, Kar S, Lemacon A, Soucy P, Glubb D, Rostamianfar A *et al* (2017) Association analysis identifies 65 new breast cancer risk loci. *Nature* 551: 92-94

Middha P, Wang X, Behrens S, Bolla MK, Wang Q, Dennis J, Michailidou K, Ahearn TU, Andrulis IL, Anton-Culver H *et al* (2023) A genome-wide gene-environment interaction study of breast cancer risk for women of European ancestry. *Breast Cancer Res* 25: 93

Park HL, Ziogas A, Chang J, Desai B, Bessonova L, Garner C, Lee E, Neuhausen SL, Wang SS, Ma H *et al* (2016) Novel polymorphisms in caspase-8 are associated with breast cancer risk in the California Teachers Study. *BMC Cancer* 16: 14

Peterson NB, Beeghly-Fadiel A, Gao YT, Long J, Cai Q, Shu XO, Zheng W (2009) Polymorphisms in tissue inhibitors of metalloproteinases-2 and -3 and breast cancer susceptibility and survival. *Int J Cancer* 125: 844-850

Przanowski P, Lou S, Tihagam RD, Mondal T, Conlan C, Shivange G, Saltani I, Singh C, Xing K, Morris BB *et al* (2020) Oncogenic TRIM37 Links Chemoresistance and Metastatic Fate in Triple-Negative Breast Cancer. *Cancer Res* 80: 4791-4804

Rebouissou C, Sallis S, Forne T (2022) Quantitative Chromosome Conformation Capture (3C-qPCR). *Methods Mol Biol* 2532: 3-13

Rivas MA, Beaudoin M, Gardet A, Stevens C, Sharma Y, Zhang CK, Boucher G, Ripke S, Ellinghaus D, Burtt N *et al* (2011) Deep resequencing of GWAS loci identifies independent rare variants associated with inflammatory bowel disease. *Nat Genet* 43: 1066-1073

Stolarova L, Kleiblova P, Zemankova P, Stastna B, Janatova M, Soukupova J, Achatz MI, Ambrosone C, Apostolou P, Arun BK *et al* (2023) ENIGMA CHEK2gether Project: A Comprehensive Study Identifies Functionally Impaired CHEK2 Germline Missense Variants Associated with Increased Breast Cancer Risk. *Clin Cancer Res* 29: 3037-3050

van de Bunt M, Cortes A, Consortium I, Brown MA, Morris AP, McCarthy MI (2015) Evaluating the Performance of Fine-Mapping Strategies at Common Variant GWAS Loci. *PLoS Genet* 11: e1005535

Wang CQ, Tang CH, Wang Y, Jin L, Wang Q, Li X, Hu GN, Huang BF, Zhao YM, Su CM (2017) FSCN1 gene polymorphisms: biomarkers for the development and progression of breast cancer. *Sci Rep* 7: 15887

Zhang H, Ahearn TU, Lecarpentier J, Barnes D, Beesley J, Qi G, Jiang X, O'Mara TA, Zhao N, Bolla MK *et al* (2020) Genome-wide association study identifies 32 novel breast cancer susceptibility loci from overall and subtype-specific analyses. *Nat Genet* 52: 572-581

Zhang Z, Liu X, Li L, Yang Y, Yang J, Wang Y, Wu J, Wu X, Shan L, Pei F *et al* (2021) SNP rs4971059 predisposes to breast carcinogenesis and chemoresistance via TRIM46-mediated HDAC1 degradation. *EMBO J* 40: e107974

Dear Dr. Bhatnagar,

Thank you for the submission of your revised manuscript to our editorial offices. As you know, I have received the reports from the referees that I asked to re-evaluate the study, you will find again below. Referee #1 now supports publication of the study, whereas referee #3 has remaining concerns and suggestions to improve the study. Going through your further revision plan (preliminary p-b-p-response), I think the remaining points of the referee #3 will be adequately addressed. Referee #2 basically supported already publication of the previous version of the manuscript and was not involved again.

I thus invite you to revise the manuscript further as indicated in your revision plan. Please also provide a final p-b-p-response addressing the remaining points.

Moreover, I have these editorial requests I ask you to address in a final revised manuscript:

- Please provide a final title with not more than 100 characters (including spaces).
- There are author name discrepancies. It is Rachisan Djake Tihagam in the manuscript text file but Rachisan Tihagam in the submission system, Kammi Song-Yan Liu in the text but Kammi Liu in the system, Arjun Tushir Singh in the manuscript but Arjun Singh in the system, and Jogender Tushir-Singh in the text but Jogender Singh in the submission system. Please check and make sure that identical names are used.
- Please provide the abstract written in present tense throughout.
- Please combine the references and the dataset references into one reference section (named 'References'). See: <https://www.embopress.org/page/journal/14693178/authorguide#referencesformat>

Moreover, the data citation callout "Data ref: The Genotype-Tissue Expression (GTEx)" in the text does not have any matching entries in the reference list. Please check.

- Please make sure that all the funding information is also entered into the online submission system and that it is complete and similar to the one in the acknowledgement section of the manuscript text file. E.g. NCI P30CA093373 and the UVA Cancer Training Grant are presently missing in the submission system. Please check.
- Please label the EV figures according to the nomenclature 'Figure EVx' (in the figure itself).
- Please make sure that the number "n" for how many independent experiments were performed, their nature (biological versus technical replicates), the bars and error bars (e.g. SEM, SD) and the test used to calculate p-values is indicated in the respective figure legends (main, EV and Appendix figures). Please also check that all the p-values are explained in the legend, and that these fit to those shown in the figure. Please provide statistical testing where applicable. Please avoid the phrase 'independent experiment', but clearly state if these were biological or technical replicates. Please also indicate (e.g. with n.s.) if testing was performed, but the differences are not significant. In case n=2, please show the data as separate datapoints without error bars and statistics. See also: <http://www.embopress.org/page/journal/14693178/authorguide#statisticalanalysis>

If n<5, please show single datapoints for diagrams. It seems that presently some diagrams have no (e.g. 3D) or only partial stats (error bars missing e.g. in 1A-C) or the 'ns' is missing. In 5D the 'ns' is covered by bars. Moreover:

- Please note that the error bars not shown in Figs 1 and EV 1; However, error bar definition is provided in legend of respective figures.
- Please note that the exact p values are not provided in the legends of figures 1A-D, 1G, 1I, 1J, 3C, 3E, 3H, 3I, 4B, 4C, 4E-I, 5B-J, 6B-E, EV1A, EV1F, EV3C, EV3E, EV4E, EV4F, EV4H, EV4K, EV5F-H, EV5J-M, EV6B, EV6C.
- Please indicate the statistical test used for data analysis in the legends of figures 1A-G, 1I, 1K, 2A, 2D, 2F-I, 3C-E, 3H, 3I, 4B, 4C, 4E-K, 5B-J, 6B-E, EV1A-D, EV1F-J, EV3C-E, EV4E, EV4F, EV4H, EV4K, EV5F-H, EV5J-M, EV6B, EV6C.
- Please note that in figures EV1A, EV1D, EV1F-J, EV3C, EV3E, EV4E, EV4F, EV4H, EV4K, EV6B-C; there is a mismatch between the annotated p values in the figure legend and the annotated p values in the figure file that should be corrected.
- Please note that the measure of center for the error bars needs to be defined in the legends of figures 3C, 3H, 3I, 4B, 4C, 4E-I, 5B-J, 5M-O, 6B-E, 6H-J, EV3C, EV3E, EV4E, EV4F, EV4H, EV5F-H, EV5J-M, EV6B, EV6C.
- Please note that the box plots need to be defined in terms of minima, maxima, centre, bounds of box and whiskers, and percentile in the legend of figure EV4K.
- Please note that information related to n is missing in the legend of figure EV4K.
- Please note that the box plots need to be defined in terms of centre in the legends of figures 1D-E, 1G, 1I-J, 3E, EV1D, EV1F-J.
- Please note that the red asterisk (*) symbols are not define in figures 4C, 4G, EV4B, EV4G.
- Please note that axis gap is not labelled appropriately in figure 5H.

- Please add to each legend (main, EV figures, where applicable) a 'Data Information' section explaining the statistics used or providing information regarding replicates and scales. See:

- Please remove the legend for APPENDIX TABLEs from the manuscript text file.

- Please upload Tables I and II as EV Tables using the names 'Table EV1' and 'Table EV2'. Please update any callouts accordingly and also use these names in the table files.

- Table III (Minor allele frequency (MAF) for TRIM37 associated Snps) is a dataset. Please name this Dataset EV1 and upload the original excel sheet using this name with a legend on the first TAB. Please change all callouts to 'Dataset EV1'.

- Please add Table IV (list of primers and shRNA) directly to the reagents and tools table and remove it from the manuscript files. Please update any callouts and add callouts to the reagents and tools table were applicable.

- The Data Availability section (DAS) is restricted to datasets generated in this study and deposited. Already published or accessible datasets not generated in this study should not be mentioned here (but in the respective sections of the Methods). Please remove the referee token from the DAS and make sure the dataset is public latest on the publication date of the manuscript. Moreover, please clearly indicate here that image source data for this manuscript can be found at the BioImage database and add a direct link for this.

- Thank you for providing the requested source data. Please upload this as one folder per figure (with all files for one figure in one folder and ZIPed together).

- Please have the final manuscript text carefully proofread by a native speaker.

In addition, I would need from you uploaded separately:

- a short, two-sentence summary of the manuscript (not more than 35 words).

- two to four short (!) bullet points highlighting the key findings of your study (two lines each).

- a schematic summary figure as separate file that provides a sketch of the major findings (not a data image) in jpeg or tiff format (with the exact width of 550 pixels and a height of not more than 400 pixels) that can be used as a visual synopsis on our website.

Best,

Referee #1:

The authors have addressed reviewers' concerns and the manuscript has been further strengthened.

Minor typo:

page 18 "The immunohistochemistry (IHC)-based staining for the Estrogen- α (ER)", the word "receptor" is missing here.

Referee #3:

The authors have not provided sufficient evidence that rs57141087 is associated with breast cancer risk. This locus was not identified by Jia et al, as a genome-wide significant risk signal. The authors have reanalyzed data from Jia et al and have shown rs57141087 to be nominally significant ($p=0.02$), however, it is likely this association would not hold up after multiple testing correction.

The functional analysis largely supports an African ancestry-specific association with expression but there is insufficient evidence to suggest that this affects breast risk.

Moreover, the applicants have not provided sufficient evidence that the eQTL signal co-localises with the breast cancer risk signal. eQTLs are widespread and therefore overlap between GWAS and eQTL signals is likely to occur by chance. To mitigate false positive findings, it is therefore important to show that the same genetic signal underlies gene expression and disease susceptibility. Several statistical colocalization approaches (e.g. COLOC) have been developed that the authors could have used. As it stands, the authors have demonstrated overlap but not colocalization.

Response to Reviewers' comments:

The authors have not provided sufficient evidence that rs57141087 is associated with breast cancer risk. This locus was not identified by Jia et al, as a genome-wide significant risk signal. The authors have reanalyzed data from Jia et al and have shown rs57141087 to be nominally significant ($p=0.02$), however, it is likely this association would not hold up after multiple testing correction.

We respectfully disagree with the reviewer and believe that the reviewer misinterpreted our statement. Nonetheless, to address the reviewer's concern, we collaborated with biostatistician Dr. Hong Li, an expert in statistical methods for evaluating diagnostic tests, modeling and evaluating biomarkers, medical decision-making, personalized medicine in disease diagnosis, prognosis, and treatment, and the design and analysis of clinical trials.

We would like to point out that this dataset has been previously analyzed by other researchers (Jia et al, 2024), and our analysis focused exclusively on a single hypothesis regarding the association between rs57141087 and TNBC risk. Unlike the original GWAS, we did not conduct multiple comparisons or test numerous hypotheses on this dataset. Our analysis was done independently and only once for a specific research question. Therefore, no p-value adjustment for multiple comparisons is necessary, as our work did not involve repeated testing on the data. The reported p-value of 0.02 directly reflects the significance of this focused analysis.

We would like to emphasize that GWAS routinely tests thousands of genetic variants (SNPs) across the genome to find associations with a particular disease or trait. For instance, if a study tests 1 million SNPs for association with a disease, the probability of finding at least one statistically significant result purely by chance is high. Without adjustment, a p-value of 0.05 would result in 50,000 false positives on average. Therefore, p-value adjustments, such as the Bonferroni correction or false discovery rate (FDR) control, are applied to account for multiple comparisons. However, this is not valid for the hypothesis-driven analysis of rs57141087 association with TNBC risk using the dataset provided in the Jia et al. study.

The functional analysis largely supports an African ancestry-specific association with expression but there is insufficient evidence to suggest that this affects breast risk. Moreover, the authors have not provided sufficient evidence that the eQTL signal co-localises with the breast cancer risk signal. eQTLs are widespread and therefore overlap between GWAS and eQTL signals is likely to occur by chance. To mitigate false positive findings, it is therefore important to show that the same genetic signal underlies gene expression and disease susceptibility. Several statistical colocalization approaches (e.g. COLOC) have been developed that the authors could have used. As it stands, the authors have demonstrated overlap but not colocalization.

We agree with the reviewer that colocalization analysis, such as COLOC, will strengthen the association between the rs57141087, TRIM37 expression and breast cancer risk in

Black women. However, the data inaccessibility limits our ability to run the analysis. We would like to point out that the individual-level gene expression data for the meta-analysis done in the original GWAS study is unavailable (Jia *et al.*, 2024). As such, the colocalization analysis is not plausible for large-scale GWAS, as only summary statistics are available.

Also, COLOC and other colocalization analysis methods are proportional approaches. For example, Gay *et al.* conducted a COLOC analysis using GWAS data for 142 samples and eQTL data from 117 samples (Gay *et al.*, 2020). However, the genotype and transcriptomic data is only available for small number of normal breast tissue from Black women in the GTEx and 1000 genome datasets (Table I). As such, the small sample size contributes to bias towards the null hypothesis and limits the power of analysis.

Table I: Sample size included in GTEx, 1000 genome, and GWAS (Gay *et al.*, 2020).

Study	Black	White
GTEX	6	58
E-GEUV-1 (1000 genome)	86	89
GCST90296722 (GWAS study)	19,122	0

The small number of samples added multiple complications to the COLOC analysis, as outlined below and discussed previously (Deng & Pan, 2020; Giambartolomei *et al.*, 2014; Zhang *et al.*, 2024):

- 1) The interpretation of the posterior probabilities requires caution. For example, low PP4 and high values of PP0, PP1, and/or PP2 may not indicate evidence against colocalization due to the limited power.
- 2) It can be argued that genotype/transcriptomic data from other tissues can be used to increase sample size. However, like many diseases, breast cancer manifests its phenotype exclusively in breast tissues; therefore, colocalization results will be dependent on the expression dataset used. In addition to identifying the causal genes, the identification of tissue specificity for the molecular effects underlying GWAS signals is a key outcome.
- 3) Given the samples' admixed ancestry, ancestry-based adjustments are needed in GTEx to improve the power of the analysis and characterize the impact of population structure on GWAS colocalization.
- 4) A key assumption in COLOC is that only one causal variant exists for each trait in a genomic locus (Giambartolomei *et al.*, 2014), inconsistent with the recently discovered widespread allelic heterogeneity.
- 5) High-density genotyping and/or accurate use of imputation techniques are key to increasing the sample number. However, the imputation quality, such as poorly imputed SNPs, can add bias towards null hypothesis.

We would also like to highlight that this study demonstrates higher TRIM37 expression in Black women's breast tissue. We support this conclusion through multiple lines of evidence using genomic (Fig.1), transcriptomic (Fig.2), and functional (Fig.3-6) data included in the paper. We also demonstrate that genetically introducing the risk rs57141087 increases the tumorigenic potential and induces a transcriptomic program favoring EMT and CSC phenotype (Fig.4). Notably, we have previously shown that TRIM37 drives tumorigenesis, chemoresistance, and metastasis in TNBC tumors (Bhatnagar *et al*, 2014; Bhatnagar & Green, 2015; Przanowski *et al*, 2020).

Together, we comprehensively demonstrate that higher expression of TRIM37 linked to the risk allele rs57141087 predisposes Black women to TNBC.

Bhatnagar S, Gazin C, Chamberlain L, Ou J, Zhu X, Tushir JS, Virbasius CM, Lin L, Zhu LJ, Wajapeyee N *et al* (2014) TRIM37 is a new histone H2A ubiquitin ligase and breast cancer oncoprotein. *Nature* 516: 116-120

Bhatnagar S, Green MR (2015) TRIMming down tumor suppressors in breast cancer. *Cell Cycle* 14: 1345-1346

Deng Y, Pan W (2020) A powerful and versatile colocalization test. *PLoS Comput Biol* 16: e1007778

Gay NR, Gloudemans M, Antonio ML, Abell NS, Balliu B, Park Y, Martin AR, Musharoff S, Rao AS, Aguet F *et al* (2020) Impact of admixture and ancestry on eQTL analysis and GWAS colocalization in GTEx. *Genome Biol* 21: 233

Giambartolomei C, Vukcevic D, Schadt EE, Franke L, Hingorani AD, Wallace C, Plagnol V (2014) Bayesian test for colocalisation between pairs of genetic association studies using summary statistics. *PLoS Genet* 10: e1004383

Jia G, Ping J, Guo X, Yang Y, Tao R, Li B, Ambs S, Barnard ME, Chen Y, Garcia-Closas M *et al* (2024) Genome-wide association analyses of breast cancer in women of African ancestry identify new susceptibility loci and improve risk prediction. *Nat Genet* 56: 819-826

Przanowski P, Lou S, Tihagam RD, Mondal T, Conlan C, Shivange G, Saltani I, Singh C, Xing K, Morris BB *et al* (2020) Oncogenic TRIM37 Links Chemoresistance and Metastatic Fate in Triple-Negative Breast Cancer. *Cancer Res* 80: 4791-4804

Zhang W, Lu T, Sladek R, Li Y, Najafabadi H, Dupuis J (2024) SharePro: an accurate and efficient genetic colocalization method accounting for multiple causal signals. *Bioinformatics* 40

Dr. Sanchita Bhatnagar
University of California Davis
Medical Microbiology and Immunology
1275 Health Sciences
Davis, CA 95616
United States

Dear Dr. Bhatnagar,

I am very pleased to accept your manuscript for publication in the next available issue of EMBO reports. Thank you for your contribution to our journal.

Please make sure that the GEO dataset is public latest at the day of online publication of the study.

Yours sincerely,
